# An extra-erythrocyte role of haemoglobin body in chondrocyte hypoxia adaption

Feng Zhang[1,14 ✉], Bo Zhang[2,3,4,14], Yuying Wang[1], Runmin Jiang[5], Jin Liu[1], Yuexian Wei[2,4], Xinyue Gao[2,4], Yichao Zhu[2,4], Xinli Wang[6], Mao Sun[7], Junjun Kang[8], Yingying Liu[8], Guoxing You[9], Ding Wei[10], Jiajia Xin[11], Junxiang Bao[12], Meiqing Wang[13], Yu Gu[1], Zhe Wang[1], Jing Ye[1], Shuangping Guo[1], Hongyan Huang[3] & Qiang Sun[2,4 ✉]

Although haemoglobin is a known carrier of oxygen in erythrocytes that functions to transport oxygen over a long range, its physiological roles outside erythrocytes are largely elusive[1,2]. Here we found that chondrocytes produced massive amounts of haemoglobin to form eosin-positive bodies in their cytoplasm. The haemoglobin body (Hedy) is a membraneless condensate characterized by phase separation. Production of haemoglobin in chondrocytes is controlled by hypoxia and is dependent on KLF1 rather than the HIF1/2α pathway. Deletion of haemoglobin in chondrocytes leads to Hedy loss along with severe hypoxia, enhanced glycolysis and extensive cell death in the centre of cartilaginous tissue, which is attributed to the loss of the Hedy-controlled oxygen supply under hypoxic conditions. These results demonstrate an extra-erythrocyte role of haemoglobin in chondrocytes, and uncover a heretofore unrecognized mechanism in which chondrocytes survive a hypoxic environment through Hedy.

$O_2$ is an indispensable metabolic substrate for numerous reactions and essential for cell survival[3–5]. The oxygen supply to most mammalian cells is dependent on the continuous delivery of $O_2$ through the vascular system by haemoglobin in red blood cells (RBCs). By contrast, cartilage tissue is uniquely avascular, and the oxygen required by chondrocytes within cartilages diffuses from the surrounding tissue[6]. During embryonic development, fetal growth plates expand in the absence of blood vessels, leading to enhanced hypoxia in the centre of the cartilage mould[7,8]. However, the mechanism in which chondrocytes adapt to the hypoxic environment remains largely unknown[6].

With a limited oxygen supply, chondrocytes mainly rely on glycolysis rather than mitochondrial oxidative phosphorylation to produce energy, which is controlled by the hypoxia-induced factor (HIF) signalling pathway[9,10]. Deletion of the gene encoding HIF1α, a key player of HIF signalling, unfreezes mitochondrial oxidative phosphorylation and increases oxygen consumption, resulting in severe hypoxia and massive chondrocyte death[9,10]. Therefore, sustained activation of hypoxia signalling at a proper level is essential for the survival of cartilage cells. However, it is unclear how chondrocytes manage to maintain intracellular oxygen homeostasis. Here we provided evidence that, in response to hypoxia, chondrocytes produced large quantities of haemoglobin to form membraneless bodies (termed Hedy in this study) within their cytoplasm, which is essential for the survival of chondrocytes in the avascular fetal growth plate under a hypoxic environment.

## Eosin-positive structure in chondrocyte

When carefully checking the cartilage growth plates of neonatal mice, we observed a type of eosin-positive structure in the hypertrophic chondrocytes. The size and shape of the structures were similar to those of RBCs in the bone marrow as confirmed by scanning electron microscopy (Fig. 1a). The eosin-positive structures were also detected in the hypertrophic chondrocytes of other cartilaginous tissues, such as the ribs and calcaneus of mice (Fig. 1b). In addition to hypertrophic chondrocytes, these structures were also present in the chondrocytes from the resting and proliferative zones of mouse cartilage, although they were irregular (Fig. 1c). Moreover, similar structures were detected in human cartilage (Fig. 1d). Thus, the eosin-positive structure might be a common feature of chondrocytes irrespective of the source and species.

## Haemoglobin body in chondrocyte

To determine the components of the eosin-positive structures, laser-based microdissection was performed on the hypertrophic

[1]Department of Pathology, School of Basic Medicine and Xijing Hospital, State Key Laboratory of Cancer Biology, Air Force Medical Center, The Fourth Military Medical University, Xi'an, China. [2]Frontier Biotechnology Laboratory, Beijing Institute of Biotechnology, Academy of Military Medical Science; Research Unit of Cell Death Mechanism, 2021RU008, Chinese Academy of Medical Science, Beijing, China. [3]Department of Oncology, Beijing Shijitan Hospital of Capital Medical University, Beijing, China. [4]Nanhu Laboratory, Jiaxing, China. [5]Department of Thoracic Surgery, Tangdu Hospital, The Fourth Military Medical University, Xi'an, China. [6]Department of Orthopedics, Xijing Hospital, The Fourth Military Medical University, Xi'an, China. [7]Department of Biochemistry and Molecular Biology, The Fourth Military Medical University, Xi'an, China. [8]Department of Neurobiology, The Fourth Military Medical University, Xi'an, China. [9]Institute of Health Service and Transfusion Medicine, Academy of Military Medical Sciences, Beijing, China. [10]Department of Cell Biology, National Translational Science Center for Molecular Medicine, State Key Laboratory of Cancer Biology, The Fourth Military Medical University, Xi'an, China. [11]Department of Blood Transfusion, Xijing Hospital, The Fourth Military Medical University, Xi'an, China. [12]Department of Aerospace Hygiene, The Fourth Military Medical University, Xi'an, China. [13]Department of Oral Anatomy and Physiology, School of Stomatology, The Fourth Military Medical University, Xi'an, China. [14]These authors contributed equally: Feng Zhang, Bo Zhang. ✉e-mail: zhf1975@fmmu.edu.cn; sunq@bmi.ac.cn

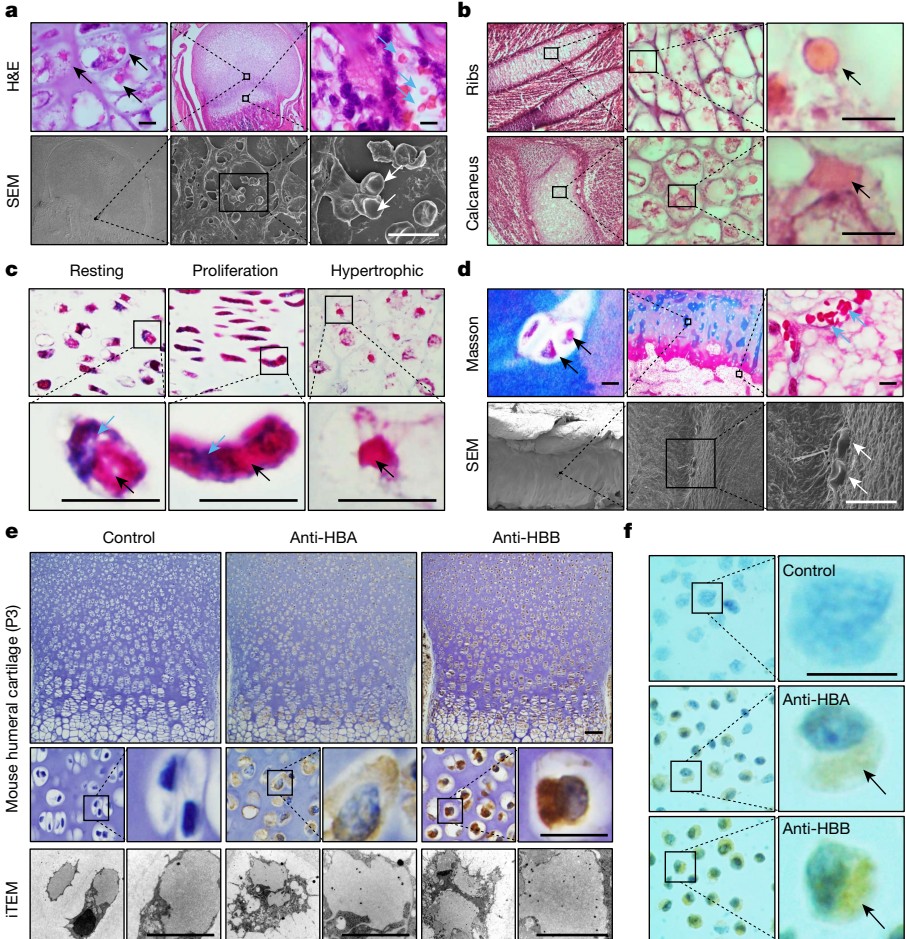

**Fig. 1 | Eosin-positive structures consisting of haemoglobin in the cartilaginous tissues. a**, Haematoxylin and eosin (H&E) staining indicate eosin-positive structures (black arrows) in the hypertrophic chondrocytes of the cartilaginous growth plate of P7 mice (zoomed-in images of the middle and right top panels), which were confirmed by scanning electron microscopy (SEM) (white arrows; bottom panels). The blue arrows indicate RBCs in the bone marrow. Scale bars, 10 µm. **b**, The eosin-positive structures (black arrows) in hypertrophic chondrocytes of ribs (top panels) and calcaneus (bottom panels) of E15.5 mice. Scale bars, 10 µm. **c**, The presence of eosinophilic structures (black arrows) in chondrocytes of the resting, proliferative and hypertrophic zones of growth plates from E18.5 mice by Masson trichrome staining.

The blue arrows indicate nuclei. Scale bars, 10 µm. **d**, The presence of eosinophilic structures (black arrows) in the non-hypertrophic zone of femoral articular cartilage from a man 16 years of age (zoomed-in images of the middle and right top panels), which were confirmed by SEM (white arrows; bottom panels). The blue arrows indicate RBCs in the bone marrow. Scale bars, 10 µm. **e**, Immunohistochemistry (top panels) and immunoelectron microscope staining (iTEM; bottom panels) of HBA and HBB in mouse humeral cartilage from P3 mice. Scale bars, 50 µm (top and middle row panels) and 5 µm (iTEM). **f**, The expression of HBA and HBB in chondrocytes isolated from P7 mouse humeral cartilage as detected by immunohistochemistry staining. The black arrows indicate haemoglobin. Scale bar, 50 µm.

chondrocytes from the cartilage of 6 day postnatal (P6) mice, followed by mass spectrometry (Extended Data Fig. 1a,c). To our surprise, the top hits were predominantly the haemoglobin-β subunit (HBB) (Extended Data Fig. 1b). This was in agreement with SDS–PAGE results that indicated that the chondrocytes contained a considerable amount of protein that shifted at a rate similar to haemoglobin in RBCs (Extended Data Fig. 1d). Mass spectrometry and western blot confirmed the presence of the HBB and the haemoglobin-α subunit (HBA) (Extended Data Fig. 1e–h). RNA sequencing and quantitative proteomic analysis showed that the two haemoglobin subunits were not equivalently expressed, with HBB expressed more than HBA at the protein level (HBA:HBB ≈ 3:5) (Extended Data Fig. 1i–k). Next, the expression of haemoglobin in chondrocytes was examined in situ by immunohistochemistry and immunoelectron microscope staining of mouse humeral cartilage as shown in Fig. 1e, which indicated a clear pattern of cytoplasmic staining for both HBA and HBB. Similar results were obtained in chondrocytes isolated from mouse cartilage (Fig. 1f). Together, these results demonstrated that chondrocytes of cartilaginous

tissue produced a massive amount of haemoglobin, mainly HBB, to form a kind of cytoplasmic eosin-positive structure that we call Hedy hereafter.

## Phase separation of haemoglobin

To further explore the nature of Hedy, transmission electron microscopy analysis was performed. As shown in Fig. 2a, the Hedy structure was clearly a type of membraneless condensate isolated in the cytoplasm of chondrocytes. Consistently, hyposmotic rupture of the chondrocytes led to the release of the eosin-positive structure as an isolated individual body (Fig. 2b). Next, we set out to test whether the formation of Hedy was a result of protein condensation by phase separation. Expression of *Hbb* alone, or together with *Hba*, gave rise to cytoplasmic condensates (also termed foci) in different cell lines (Fig. 2c,d and Extended Data Fig. 2a). The foci were not enriched in lipid and nucleic acids as indicated by fluorescent staining (Extended Data Fig. 2b,c). Taking HBB as an example, we explored the characteristics of the cytoplasmic

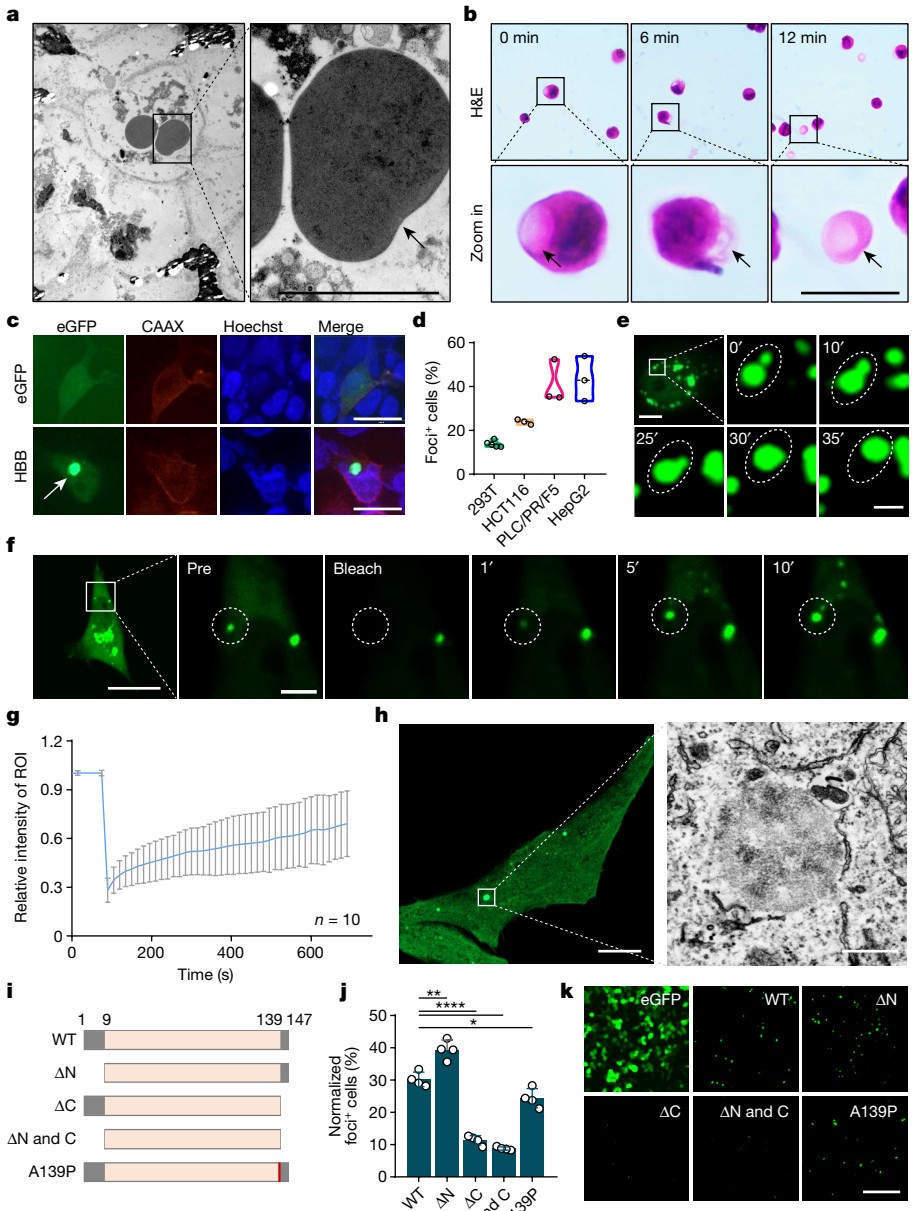

**Fig. 2 | Phase separation promotes Hedy formation. a**, Transmission electron microscopy showed the condensate structures (black arrow) in the cytoplasm of hypertrophic chondrocytes from the growth plates of E14.5 mice. Scale bar, 5 µm. **b**, Hyposmotic rupture by ddH$_2$O incubation led to the release of the eosin-positive structures (black arrows) from the humeral cartilage chondrocytes of P7 mice. Scale bar, 10 µm. **c**, Representative images of eGFP and HBB–eGFP expressed in 293T cells. CAAX in red indicates the cell membrane. Hoechst in blue indicates cell nuclei. The arrow indicates foci formed by HBB–eGFP. Scale bars, 20 µm. **d**, Quantification of foci formation in different cell lines. Data are mean ± s.d. of 3 or more fields with more than 300 cells analysed each. **e**, Image sequence shows an example of fusion of two HBB–eGFP foci. Scale bars, 10 µm (original view) and 2 µm (zoomed-in views). **f**, Image sequence shows an example of the fluorescence recovery after photobleaching experiment of

HBB–eGFP foci. Scale bars, 10 µm (original view) and 5 µm (zoomed-in views). **g**, Quantification of fluorescence recovery after photobleaching data (mean ± s.e.m.; $n = 10$ experiments) for HBB–eGFP foci. ROI, region of interest. **h**, Fluorescence (left) and electron transmission microscopic (right) images of HBB–eGFP condensate by correlative light and electron microscopy. Scale bars, 10 µm (left) and 500 nm (right). **i**, Schematic demonstration of *Hbb* mutants with truncations in single or combined disorder motifs (grey boxes) (ΔN, ΔC, and ΔN and ΔC), or with a point mutation of A139P (red bar). **j,k**, Quantification (**j**) and representative images (**k**) of foci formation of the indicated HBB–eGFP mutants in 293T cells. $n > 200$ cells over 3 biologically independent experiments. Error bars represent s.e.m. $P$ values were calculated using two-tailed Student's $t$-test (**j**). *$P < 0.05$, **$P < 0.01$ and ****$P < 0.0001$. Scale bar, 200 µm.

foci. Timelapse microscopy demonstrated that these foci readily fused with each other (Fig. 2e) and rapidly recovered from photobleaching in cells (Fig. 2f,g), suggesting a dynamic nature of the foci resembling that of protein condensates by phase separation. The characteristics of liquid droplet formation were validated in vitro with purified untagged HBB protein (Extended Data Fig. 3a–d) and GFP-tagged protein as well

(Supplementary Fig. 2a,b). Moreover, these cytoplasmic condensates were not enclosed with a bi-leaflet membrane as determined by correlative light electron microscopy and immunoelectron transmission microscopy (Fig. 2h and Extended Data Fig. 4a–j). Sequence analysis identified two short intrinsically disordered regions (IDRs), which are often enriched in phase-separating proteins[11,12], located at the N

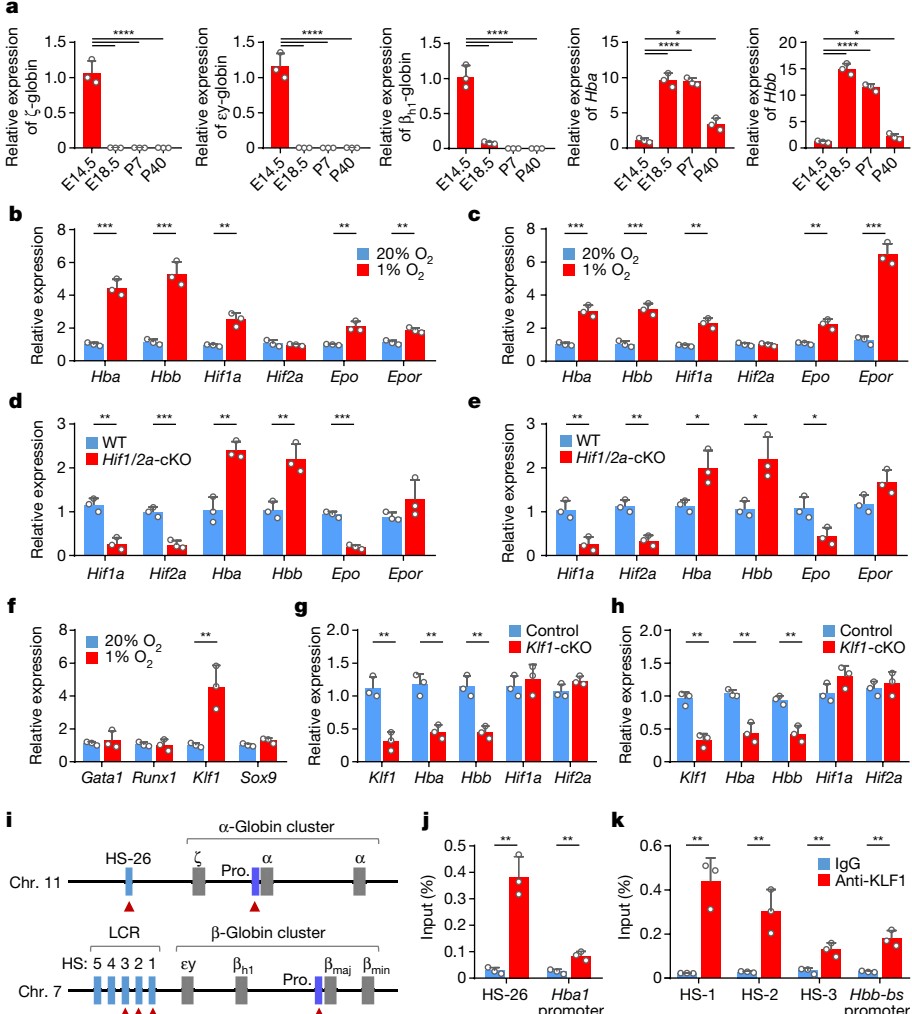

**Fig. 3 | Globin switching and expression regulation of haemoglobin in chondrocytes. a**, Expression of haemoglobin in cartilages from mice at different developmental stages by RT–qPCR. *$P = 0.0104$ (*Hba*) and 0.0179 (*Hbb*), and ****$P < 0.0001$. **b,c**, Expression level of the indicated genes in 4-day in vitro cultured cartilaginous tissues (**b**) and primary chondrocytes culture for 12 h (**c**). For panel **b**, ***$P = 0.0003$ (*Hba*), ***$P = 0.0007$ (*Hbb*), **$P = 0.0029$ (*Hif1a*), $P = 0.5285$ (*Hif2a*), **$P = 0.0042$ (*Epo*) and **$P = 0.0030$ (*Epor*). For panel **c**, ***$P = 0.0007$ (*Hba*), ***$P = 0.0008$ (*Hbb*), **$P = 0.0016$ (*Hif1a*), $P = 0.8062$ (*Hif2a*), **$P = 0.0031$ (*Epo*) and ***$P = 0.0002$ (*Epor*). **d,e**, Expression of the indicated genes in primary cultured chondrocytes upon conditional knockout (cKO) of both *Hif1a* and *Hif2a* (*Hif1/2a*) in either 20% (**d**) or 1% (**e**) $O_2$ for 6 h. For panel **d**, **$P = 0.0017$ (*Hif1a*), ***$P = 0.0009$ (*Hif2a*), **$P = 0.0027$ (*Hba*), **$P = 0.0067$ (*Hbb*), ***$P = 0.0001$ (*Epo*) and $P = 0.1996$ (*Epor*). For panel **e**, **$P = 0.0065$ (*Hif1a*), **$P = 0.0018$ (*Hif2a*), *$P = 0.0236$ (*Hba*), *$P = 0.0198$ (*Hbb*), *$P = 0.0270$ (*Epo*) and $P = 0.0671$ (*Epor*). **f**, Expression of the indicated genes in the primary

chondrocyte culture in 20% or 1% $O_2$ for 6 h. **$P = 0.0098$ (*Klf1*). **g,h**, Expression of the indicated genes in primary cultured chondrocytes upon cKO of *Klf1*⁻ in either 20% (**g**) or 1% (**h**) $O_2$ for 6 h. For panel **g**, **$P = 0.0030$ (*Klf1*), **$P = 0.0030$ (*Hba*), **$P = 0.0031$ (*Hbb*), $P = 0.5240$ (*Hif1a*) and $P = 0.0919$ (*Hif2a*). For panel **h**, **$P = 0.0014$ (*Klf1*), **$P = 0.0031$ (*Hba*), **$P = 0.0031$ (*Hbb*), $P = 0.1152$ (*Hif1a*) and $P = 0.5463$ (*Hif2a*). **i**, Schematic loci of mouse globin. β_maj, β_major; β_min, β_minor; Chr., chromosome; HS, DNase I hypersensitive site; LCR, locus control region; pro., promotor. The red triangles indicate KLF1-binding sites. **j,k**, ChIP–qPCR results for KLF1 binding to the LCR and promoters of α-globin (**j**) and β-globin (**k**) loci in chondrocytes. $P = 0.0014$ (HS26), 0.0052 (*Hba*1-pro), 0.0027 (HS-1), 0.0087 (HS-2), 0.0078 (HS-3), 0.0023 (*Hbb-bs*-Pro). $n = 3$ biologically independent experiments (**c–h,j,k**) or samples (**a,b**). All error bars represent s.e.m. $P$ values were calculated using one-way ANOVA test (**a**) or two-tailed Student's *t*-test (**b–h,j,k**); adjustment was not made for multiple comparisons.

terminus and C terminus of HBB, respectively. Truncation of either the C-terminal IDR or both IDRs, but not the N-terminal IDR alone, markedly suppressed the formation of cytoplasmic foci. An A139P mutation in the C-terminal IDR, a causal mutation linked to β-thalassaemia[13], also significantly compromised foci formation in cultured cells with essentially no effect on the protein expression level (Fig. 2i–k and Supplementary Fig. 2c). These mutations impaired, but did not completely prevent, condensation of HBB in vitro (Extended Data Fig. 3e,f). Thus, the C-terminal IDR of HBB was required for the effective formation of the cytoplasmic condensate. Together, these data fit well with the idea that phase separation of haemoglobin promoted Hedy formation.

## Globin switching in cartilage

The mouse α-globin gene locus contains an embryonic ζ-globin and two fetal/adult α-globin genes (*Hba1* and *Hba2*). The mouse β-globin locus has four genes in the order of 5'-εy-globin-β_h1 globin (embryonic)-*Hbb-bs* (β_maj)-*Hbb-bt* (β_min) (fetal/adult)-3' (Fig. 3i). During development, ζ-globin, εy-globin and β_h1-globin are embryonically expressed in primitive erythrocytes (embryonic day 7.5 (E7.5)–E14.5), and their expression is silenced in definitive erythrocytes that express α-globins and β-globins at the fetal and adult stages of development[14,15], a process called globin switching[14,16,17]. To examine whether globin switching occurred in cartilage, quantitative PCR with reverse

transcription (RT–qPCR) was performed on cartilage of different developmental stages. The results showed that the embryonic ζ-globin, εγ-globin and $\beta_{h1}$-globin were highly expressed in the chondrocytes of early embryonic (E14.5) fetal growth plates (Fig. 3a), but sharply decreased to an undetectable level in the chondrocytes of late fetal and adult growth plates (E18.5 and P7) (Fig. 3a). By contrast, the fetal/adult α-globin and β-globin were expressed minimally in E14.5 fetal growth plates, but highly in the growth plates of E18.5 and P7 mice (Fig. 3a). Therefore, globin switching also occurred in the developing cartilage in a way resembling that during the erythroid development.

## Regulation of haemoglobin in chondrocyte

Hypoxia is an established inducer of haemoglobin expression. To test whether the mechanism also works in chondrocytes, we examined the transcription of *Hba* and *Hbb* in cultured cartilaginous tissue under hypoxic environments. As expected, the mRNA levels of *Hba* and *Hbb* were upregulated upon hypoxia (Fig. 3b), which was confirmed in primary chondrocytes (Fig. 3c). Meanwhile, transcription of *Hif1a*, *Epo* and *Epor*, but not *Hif2a*, was induced as well (Fig. 3b,c). To test whether the haemoglobin expression was regulated by HIF signalling, a major pathway activated by hypoxia, we examined the mRNA levels of *Hba* and *Hbb* in *Hif1a*-deleted and *Hif2a*-deleted chondrocytes. Consistent with previous reports[18–20], homozygous deletion of *Hif1a* in chondrocytes, but not *Hif2a*, resulted in massive cell death in the centre of the cartilaginous growth plate (Extended Data Fig. 5a–g). Knockout of *Hif1a* and *Hif2a*, either alone or in combination, reduced *Epo* expression in primary cultured chondrocytes and fetal growth plates under both normoxic and hypoxic conditions (Fig. 3d,e and Extended Data Fig. 5h–m). However, the expressions of *Hba* and *Hbb* were unexpectedly induced upon knockout of *Hif1a* and/or *Hif2a* (Fig. 3d,e and Extended Data Fig. 5h–m), which was further confirmed by the results from chemical inhibition of HIF1α by GN44028 and HIF2α by PT2385 or PT2399, and from chemical activation of HIF1α and HIF2α by IOX2, roxadustat and DMOG (Extended Data Fig. 6a–l). These results suggest that hypoxia was unlikely to promote haemoglobin expression in chondrocytes via HIF1/2α.

## KLF1 mediates haemoglobin expression

We next examined the expression of KLF1, RUNX1 and GATA1, the transcription factors critical for erythropoiesis[21–23], under hypoxic stress. Hypoxia significantly promoted the transcription of *Klf1*, but not of *Runx1* or *Gata1*, in the primary chondrocytes (Fig. 3f). KLF1 has been previously reported to be essential for globin switching[24,25], and its deletion resulted in β-thalassaemia in mice[26,27]. We therefore examined its regulation on haemoglobin expression. Conditional knockout or short interfering RNA-mediated knockdown of *Klf1* significantly reduced the expression of *Hba* and *Hbb* in primary chondrocytes, as well as ATDC5 chondrocyte cells (Fig. 3g,h, Extended Data Fig. 7a–f and Supplementary Figs. 3 and 4). In agreement with the results, chromatin immunoprecipitation (ChIP)–qPCR results showed that KLF1 directly bound the locus control region enhancer and promoter regions of both the α-globin and the β-globin gene loci in chondrocytes (Fig. 3i–k). Therefore, these results support that KLF1 could mediate hypoxia-induced *Hba* and *Hbb* expression in chondrocytes.

Consistent with a nonessential role of the HIF signalling pathway in chondrocyte haemoglobin regulation by hypoxia, conditional knockout of *Hif1a* and *Hif2a*, either alone or in combination, did not compromise *Klf1* expression. Instead, it resulted in a significant upregulation of *Klf1* under both normoxic and hypoxic conditions (Extended Data Fig. 8a–f). A recent study has reported a novel HIF-independent gene regulation, in which hypoxia-mediated inactivation of KDM5A, an oxygen-dependent dioxygenase that epigenetically regulates gene expression via its JmjC-histone demethylase[28], was able to mediate

the upregulation of *Klf10* by increasing the genomic H3K4me3 level[29]. Coincidently, bioinformatics analysis identified an H3K4me3 modification region within genes of *Klf1*, but not within *Hba* and *Hbb* (Supplementary Fig. 5), suggesting *Klf1* as a potential target of hypoxia via KDM5A. In agreement with this notion, knockdown of *Kdm5a*, but not its close family member *Kdm5b*, significantly increased *Klf1* expression even under normoxic conditions (Extended Data Fig. 8g–j), which was associated with increased H3K4me3 modifications of the *Klf1* locus, but not that of the *Bap1* negative control, as shown by ChIP–qPCR analysis (Extended Data Fig. 8k,l).

## Chondrocyte survival requires Hedy

We then investigated the role of haemoglobin in cartilage development by using gene knockout mice. E14.5 mouse embryos with a homozygous deletion of *Hba* or *Hbb* were noticeably smaller than their heterozygous or wild-type (WT) littermates. Skeleton and histological assays indicated a mild delay of cartilage hypertrophy with few pattern defects in the homozygous mice compared with their heterozygous or WT littermates (Supplementary Fig. 6a–h). However, all embryos became hydropic and died at approximately E16.5–E18.5, whereas the heterozygous mice were fertile with few skeletal defects at E14.5 or E18.5 (Supplementary Fig. 7a–e). Furthermore, at P5, massive cell death occurred in the inner zones of developing growth plates as detected by histological examination and TUNEL assays (Fig. 4a and Supplementary Fig. 7f), which resembled the phenotype of *Hif1a* deletion in cartilage[9,10] (Extended Data Fig. 5d).

To confirm our findings from the complete knockout mouse model, mice with *Hbb*-floxed loci were crossed with *Prx1-Cre* mice, where the expression of *Cre* was driven by the *Prx1* promoter, which allows conditional gene deletion in mesenchymal cells and chondrocytes (Supplementary Figs. 8 and 9a–c). The neonatal mice with homozygous *Hbb* deletion died within 1–7 days after birth, whereas the heterozygous *Hbb* mice survived to adulthood and were fertile (Supplementary Fig. 9d,e); however, few defects were detected in the growth plates of $Hbb^{flox/flox}$ ($Hbb^{F/F}$)/*Prx1-Cre* mice at E16.5. Increased death was detected in the centre of the $Hbb^{F/F}$/*Prx1-Cre* cartilages, but not those of the WT P1 and P5 mice (Supplementary Fig. 9f–l). To exclude the effects of anaemia-related hypoxia on chondrocyte survival, the $Col2a1-Cre^{ERT2}$ mice were crossed with $Hbb^{F/F}$ mice to produce chondrocyte-specific tamoxifen-induced *Hbb* knockout (*Hbb*-cKO) mice that exhibited no anaemia or hypoxia in other tissues, such as the liver and muscle (Extended Data Fig. 9a–c and Supplementary Fig. 10), and died within 7 days after birth (Extended Data Fig. 9d). Still, chondrocyte death occurred in the cartilages of P1 and P5 *Hbb*-cKO mice, as did the $Hbb^{F/F}$/*Prx1-Cre* mice (Fig. 4b and Supplementary Fig. 9f–l), which was associated with loss of expression of *Hbb* in a temporally induced cKO assay (Extended Data Fig. 10a–c). Together, these results demonstrate an essential role of haemoglobin in the survival of chondrocytes within cartilage.

## Hedy is required for hypoxia adaption

As haemoglobin is an oxygen vector, we hypothesized that haemoglobin knockout in cartilage may result in an oxygen shortage contributing to the death of chondrocytes (Fig. 4a,b, Extended Data Figs. 9 and 10 and Supplementary Figs. 7f and 9f). To test this idea, the nitroimidazole EF5 was injected into pregnant female mice at E19.5, followed by immunostaining with anti-EF5 antibody to label cells under hypoxic stress in situ as previously described[9,10]. As shown in Extended Data Fig. 11a–e, *Hbb*-cKO resulted in Hedy loss in the chondrocytes of cartilage growth plates as indicated by tissue histology and transmission electron microscopy analysis, which took place along with increased EF5 signals (Fig. 4c and Extended Data Fig. 11f, top panel), indicating an increase in hypoxic levels in the cartilages upon *Hbb* knockout.

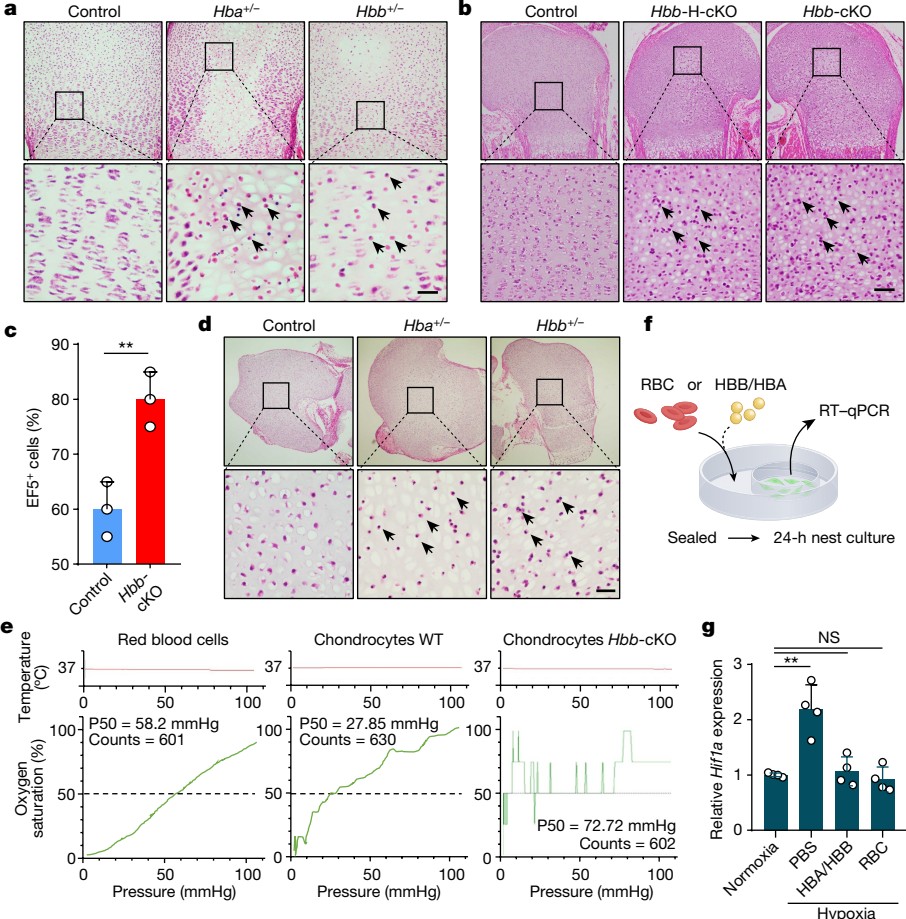

**Fig. 4 | Haemoglobin is essential for chondrocyte hypoxia adaption and survival of the fetal cartilage. a**, Histological examination of proximal humeral cartilages from P5 mice of different genotypes. The arrows indicate dead chondrocytes. Scale bar, 50 μm. *n* = 6 biologically independent samples. **b**, Histological images of proximal humerus from newborn mice upon cKO of the *Hbb* gene. Control: *Hbb*^+/+^/*Col2a1-Cre*^ERT2^ mice, *Hbb*-H-cKO: *Hbb*^F/+^/*Col2a1-Cre*^ERT2^ mice (heterozygous deletion), and *Hbb*-cKO: *Hbb*^F/F^/*Col2a1-Cre*^ERT2^ mice. The arrows indicate dead chondrocytes. Scale bar, 50 μm. *n* = 6 biologically independent samples. **c**, Quantification of EF5-positive cells in cartilages from WT and *Hbb*-cKO mice. *n* = 3 biologically independent samples for each. Error bars represent s.e.m. **P* = 0.0080. **d**, Histological examination of cartilages of E18.5 mice cultured in vitro under hypoxia (1% O_2) for 3 days. Black arrows indicate the dead chondrocytes. Scale bar, 50 μm. **e**, Oxygen dissociation curves of RBCs, WT and *Hbb*-cKO chondrocytes. The chondrocytes of *Hbb*-cKO displayed poor oxygen-binding capability as indicated by the fluctuated curve. Counts indicate the measurement time (in seconds), the horizontal grey lines indicate oxygen partial pressures in the environment of chondrocytes. **f**, Schematic diagram of the nest co-culture experiment, in which the hypoxia-responsive cells were cultured in the inner dish, whereas the RBCs or the haemoglobin condensates were placed in the outer dish that was sealed for 24 h to create hypoxic conditions. **g**, Expression of *Hif1a* under the indicated conditions of nest co-culture as examined by qPCR. The data are mean ± s.e.m. of triplicate experiments; two-sided Student's *t*-test was used for the data analysis, and adjustment was not made for multiple comparisons. **P* = 0.0019 (phosphate-buffered saline), *P* = 0.6454 (HBA/HBB) and *P* = 0.4890 (RBC). NS, not significant.

Consistently, the *Hbb*-cKO cartilages expressed higher levels of HIF1a (Extended Data Fig. 11f, bottom panel). The hypoxic effect was further enhanced by an increased loading of pressure, a common stimulus produced during physical exercise[30], suggesting important physiological implications for chondrocyte haemoglobin (Extended Data Fig. 11g–j). Moreover, the metabolomic profile indicated that the intracellular lactate significantly increased (Extended Data Fig. 11k), together with a significant decrease of the intracellular glucose (Extended Data Fig. 11l), leading to a higher lactate-to-glucose ratio (Extended Data Fig. 11m), in the *Hbb*-cKO cartilages over the control. Collectively, these data suggest that enhanced intracellular hypoxia promoted glycolysis and death of chondrocytes in *Hbb*-cKO fetal growth plates. However, it was unlikely that an impaired energy supply or enhanced production of reactive oxygen species drove the chondrocyte death, which seemed to be caspase 3 independent (Supplementary Figs. 11a–g and 12), because *Hbb* deletion neither decreased intracellular ATP levels, the ATP-to-ADP ratio or the pAMPK-to-AMPK ratio (Supplementary Fig. 11a,b), nor increased total or mitochondrial reactive oxygen species (Supplementary Fig. 11c,d).

Furthermore, the chondrocyte death was not attributed to activation of HIF signalling by persistent hypoxia as treatment with HIF1/2α activators, such as IOX2, roxadustat and DMOG, rescued cartilages from death induced by haemoglobin deficiency (Supplementary Fig. 13a–d) rather than promoting their death.

To test the essential role of haemoglobin in hypoxia tolerance of chondrocytes, a hypoxia tolerance experiment was performed by culturing E18.5 humeral cartilage growth plates in hypoxic environment (1% O_2) for 3 days. The results showed that even partial deletion of either *Hba* or *Hbb* sensitized chondrocytes to death induced by hypoxia in the cartilage growth plates (Fig. 4d and Extended Data Fig. 12a), which was further confirmed in E14.5 cartilage cultured for 6 days (Extended Data Fig. 12b,c). Consistently, the WT chondrocytes with intact haemoglobin expression tended to release oxygen under a more hypoxic condition than RBCs, as indicated by a much lower P50 (the partial pressure of oxygen at which haemoglobin is 50% saturated with oxygen) (27.85 mmHg versus 58.2 mmHg) (Fig. 4e); conversely, the *Hbb*-deleted chondrocytes exhibited marginal capacity to bind to and supply oxygen

(Fig. 4e, right panel). To confirm that Hedy might function as a source of oxygen during hypoxia, Hedies were isolated by hyposmotic rupture of 239T cells co-transfected with HBA–mCherry and HBB–GFP (Supplementary Fig. 14a). A 24-h nested co-culture of the isolated Hedies with PC12, a hypoxia-sensitive cell, effectively reversed the upregulated HIF1α expression to a level comparable with that of an RBC co-culture and a nomoxia control (Fig. 4f,g). Moreover, in a co-culture experiment, the ATDC cells expressing haemoglobin were more tolerant to hypoxia than the adjacent haemoglobin-negative cells as determined by the nuclear localization of HIF1α (Supplementary Fig. 14b,c). Thus, these data are consistent with the idea that the intracellular haemoglobin (Hedy) serves as a local oxygen storage that supplies oxygen to sustain chondrocyte survival over regional hypoxia in cartilage.

## Conclusions

Timely supply of oxygen is a prerequisite for cells within tissues. Although RBCs transport oxygen from the respiratory system to different vascular tissues over a long range, oxygen supply to individual cells within tissues can only be achieved by diffusion, which is rather low in efficiency for satisfying the oxygen needs of tissues that are either high-oxygen consuming or avascular. To cope with this, cells within these tissues develop additional mechanisms to obtain sufficient oxygen over a short range. For example, muscle cells express a large amount of myoglobin that can bind to and store oxygen, which enables timely and prolonged supply of oxygen during movement[31–33]. Neurons, another type of high-oxygen-demanding cells, express neuroglobin to store oxygen[34,35]. Nevertheless, the corresponding mechanisms for cells of avascular tissues with limited oxygen availability, such as chondrocytes in cartilages, to supply oxygen are poorly understood.

In this study, we report here that chondrocytes use a similar strategy to adapt to the hypoxic environment of cartilage growth plates. Conversely, instead of producing a tissue-specific globin, the chondrocytes express a unique composition of haemoglobin to form membrane-less Hedy within their cytoplasm. P50 is markedly left-shifted for the Hedy-containing chondrocytes (27.58 mmHg) compared with the RBCs (58.2 mmHg) from the same mice. The left-shifted P50 enables chondrocytes to bind to and store $O_2$ diffused from the hypoxic environment for short-range supply. This is critical for chondrocyte survival in the developing growth plates as depletion of haemoglobin, and consequently Hedy loss, resulted in massive death of chondrocytes and retarded skeleton development. To our knowledge, this is the first study demonstrating an extra-erythrocyte role of haemoglobin in chondrocytes based on a mouse model.

Our finding updates the usually well-accepted opinion that haemoglobin is erythrocyte specific. In fact, ectopic expression of haemoglobin in cells other than RBCs had been reported sporadically over the past decades. The target cells include the alveolar epithelial cells (ATII and Clara)[36], macrophages[37], mesangial cells[38], mesencephalic dopaminergic neurons and glial cells[39], retinal pigment epithelium[40], tumour cells[41] and the like. Nevertheless, despite multiple lines of documentation, the functional implications of the ectopic expression were largely speculative with little in vivo evidence. Our study demonstrated that, in a mouse model, haemoglobin expression in chondrocytes was induced in responding to hypoxia, and haemoglobin depletion resulted in increased hypoxia and glycolysis, and activated the HIF signalling pathway. Thus, these results are consistent with a role of haemoglobin in local storage and the timely supply of oxygen. In agreement with these findings, it has been reported that patients with thalassaemia syndromes usually have joint pain[42], and anaemia was common in individuals with cartilage-relative diseases, such as rheumatoid arthritis (about 30–70%)[43,44] and cartilage–hair hypoplasia (about 73%)[45]. Furthermore, patients with rheumatoid arthritis with lower haemoglobin levels exhibited more severe joint disease[46], and treated anaemia substantially relieved joint disease[47].

Another interesting finding of this study is that haemoglobin forms condensates that manifested with features of phase separation. This is unlikely to be an artefact of experimental context as these condensates were readily detected in chondrocytes of different sources (Fig. 1), and truncation analysis indicated that the formation of the condensates was actually a controlled process (Fig. 2i–k). Moreover, haemoglobin was found to be granularly distributed within retinal epithelium[40] and glaucoma cells[48], in line with condensate morphology. These results fit well with the idea that haemoglobin forms condensates in a defined context, although its regulation warrants further study. A plausible explanation for this phenomenon is that the condensed haemoglobin may help to store more oxygen within limited space to sustain relatively longer oxygen demand to cells from tissues that are either high-oxygen consuming or avascular.

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

## Methods

### Mouse strains and genotyping

Mice heterozygous for the α-globin null allele ($Hba^{+/-}$, both of the adult haemoglobin genes, α1 and α2, and the region between them were deleted and replaced with a neomycin resistance cassette by homologous recombination) or for the β-globin null allele ($Hbb^{+/-}$, a genomic fragment encompassing all of $Hbb$-$b1$ and a 5′ portion of $Hbb$-$b2$ was replaced with a neomycin cassette inserted by homologous recombination) were produced by crossing mice of $Hba^{tm1Paz}$ $Hbb^{tm1Tow}$ Tg (HBA−HBBs) 41Paz/J (no.: 003342, Jackson Labs)[49] with WT C57BL/6J mice. The $Hba^{tm1Paz}$ $Hbb^{tm1Tow}$ Tg (HBA−HBBs) 41Paz/J mice are called sickle cell mice (Berkeley model), which are homozygous for the both α-globin and β-globin null allele and carrying the human sickle transgene ($Hba^{0/0}$ $Hbb^{0/0}$Tg (Hu-miniLCRα$1^G$γ$^A$γδβ$^S$)). Mice homozygous for $Hba$-knockout or $Hbb$-knockout mutation die in utero from severe anaemia. $Hif1a^{F/F}$ (no. 007561)[50], $Col2a1$-$Cre^{ERT2}$ (no. 006774)[51], $Prx1$-$Cre$ (no. 005584)[52] mice were from Jackson Labs. $Hif2a^{F/F}$ (no. NM-CKO-200163) mice were from Shanghai Model Organisms Center. To specifically knockout the $Hif1a$ and/or $Hif2a$ genes in chondrocytes, $Hif1a^{F/F}$ and/or $Hif2a^{F/F}$ mice were bred to $Col2a1$-$Cre^{ERT2}$ mice. After tamoxifen treatment, the $Hif1a^{F/F}$/$Col2a1$-$Cre^{ERT2}$, $Hif2a^{F/F}$/$Col2a1$-$Cre^{ERT2}$ or $Hif1a^{F/F}$/$Hif2a^{F/F}$/$Col2a1$-$Cre^{ERT2}$ mice will delete the second exons of $Hif1a$ and/or $Hif2a^{F/F}$ genes in the $Cre$-expressing chondrocytes. Mice of both genders were used for this study. Genotyping was performed as described in the web of Jackson Labs.

To investigate the role of haemoglobin in vivo, we generated conditional $Hbb$-floxed mice with the C57BL/6J background by homologous recombination. A targeting vector was designed to replace both the 1–3 exons of $Hbb$-$bs$ and that of $Hbb$-$bt$ ($Hbb^{F/+}$ or $Hbb^{F/F}$; Supplementary Fig. 8a). A candidate of the embryonic stem cells was screened and further confirmed by PCR. F1 mice were verified by PCR. The following primers were used for the genotyping of $Hbb^{F/F}$ mice: P1, 5′-TGCATCTGCAGATCCCAAAAA-3′ and P2, 5′-GGAGGAGTGTACAAGGAGTTCAATAA-3′. With the two primers, it is possible to amplify specific WT (574 bp) and mutant (766 bp) DNA fragments (Supplementary Fig. 8b,c). To conditionally delete $Hbb$ in mesenchymal cells or chondrocytes, $Hbb^{F/F}$ mice were bred to $Prx1$-$Cre$ or $Col2a1$-$Cre^{ERT2}$ mice.

To investigate the role of KLF1 in chondrocytes, we generated conditional $Klf1$-floxed mice with the C57BL/6J background by homologous recombination. A targeting vector was designed to replace the second exon of $Klf1$ (Supplementary Fig. 3a). A candidate of embryonic stem cells was screened and further confirmed by PCR. F1 mice were verified by PCR. The following primers were used for the genotyping of $Klf1^{F/F}$ mice: P1, 5′-AGGGGTCTGAGATCAAGGTGA-3′ and P2, 5′-CGGTTCCCCTAACCCCTTTC-3′. With the two primers, it is possible to amplify specific WT (317 bp) and mutant (383 bp) DNA fragments (Supplementary Fig. 3b,c). To conditionally delete $Klf1$ in chondrocytes, $Klf1^{F/F}$ mice were bred to $Col2a1$-$Cre^{ERT2}$ mice.

All the clones were maintained in pathogen-free conditions at the Fourth Military Medical University. Animal studies were approved by the Institutional Animal Care and Use Committee at the Fourth Military Medical University.

### Analysis of recombination in newborn mice in vivo

Cartilage from growth plates of newborn or embryonic mice collected and digested with collagenase type II. Next, genomic DNA was extracted from those digested growth plates. Efficient recombination of the $Hba$, $Hbb$, $Hif1a$, $Hif2a$ or $Klf1$ floxed allele was quantified by qPCR of genomic DNA with corresponding primers.

### Histological analysis

Mice were euthanized at the indicated ages, and cartilaginous tissues were collected and fixed in 4% paraformaldehyde overnight at room temperature. Postnatal skeletal tissues were decalcified in 0.5 M EDTA for 10 days and then embedded in paraffin. Serial sections were taken at 4-µm thickness and stained with haematoxylin and eosin (H&E) according to standard protocols. The Masson trichrome staining was conducted with a kit (BA-4079A, Baso) according to the instructions of the manufacturer. All images were visualized with a microscope (BX51, Olympus), and images were captured with the digital camera (DP71, Olympus) using the accompanying software.

### Immunohistochemical and immunofluorescence staining

Immunohistochemistry was performed using horseradish peroxidase (HRP)-coupled goat anti-rabbit secondary antibody (1:500; ab7090, Abcam) with diaminobenzidine serving as the substrate. Expression of HBA or HBB was detected by immunohistochemistry with antibodies to HBA (MA5-32328, Invitrogen) or HBB (PA5-60287, Invitrogen). Paraffin sections were dewaxed, rehydrated and washed in 0.1 M phosphate-buffered saline (PBS). Endogenous peroxidase activity was blocked by immersing the sections in 0.3% $H_2O_2$ for 30 min. Nonspecific binding was blocked by incubation of the sections in blocking buffer (5% BSA, 20% normal goat serum and 0.1% Triton X-100 in TBS) for 30 min. Subsequently, sections were incubated with primary antibody to HBA or HBB with dilution of 1:50 in blocking buffer in a humidified chamber overnight at 4 °C. They were then washed in PBS and incubated in a mixture of secondary anti-rabbit antibodies conjugated with HRP at the 1:100 dilution. Diaminobenzidine substrate was used for detection and haematoxylin was used for counterstaining. The samples were then dehydrated and mounted for visualization. The cells with brown nuclei were considered positively stained. Immunofluorescences with anti-HIF1α (rabbit polyclonal; PA5-60287, Invitrogen), HBA (MA5-32328, Invitrogen) or HBB (PA5-60287, Invitrogen) antibodies were performed on paraffin-embedded sections according to standard protocols.

### Scanning electron microscopy

The proximal end of the cartilaginous growth plate of the humerus or the distal end of the cartilaginous growth plate of the femur from embryonic or neonatal mice killed by pentobarbitone overdose, ranging in age from E14.5 to P7, were dissected and the articular capsules were removed under a stereoscope. Human knee articular cartilages were from patients with acute trauma. The cartilaginous growth plates or articular cartilage were washed with 0.1 M PBS for three times and fixed for 24 h with 4% paraformaldehyde in 2.5% glutaraldehyde. Samples to be imaged by electron microscopy were dehydrated in an ascending 70%, 80%, 90% and 100% (for 15 min each change) ethanol series. Once dried, the samples were cooled with liquid nitrogen, fractured and were placed in a sealed dish for 4 days for further dehydration under room temperature (24–26 °C). The samples were coated with platinum using a high-resolution sputter coater (Shinkku VD MSP 1S). Samples for electron microscopy were imaged using a scanning electron microscope (S-4800, Hitachi).

To exclude the possibility of cross-contamination by peripheral blood in cartilage tissues, the paraffin-embedded cartilaginous growth plate of P6 mice were cut into sections (4 µm). The sections were dewaxed in xylene (30 min) and subsequently dexylened in 100% (for 10 min) ethanol. The sections were dried in a vacuum freeze-dryer for 12 h. Once dried, the region (1 cm$^2$) that contained cartilaginous tissues on the slide were cut by glass cutter and observed by a scanning electron microscope (S-4800, Hitachi).

### Transmission electron microscopy

To observe the detail of Hedies within chondrocytes, transmission electron microscopy (TEM) of the epiphyseal growth plate was performed following the standard procedures. The epiphyseal growth plate of E14.5, E18.5 or P3 mice were surgically collected quickly by removing the joint capsule, ligaments and metaphyseal bone with

scissors and scalpel. The cartilaginous tissues were fixed in 4% paraformaldehyde in 2.5% glutaraldehyde (pH 7.3) for 24 h. The tissue segments were dehydrated in increasing concentrations of ethanol from 70% to 100% and infiltrated and embedded in SPI-PON812 resin (SPI-CHEM). They were then sectioned at 5 μm thickness and stained with 1% toluidine blue for light microscopic assessment. Tissue samples from the selected regions were cut into sections on an ultramicrotome (EM UC6, Leica), and prepared for the study with the JEM-1230 electron microscope. Blocks chosen for ultrastructural assessment were trimmed, sectioned at 60 nm, stained with lead citrate and uranyl acetate, and examined with a transmission electron microscope (JEOL). Electron micrographs were captured by a Gatan digital camera (832 SC1000, Gatan) and its application software (Gatan Digital Micrograph 3.0 software).

To observe the detail of Hedies within HepG2 and PLC/PR/F5 cells transfected with HBB–eGFP, TEM was performed following the standard procedures. About $1 \times 10^7$ cells were plated in a six-well plate precoated with type I collagen. Cells were pelleted 24 h after transfection, and fixed in 2.5% (v/v) glutaraldehyde with phosphate buffer (0.1 M, pH 7.4) for 12 h at 4 °C followed by fixing in 1% (w/v) osmium tetraoxide in phosphate buffer for 2 h at 4 °C. After that, cells were dehydrated through a graded ethanol series (30%, 50%, 70%, 80%, 90%, 100% and 100%, 5 min each at 4 °C) till pure acetone (2 × 5 min). Samples were infiltrated in graded mixtures (3:1, 1:1 and 1:3) of acetone and SPI-PON812 resin (16.2 ml SPI-PON812, 10 ml DDSA and 8.9 ml NMA), then changed to pure resin. Finally, cells were embedded in pure resin with 1.5% BDMA and polymerized for 12 h at 45 °C, and then 48 h at 60 °C. The ultrathin sections (70 nm) were sectioned with microtome (EMUC7, Leica), double-stained with uranyl acetate and lead citrate, and examined by a transmission electron microscope (FEI Tecnai Spirit 120 kV).

## Immunoelectron microscopy

To determine whether the Hedies within chondrocytes contain HBA and HBB, immunoelectron microscopy was performed to detect HBA and HBB by the immunogold silver-staining method. Epiphyseal growth plates of P3 mice were cut into small pieces with a diameter of about 0.5 mm and fixed in 4% paraformaldehyde and 2.5% glutaraldehyde (pH 7.3) for 24 h. Small pieces of cartilaginous tissues were washed thoroughly with 0.1 M PBS. After incubation with blocking buffer (5% BSA, 20% normal goat serum and 0.1% Triton X-100 in TBS) for 1 h to block nonspecific binding, the small pieces of cartilaginous tissues were incubated with the primary antibodies HBA (1:100; MA5-32328, Invitrogen) or HBB (1:100; PA5-60287, Invitrogen) diluted at 1:50 in blocking buffer in a humidified chamber overnight at 4 °C. They were then washed in PBS and incubated overnight in a mixture of secondary antibodies, anti-rabbit IgG conjugated to 1.4-nm gold globes (Nanoprobes) at 1:100 dilution and biotinylated anti-guinea pig IgG at 1:200 dilution. After rinsing, sections were post-fixed in 2% glutaraldehyde in PBS for 45 min. Silver enhancement was performed in the dark with HQ Silver Kit (Nanoprobes) for visualization of HBA or HBB immunoreactivity. Before and after the silver enhancement step, sections were rinsed several times with deionized water. They were then incubated in the ABC solution (Sigma) for 4 h and visualized by the glucose oxidase-3, 3′-diaminobenzidine method. Immunolabelled cartilaginous tissues were fixed with 0.5% osmium tetroxide in 0.1 M phosphate buffer for 1 h, dehydrated in graded ethanol series, then in propylene oxide, and finally flat-embedded in SPI-PON812 (SPI-CHEM). After polymerization, cartilaginous tissues were trimmed under a stereomicroscope and mounted onto blank resin stubs. Ultrathin sections were cut with an ultramicrotome (EM UC6, Leica) and mounted on mesh grids (6–8 sections per grid). They were then counterstained with uranyl acetate and lead citrate, and observed under an electron microscope (JEM-1230, JEOL). Electron micrographs were captured by a Gatan digital camera (832 SC1000, Gatan) and its application software (Gatan Digital Micrograph 3.0 software).

## Culture of cartilage tissues in vitro

To examine the role of HBA or HBB in chondrocytes in vitro, culture of cartilage tissues was performed. The whole humerus or femur of E14.5 or E18.5 embryonic littermates with genotypes of WT, *Hba*$^{+/-}$, *Hbb*$^{+/-}$, *Hba*$^{-/-}$ or *Hbb*$^{-/-}$ were isolated. The mice were numbered and grouped based on genotypes. For each mouse, the humerus or femurs were used for culture in vitro. The humeri or femurs were digested with 0.25% trypsin in six-well plates at 37 °C for 15 min. The samples were repeatedly pipetted until the muscle and tendon on the cartilaginous growth plates were removed. Then, cartilaginous growth plates were collected gently by sharp tweezers and cultured with α-MEM medium supplemented with 10% FCS, 100 units per ml penicillin, 50 μg ml$^{-1}$ streptomycin (Gibco) in 12-well plates. For the hypoxia experiment, cartilaginous tissues were grown at 1% O$_2$ in α-MEM supplemented with 10% FCS for 3–6 days. After the hypoxia experiment, these cultured cartilages were collected and prepared for histology and TUNEL examination.

## Primary culture of chondrocytes under hypoxia

For primary culture of chondrocytes, the growth plates of P3 mice were sheared by scissors into small pierce and digested with 0.1% collagenase type II (Gibco) dissolved in α-MEM medium at 37 °C for 12 h. Then, the samples were repeatedly pipetted into single cells. The cells were filtered through a 75-μm nylon mesh and cultured in six-well plates with α-MEM medium supplemented with 10% FCS. Primary chondrocytes were seeded at a density of $6 \times 10^5$ cells per cm$^2$. The medium was changed every other day. For the hypoxia experiment, primary chondrocytes were grown at 1% O$_2$ in α-MEM supplemented with 10% FCS for 48–72 h.

## TUNEL assay

The TUNEL assay was conducted as previously described[53] by using an In Situ Apoptosis Detection kit (11684795910, Roche) according to the instructions of the manufacturer.

## EF5 staining

To examine the state of chondrocyte oxygenation in the mice growth plate, EF5, a hypoxia-sensing drug, was used. EF5 is a pentafluorinated derivative of the 2-nitroimidazole, etanidazole, that is metabolically reduced by oxygen-inhibitable nitroreductase. EF5 staining was performed on fixed-frozen sections from growth plates of mice. E19.5 *Hbb* heterozygous pregnant females were injected with 10 mM EF5 (Merck) at 1% (v/w) of body weight. Two to three hours later, the mice were dissected out in cold PBS. Epiphyseal growth plates were surgically isolated quickly by cutting the joint capsule, ligaments and metaphyseal bone with scissors and fixed in cold acetone for 10 min at 4 °C, and air dried and rinsed in PBS. Blocking was performed with 5% mouse serum in PBS for 30 min at room temperature. Sections were stained with a mouse anti-EF5 Cy3-conjugated antibody (EF5-30C3, Merck), diluted at 1:20 in 3% BSA and PBS, for 1 h at 37 °C. Slides were rinsed with PBS for 5 min and mounted with an aqueous mounting medium. EF5 and DNA fluorescence was recorded using a fluorescence microscope with filters appropriate for DAPI and Cy3 and a digital camera.

## Western blot analysis

The cartilaginous growth plates from P1 to P7 mice were collected by quickly removing the joint capsule, ligament and metaphyseal bone with scissors. The growth plates were washed three times in 0.1 M cold PBS. Then, the cartilaginous tissues were sheared, grinded and lysed into radioimmunoprecipitation assay (RIPA) buffer containing protease inhibitor cocktails (Roche). Cartilaginous tissues or chondrocytes lysates (50 μg) were separated by 12% SDS–PAGE gel and transferred onto a polyvinylidene fluoride (PVDF) membrane (Millipore). PVDF membranes were blocked with 5% (w/v) skimmed milk in Tris-buffered

saline with 0.1% (v/v) Tween-20 for 1 h at room temperature. Then, the membranes were probed with antibodies to HBA (1:1,000; MA5-32328, Invitrogen) or HBB (1:1,000; PA5-60287, Invitrogen) and α-tubulin (1:3,000; 2125, Cell Signaling Technology) antibodies overnight at 4 °C. Secondary detection was performed using anti-rabbit (1:10,000; A0545, Millipore) or anti-mouse (1:10,000; M4155, Millipore) antibodies at room temperature for 1 h. After being extensively washed with PBS, the protein signals of interest were detected by enhanced chemiluminescence and exposure to X-ray film.

Proteins from primary chondrocytes or ATDC5 chondrocytes were extracted in RIPA buffer. Each sample of 50 μg protein was electrophoresed in a pre-cast 4–20% Tris gel. After gel transfer to PVDF membranes using a Bio-Rad Criterion system, blots were blocked in 5% non-fat milk/1× TBST for 1 h at room temperature and incubated overnight at 4 °C with the following primary antibodies: HIF1α (20960-1-AP, Proteintech) at 1:1,000, HIF2α (also known as Epas1) (NB100-122, Novus) at 1:1,000, KLF1 (also known as EKLF) (PAB5859, Abnova) at 1:1,000, AMPK (A1229, ABclonal) at 1:1000, pAMPK (AP1002, ABclonal) at 1:1,000, caspase 3 (A19654, ABclonal) at 1:1,000, KDM5A (A4755, ABclonal) at 1:1,000, and KDM5B (A15740, ABclonal) at 1:1,000. The membranes were then incubated with an HRP-conjugated anti-rabbit (1:10,000; Millipore) or anti-mouse (Millipore) antibodies at room temperature for 1 h in 1× TBST. Signal was detected by using enhanced chemiluminescence. Protein molecular weight was determined using the protein marker (PM2610, SMOBIO). Western blot images were acquired and analysed via the Bio-Rad Image Lab system. Quantification was performed using ImageJ. The α-tubulin signal was used to normalize for protein amount.

### Laser capture microdissection

Frozen humerus longitudinal cartilages of P3 mice were cut into sections (10 μm) with a cryostat (Leica Biosystems). The sections were placed on PEN membrane frame slides (Leica Microsystems) and were fixed in 95% ethanol for 1 min. The slides were immersed in distilled water for 1 min three times to get rid of the OTC reagent. Then, the sections were stained by H&E staining as previously described with the following modifications: Harris haematoxylin (BA-4041, Baso), 30 s; deionized water, 3 min; eosin (BA-4042, Baso), 5 s; 95% ethanol, 15 s; and 100% ethanol, 15 s. The sections were dried in a vacuum freeze-dryer for 24 h under room temperature (24–26 °C). Once dried, the chondrocytes were microdissected one by one with LMD6000 (Leica Microsystems) according to the manufacturer's instructions. The samples dissected from about 200 chondrocytes were pooled into the cap of a 0.5-ml microcentrifuge tube and dissolved in tissue extraction buffer (PicoPure RNA Isolation Kit, Applied Biosystems) for mass spectrometry (instrument: Orbitrap Ascend Tribrid MS).

### SDS–PAGE and mass spectrometry

Protein from chondrocytes of the growth plates of neonatal mice was assayed by mass spectrometry. The lysis buffer was made from PBS buffer, pH 7.2, with the addition of 2% SDS, 10% glycerol, 10 mM dithiothreitol (DTT), 1 mM EDTA and protease inhibitor mixture (Roche Applied Science). The total protein in the samples was estimated on a Coomassie Blue-stained SDS gel according to a standard protein marker with known concentration. For mass spectrometry analysis, proteins in each sample were separated on a 12% SDS gel (1.0 mm thick) and stained with Coomassie Blue G-250. The entire lane was cut into 15 pieces followed by in-gel trypsin digestion. Protein digestion was performed according to the FASP procedure described by Wisniewski et al.[54]. In brief, the protein fragments were solubilized in 30 μl SDT buffer (4% SDS, 100 mM DTT and 150 mM Tris-HCl pH 8.0) at 90 °C for 5 min. The detergent, DTT and other low-molecular-weight components were removed using 200 μl UA buffer (8 M urea and 150 mM Tris-HCl pH 8.0) by repeated ultrafiltration (Microcon units, 30 kDa). Then, 100 μl 0.05 M iodoacetamide in UA buffer was added to block reduced

cysteine residues and the samples were incubated for 20 min in darkness. The filter was washed with 100 μl UA buffer three times and then 100 μl 25 mM $NH_4HCO_3$ twice. Finally, the protein suspension was digested with 2 μg trypsin (Promega) in 40 μl 25 mM $NH_4HCO_3$ overnight at 37 °C, and the resulting peptides were collected as a filtrate.

Liquid chromatography–tandem mass spectrometry (LC–MS/MS) measurements were performed on an Easy-nano-LC (Thermo Fisher Scientific) coupled to an Q Exactive mass spectrometer (Thermo Fisher Scientific). Peptides were separated on a reverse-phase column (15 cm, 75-μm inner diameter and 3-μm Reprosil resin) using a 100-min gradient of water–acetonitrile. All MS measurements were performed in the positive ion mode. Each scan cycle consisted of one full scan mass spectrum ($m/z$ 300–1,800) followed by 20 MS/MS events of the most intense ions with the following dynamic exclusion settings: repeat count 2, repeat duration 30 s and exclusion duration 90 s. The samples were loaded onto the trap column first with 10 μl min$^{-1}$ flow rate, and then the desalted samples were eluted at a flow rate of 1,200 nl min$^{-1}$ in multidimensional liquid chromatography (MDLC) by applying a linear gradient of 0–50% B for 60 min. The Q Exactive mass spectrometer was used for the MS/MS experiment with ion transfer capillary of 160 °C and ISpary voltage of 3 kV. Normalized collision energy was 35.

All data files were created using Bioworks Browser rev. 3.1 (Thermo Electron) with precursor mass tolerance of 1.4 Da, threshold of 100, and minimum ion count of 10. The acquired MS/MS spectra were searched against the concatenated target/reverse Glycine_max database using the SEQUEST search engine (Proteome Discoverer Software 2.3.0.523). The target database contained Glycine_max protein sequences (80,292 entries) downloaded on 20 May 2010 from the NCBI database. Searches were performed in the trypsin enzyme parameter in the software. Methionine oxidation was only specified as a differential modification and cystine carbamidomethyl was the fixed ones. All output results were combined using in-house software named build summary. The filter was set to false discovery rate ≤ 0.01.

### Measurements of intracellular lactate and glucose by LC–MS

Knee cartilages isolated from newborn control and *Hbb*-cKO mutant mice were quickly frozen with liquid nitrogen. Samples were taken out after 24 h and added 100 μl water/50 mg to grind by tissue homogenizer. After that, tissue homogenates were shaken for 30 s and added with 400 μl methanol acetonitrile solution (1:1, v/v), followed by the second shock for 60 s. After ultrasonication at 4 °C for 30 min twice, the samples were placed at −20 °C for 1 h and centrifuged at 4 °C for 20 min (14,000 rcf), then the supernatants were collected to freeze-dry. Extracts were analysed by LC–MS on liquid chromatography system (1290 Infinity, Agilent) and AB Sciex API 5500 Qtrap mass spectrometer (AB Sciex). Details for high-performance LC are as follows: the sample was placed in an automatic sampler at 4 °C with column temperature of 45 °C, flow rate of 300 μl min$^{-1}$ and injection volume of 2 μl. 5500 QTRAP ESI source conditions are as follows: source temperature of 450 °C, ion source gas 1 (Gas1): 45, ion source gas 2 (Gas2): 45, curtain gas (CUR): 30, ionSapary voltage floating (ISVF): 4,500 V. The chromatographic peak area and retention time were extracted by Multiquant. The standard substance of energy metabolism was used to correct the retention time and identify the metabolites.

### Skeletal preparation

Whole-mount staining of skeletal preparation by alcian blue and Alizarin S red was performed. In brief, the embryos were skinned and eviscerated. After 4 days of fixation in 95% ethanol, embryos were stained in alcian blue solution overnight. After washing with 70% ethanol, the embryos were stained by Alizarin S red solution overnight and transferred into 1% KOH for 1 week. Finally, embryos were transferred into 1% KOH/20% glycerol for 2 days and stored in 50% ethanol/50% glycerol.

Images for skeletons were taken using a stereo microscope (SZX16, Olympus) equipped with a digital camera (DP71, Olympus).

## RNA isolation, reverse transcription and real-time PCR
The cartilaginous tissues or primary cultured chondrocytes from mice were collected and lysed in TRIzol (Invitrogen) for RNA isolation according to the manufacturer's standard protocol. cDNA was synthesized from 1 µg RNA Maxima First Strand cDNA Synthesis kit (Takara). Real-time PCR was performed on ABI Fast7500 with Maxima SYBR Green qPCR Master Mix (Takara). The primer pairs have been previously described[25,55-58] and are included in Supplementary Table 3. Fluorescence qPCR was performed by real-time fluorescence qPCR instrument (qTOWER[3]G, Jena Bioscience) and its application software (qPCRsoft 3.4). Real-time PCR results were analysed by Microsoft Excel (2306 Build 16.0.16529.20164).

## RNA-seq analysis
Cartilaginous tissues of knee joint of P6 mice were collected for RNA-seq analysis. An Agilent Bioanalyzer 2100 (Agilent Technologies) was used to check the integrity of the extracted and purified RNA. The TruSeq RNA sample preparation kit (Illumina) was used to generate the libraries. Libraries were sequenced using an Illumina HiSEq 2500 sequencer. Shanghai Biotechnology Corporation performed all the above processes.

## ChIP–qPCR
Mouse cartilages were used for the ChIP–qPCR experiment. The growth plates of P3 mice were sheared by scissors into small pierce and digested with 0.1% collagenase type II (Gibco) dissolved in α-MEM medium at 37 °C for 1 h. Then, the samples were repeatedly pipetted into single cells. The details of the procedure for the ChIP experiments have been previously described[59]. The following antibodies were used for ChIP–qPCR: KLF1 (1:50; 61233, Active motif), H3K4me3 (1:50; A2375, ABclonal), H3 (1:50; 17168-1-AP, Proteintech) and nonspecific IgG (rabbit, 1:50; 30000-0-AP, Proteintech). The primer pairs have been previously described[60,61] and are included in Supplementary Table 3.

## Human articular tissue samples
Human healthy articular cartilage specimens from patients with acute trauma were collected from the Department of Pathology of Xijing Hospital with informed consent and approval of the project by the Research Ethics Board of the Xijing Hospital, Fourth Military Medical University. Cartilage specimens were fixed in 4% paraformaldehyde for 48 h at room temperature and were decalcified in 0.5 M EDTA for 30–90 days and then embedded in paraffin. Serial sections were taken at 4 µm thickness and stained with H&E or Masson trichrome staining.

## Cell culture and compounds
The 293T, HepG2 and HCT116 cells were maintained in DMEM (Macgene Tech) supplemented with 10% FBS (Kang Yuan Biol) and 1% penicillin–streptomycin (Macgene Tech). PLC/PR/F5 cells were cultured in RPMI-1640 (Macgene Tech) supplemented with 10% FBS (Kang Yuan Biol) and 1% penicillin–streptomycin (Macgene Tech). PC12 cells were cultured in RPMI-1640 (Macgene Tech) supplemented with 10% horse serum (Kang Yuan Biol), 5% FBS (Kang Yuan Biol) and 1% penicillin–streptomycin (Macgene Tech). All cells were incubated with 5% $CO_2$ at 37 °C.

## Constructs
The plasmids expressing HBB and HBA protein in pQCXIP–eGFP or pQCXIP–mCherry were constructed in this study. The *Hbb* and *Hba* cDNAs were synthesized at Beijing Genomics Institute (Beijing, China). The WT *Hbb* and *Hba* and their mutants were cloned into pQCXIP–eGFP or pQCXIP–mCherry through seamless homologous recombination.

pBABE–CAAX–mCherry is maintained in the laboratory. Construction details are shown as Supplementary Table 4.

## Plasmid transfection
For plasmid transfection, about $3 \times 10^5$ cells were plated per well in 12-well plate precoated with type I collagen (354236, BD Bioscience) and cultured for 12 h. Cells were then transfected with the respective constructs by Lipofectamine 2000 reagent (11668019, Thermo Fisher Scientific) following the protocol provided.

## Quantification of HBB–eGFP foci
Wide-field imaging was performed on cells plated in glass bottom plate (Nest Biotechnology) 24 h after transfection as previously described[62]. The Nikon Ti-E microscope equipped with motorized stage and Neo Vacuum cooled Scientific CMOS Camera (Andor Technology) was used. Images were collected using ×10 or ×20 Apo objective lens with 15-ms exposure for differential interference contrast channel and 150-ms exposure for FITC channel. The number of cells expressing green fluorescence was counted as the denominator, and the number of foci-positive cells was counted as the numerator.

## Timelapse imaging
Cells were grown on a 15-mm glass bottom dish (Nest) and images were taken with the Ultraview Vox confocal system (Volocity 6.3.00, Perkin Elmer) system using a ×60 oil objective. For experiments of fusion of two HBB–eGFP foci, cells were imaged every 5 min, 24 h after transfection.

## Fluorescence recovery after photobleaching
Fluorescence recovery after photobleaching (FRAP) assay was conducted using the FRAP module of the Nikon confocal microscopy system. The HBB–eGFP was bleached using a 488-nm laser beam. Bleaching was focused on a circular region of interest using 100% laser power and timelapse images were collected. Fluorescence intensity was measured using Nikon confocal microscopy system (NIS Elements AR 4.50.00). Values are reported as ratios relative to pre-bleaching time points. GraphPad Prism is used to plot and analyse the FRAP results.

## Determination of oxygen dissociation curve and P50
The blood was collected by retro-orbital puncture into a blood collection tube (BD Vacutainer), centrifuged for 15 min at 400g to remove the supernatant, and WT and *Hbb*-cKO chondrocytes were obtained and as described above. RBCs containing 3 mg haemoglobin and $1 \times 10^7$ chondrocytes diluted in 4 ml of BLOODOX-Solution buffer were mixed with 20 µl BSA (Thermo Fisher Scientific) and 20 µl of anti-foaming agent (Sigma Aldrich). Oxygenation-dissociation analyser (BLOODOX-2018 Analyser, Softron Biotechnology) was used to determinate the oxygen dissociation curves. The sample buffer was drawn into a cuvette, equilibrated and brought to 37 °C, and oxygenated to 100% with air at the same time. After adjustment of the $pO_2$ (oxygen partial pressure) value, the sample was deoxygenated with nitrogen. A Clark oxygen electrode was used to detect changes of oxygen tension during the deoxygenation process on the $x$ axis of an $x$–$y$ recorder, whereas the deoxyhaemoglobin fraction was simultaneously monitored by dual-wavelength spectrophotometry at 560 nm and 570 nm, and displayed on the $y$ axis. Finally, the oxygen dissociation curve (ODC) was automatically recorded on graph paper, and the P50 value was extrapolated on the $x$ axis as the point at which $O_2$ saturation is 50%.

BLOODOX-Solution buffer includes 130 mM NaCl (Chinese Medicine Group Chemical Agent), 30 mM TES (Sigma Aldrich) and 5 mM KCl (Chinese Medicine Group Chemical Agent) at pH 7.4 ± 0.01.

## Extraction of Hedy consisting of HBA and HBB
About $2 \times 10^7$ 293T cells were plated in a 150-mm dish precoated with type I collagen (354236, BD Bioscience) and cultured for 12 h. pQCXIP–HBA1–mCherry and pQCXIP–HBB–eGFP were co-transfected into 293T

cells with a ratio of 1:8. The culture medium was removed 48 h after transfection, followed by digesting the cells and bursting them with 10 ml ddH₂o for 20 min at 4 °C. Then, the samples were centrifuged for 15 min at 3,000$g$ to remove the supernatant and resuspend with 4 ml of PBS with protease inhibitor cocktail (CW2200S, CWBIO). RBCs were obtained as described above.

## Hypoxia rescue by a nested co-culture

To explore whether Hedy could function as a source of oxygen under hypoxic conditions, a nested co-culture assay was developed. In brief, the hypoxia-sensitive PC12 cells were plated as hypoxia reporter cells in a 15-mm dish (Nest) precoated with type I collagen (354236, BD Bioscience) which was fixed inside a 35-mm dish (Nest) by acetone. Haemoglobin condensates of 4 ml or $5 \times 10^7$ RBCs were added into the 35-mm dish. The 35-mm dish was then sealed with plastic film and PARAFILM (Bemis) for 24 h. Subsequently, total RNA from PC12 cells was extracted for quantitative analysis by RT–qPCR with two pairs of primers: β-actin-F: GATCAAGATCATTGCTCCTCCTGA; β-actin-R: CAGCTCAGTAACAGTCCGCC; HIF1α-F: CCAGATTCAAGATCAGCCAGCA; HIF1α-R: GCTGTCCACATCAAAGCGTACTCA.

## RBC counts and haemoglobin analysis

Peripheral blood (20 µl) from each mouse born at 5 days (P5) was used for routine testing. Whole blood was collected by retro-orbital puncture into heparinized glass capillary tubes. RBC counts and haemoglobin concentration were performed by an automatic blood analyser (XP-100, Sysmex Corporation) and its application software (XT2000i1800i IPU).

## Analysis of total ROS

Primary chondrocytes were used to quantify total reactive oxygen species (ROS) accumulation with a ROS assay kit (Cayman Chemicals) according to the manufacturer's instructions. After a brief trypsinization with Trypsin-EDTA (0.05%) at 37 °C, cells were incubated in the dark with 5 mM dihydroethidium probe for 30 min at 37 °C. Thereafter, cells were washed twice with ROS staining buffer. Flow cytometry was performed with a flow cytometer (Coulter-XL). EXPO32 ADC software was used for analysis.

## Analysis of mitochondrial ROS

According to the manufacturer's instructions, the mitochondrial ROS accumulation was performed with a MitoSOX assay kit (Invitrogen). In brief, isolated and cultured primary chondrocytes were incubated with 5 mM MitoSOX probe for 20 min at 37 °C in the dark. After a brief trypsinization with Trypsin-EDTA (0.05%) at 37 °C, cells were washed twice with FACS buffer (1× DPBS + 5% FBS), and flow cytometry was performed with a flow cytometer (Coulter-XL). EXPO32 ADC software was used for analysis.

## Measurements of intracellular ATP and ADP

After tamoxifen treatment for 3 days, cartilages from growth plates of P5 $Hbb^{F/F}$ (control) or $Hbb^{F/F}/Col2a1$-$Cre^{ERT2}$ ($Hbb$-cKO) were collected and immediately quenched by liquid nitrogen. Upon evaporation of the liquid nitrogen, the cartilage were stored at −80 °C. Samples were analysed by LC–MS as described above.

## Counting of dead cells in vitro

Primary chondrocytes isolated from P5 $Hbb^{F/F}$ (control) or $Hbb$-cKO mice were cultured in vitro. After a brief trypsinization with Trypsin-EDTA (0.05%) and staining with Trypan blue, the number of alive and dead cells was counted.

## Hypoxia rescue by intracellular expression of haemoglobin

To further explore whether Hedies in cells could function as a source of oxygen under a hypoxic condition, intracellular expression of haemoglobin and cell culture in confined spaced were developed. In brief,

the $1 \times 10^5$ ATDC5 cells (ECACC) were plated in a 35-mm dish (Nest) and cultured for 12 h. Plasmids pQCXIP–HBA1–mCherry and pQCXIP–HBB–eGFP were co-transfected into 293T cells with a ratio of 1:8 (total was 900 ng) by Lipofectamine 2000 reagent (11668019, Thermo Fisher Scientific). Twenty-four hours after transfection, the growth medium was replaced with fresh culture medium pre-treated in 1% $O_2$ for 4 h. Immediately after, dishes were sealed with plastic film and PARAFILM (Bemis) for 2 h. Then, cells were fixed by 4% paraformaldehyde and stained with the antibody HIF1α (1:500, 36169, CST). Secondary antibodies were applied for 1 h followed by three 10-min washes with PBS followed by mounting with Antifade reagent with DAPI (Invitrogen). To detect the HIF1α signal, the Nikon Ti-E microscope equipped with motorized stage and a Neo Vacuum cooled Scientific CMOS Camera (Andor Technology) was used. Images were collected using ×60 Apo objective lens with DAPI, Cy3, Cy5 and FITC channel. Intensities of the HIF1α signal in the nucleus were entirely quantified, and the mean intensity value of haemoglobin foci-negative cells was counted as the denominator to normalized individual values.

## Immunofluorescence of lipid and RNA in mouse cartilage tissues and cells

The excised cartilage tissues from mice were snap frozen in liquid nitrogen and placed into OCT compound (Sakura Finetechnical) and snap-frozen again. Frozen tissue sections (6 µm thick) were obtained using a Leica cryostat (Leica).

For detecting exogenous haemoglobin bodies, about $3 \times 10^5$ PLC/PR/F5 cells were plated per well in 12-well plates precoated with type I collagen (354236, BD Bioscience) and cultured for 12 h. Cells were then transfected with the pQCXIP–HBB by Lipofectamine 2000 reagent (11668019, Thermo Fisher Scientific), and 6 h later, replaced with fresh culture medium. Twenty-four hours after transfection, cells were fixed with 4% paraformaldehyde.

All the samples were fixed with 4% paraformaldehyde for 15 min and blocked with 5% BSA for 30 min, and were incubated with the antibody HBB (1:50; PA5-60287, Invitrogen) and BODIPY (1:1,000; D3822, Thermo Fisher) labelling lipid or StrandBrite RNA Green (1:500; 23170, AAT Bioques) labelling RNA in a humidified box for 1 h, followed by three 10-min washes with PBS. Secondary antibodies were applied for 1 h, followed by three 10-min washes with PBS, mounted with Antifade reagent with DAPI (Invitrogen). A Nikon Ti-E microscope equipped with a motorized stage and Neo Vacuum cooled Scientific CMOS Camera (Andor Technology) was used. Images were collected using ×60 Apo objective lens with DAPI, Cy3 and FITC channel.

## Liquid–liquid phase separation assay in vitro

In vitro liquid phase separation assay was performed using phase buffer (150 mM $KH_2PO_4/K_2HPO_4$ (7778-77-0/7758-11-4, Solarbio) pH 7.35, PEG2000 (25322-68-3, Solarbio) 10%(w/v)) and recombinant haemoglobin α1 (RPD090Mu01, Cloud Clone), recombinant haemoglobin-β (RPD098Mu01, Cloud Clone).

For condition optimization, HBB protein was dissolved in ddH₂O to the concentration of 1 mg µl⁻¹, then diluted into final concentrations: 62.5, 125, 250 and 500 ng µl⁻¹ with ddH₂O and buffer (300 mM $KH_2PO_4/K_2HPO_4$, pH 7.35, and PEG2000 variable). The mixtures were spotted on a glass slide and sealed with coverslips. The slides were then imaged with the Nikon Ti-E microscope. Images were collected using ×60 Apo objective lens.

The concentration of HBB protein was 1 mg µl⁻¹ for PEG titration experiment in vitro. The concentrations of HBB and HBA protein were 0.5 mg µl⁻¹ for droplet formation in vitro. The concentrations of HBB IDR mutants were 1 mg µl⁻¹ for droplet formation in vitro. All the recombinant proteins were purchased from Detai Bioscience. Buffer (150 mM $KH_2PO_4/K_2HPO_4$, pH 7.35, and PEG2000 10%) was used in all experiments except for the PEG titration experiment.

HBB−eGFP recombinant protein was purified by Solarbio Clone, powder was dissolved in $H_2O$ to the concentration of 1 mg µl$^{-1}$, and then diluted into final concentrations: 500 ng µl$^{-1}$ with LLPS buffer (300 mM $KH_2PO_4$/$K_2HPO_4$, pH 7.35, and PEG2000 20%) and was detected with FITC channel.

## Western analysis for exogenous HBB and its mutant expression

For plasmid transfection, about $6 × 10^5$ 293T cells were plated per well in a six-well plate precoated with type I collagen (354236, BD Bioscience) and cultured for 12 h. Cells were then transfected with the same dose-respective constructs by Lipofectamine 2000 reagent (11668019, Thermo Fisher Scientific), and 6 h later, replaced with fresh culture medium. Twenty-four hours after transfection, the cells were washed three times in 0.1 M cold PBS, then, lysed into RIPA buffer containing protease inhibitor cocktails (Roche) and fragmented by sonication. Details of the western blotting method are as previously described. Antibody (1:1,000; 2555, CST) to GFP was used to recognize eGFP-tagged HBB and its mutants.

## Correlative light electron microscopy

For plasmid transfection, about $3 × 10^5$ HepG2 cells were plated per well in a 12-well plate precoated with type I collagen (354236, BD Bioscience) and cultured for 12 h. Cells were then transfected with the *Hbb*−eGFP expression vector by Lipofectamine 2000 reagent (11668019, Thermo Fisher Scientific) following the protocol provided. Twelve hours after transfection, cells were cultured on glass bottom Grid-500 dishes (81168, ibidi). Thirty-six hours after transfection, the cells were fixed with 2% paraformaldehyde (16005, Sigma) in 0.15 M PBS buffer pH 7.35 for 30 min at room temperature. Once the cells with the condensates were found, their positions on the grid were documented by switching from fluorescence to differential interference contrast mode. After fluorescence imaging, the selected areas with positive cells were marked to facilitate the processing of electron microscopy. After observation with a confocal microscope (Zeiss LSM 980) and image processing with accompanying software (ZEN 3.6095.01), the samples were post-fixed in 2.5% glutaraldehyde (25% glutaraldehyde ampules; 111-30-8, SPI) in 0.1 M phosphate buffer pH 7.4 at 4 °C overnight. After washing with phosphate buffer two times and ddH$_2$O two times, the samples were fixed with 1% OsO$_4$ (w/v) and 1.5% (w/v) potassium ferricyanide aqueous solution at 4 °C for 1.5 h, followed by sequential washing with ddH$_2$O (three times). The samples were then stained with 2% UA for 1 h, followed by washing with ddH$_2$O three times. Then, the samples were dehydrated by incubating with ethanol (30, 50, 70, 80, 90 and 100% two times), followed by incubation with acetone two times. After hydration, the samples were subjected to infiltration and embedding step, in which samples were infiltrated with SPI-PON812 resin (SPI) as follows: 3:1, 1:1 and 1:3 acetone:resin and 100% resin two times, followed by samples being embedded and polymerized with resin for 48 h at 60 °C. Eventually, the serial ultrathin sections (100 nm thick) were formed using the ultramicrotome (UC7, Leica) with the AutoCUTS (Zhenjiang Lehua Technology) device, then double-stained by uranyl acetate and lead citrate. The serial sections were finally automatically acquired by a Helios Nanolab 600i dual-beam SEM (Thermo Fisher) with an automated imaging software (AutoSEE 1.58).

## Statistical analysis

Quantitative data presented as bar plots show mean ± s.e.m. as well as individual data points where appropriate, with each point representing a biological replicate. Statistical analyses were performed using GraphPad Prism 8.0.1. An unpaired two-tailed Student's *t*-test was used to determine significance between two groups of normally distributed data. For comparisons between multiple groups with one fixed factor, an ordinary one-way analysis of variance (ANOVA) was used, followed by Dunnett. One-way ANOVA followed Bonferroni's multiple comparison test was used to compare the multiple groups

to a single control group, using and comparing each cell mean with the control cell mean. In cases in which more than one comparison has the same statistical range, values are listed as they appear from left to right in the corresponding panel. Exact sample sizes, statistical tests and *P* values can be found in the figure legends and/or source data. For all data, differences were considered significant when $P < 0.05$. At least three biological replicates were performed for all in vivo experiments.

For non-quantitative data (micrographs, western blots and so on), findings were reproduced at least three times by replicating the experiments and/or cross-validating with orthogonal approaches, and representative results are shown.

## Statistics and reproducibility

At least three biological replicates were performed for all in vivo experiments, and most in vitro experiments were repeated at least three times. For non-quantitative data (micrographs, western blots and so on), findings were reproduced at least three times by replicating the experiments and/or cross-validating with orthogonal approaches, and representative results are shown. Details of each exact number of replicates are provided in the figure legends.

## Reporting summary

Further information on research design is available in the Nature Portfolio Reporting Summary linked to this article.

## Data availability

RNA-seq data of the present study have been deposited in the NCBI Gene Expression Omnibus with the accession code GSE182640. Bioinformatics analysis of the H3K4me3 modification region was performed on the Cistrome Data Browser (http://cistrome.org/db/#/). Histone methylation data were from work done by Ohba et al. (https://doi.org/10.1016/j.celrep.2015.06.013). Data of microdissection sample mass spectrometry, SDS−PAGE sample mass spectrometry, oligonucleotides and construct information have been deposited in supplementary tables. Source data are provided with this paper.

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

**Acknowledgements** We are grateful to J. Ding (Institute of Biophysics, Chinese Academy of Sciences) for consultation on LLPS; X. Li and X. Tan for helping with electron microscopy sample preparation and scanning electron microscopy imaging; and Y. Teng and X. Jia for fluorescence

imaging at the Center for Biological Imaging (CBI), Institute of Biophysics, Chinese Academy of Science. This work was supported by the National Key Research & Development Program of China (2022YFC3600100 to Q.S. and 2022YFA0912400 to H.H.), the National Natural Science Foundation of China (81572631 and 31000559 to F.Z., 31970685 to Q.S., 81772865 to S.G. and 82273184 to H.H.), the CAMS Innovation Fund for Medical Sciences (2021-I2M-5-008 to Q.S.), Shaanxi Society Development Sci-Tech Research Project (2016SF-064 to F.Z.), and the State Key Laboratory of Cancer Biology, the Fourth Military Medical University (CBSKL2019ZZ28 and CBSKL2022ZZ29 to F.Z.).

**Author contributions** F.Z. and Q.S. conceived the project. F.Z. performed most of the experiments, including the discovery of haemoglobin structures in cartilage and phenotyping, component characterization and expression regulation by hypoxia, and functional evaluation in vivo and in vitro under hypoxic stress. Q.S. found the phase-separation feature of Hedy and created the concept. B.Z. performed the experiments related to phase separation and oxygen rescue assay, and measured oxygen dissociation curves with help from X.G and Y. Wei. Mouse breading was performed by Y. Wang, R.J., X.W., B.Z. and Y.Z. Scanning electron microscopy and TEM were done by F.Z., J.K., B.Z. and Y.L. Western blot and genotyping were performed by F.Z. and J.L. Circulating RBC and haemoglobin level testing was done by F.Z., J.X. and J.L. Laser capture microdissection and LC–MS/MS were performed by F.Z., M.S. and D.W. Data interpretation was done by F.Z., Q.S. and B.Z. Figures were created by Q.S. and F.Z. The manuscript was written by Q.S. and F.Z. with input from J.B., M.W., Y.G., H.H., Z.W., J.Y. and S.G. Funding was acquired by F.Z., Q.S., H.H. and S.G. All authors read and approved the final manuscript.

**Competing interests** The authors declare no competing interests.

**Additional information**
**Correspondence and requests for materials** should be addressed to Feng Zhang or Qiang Sun.

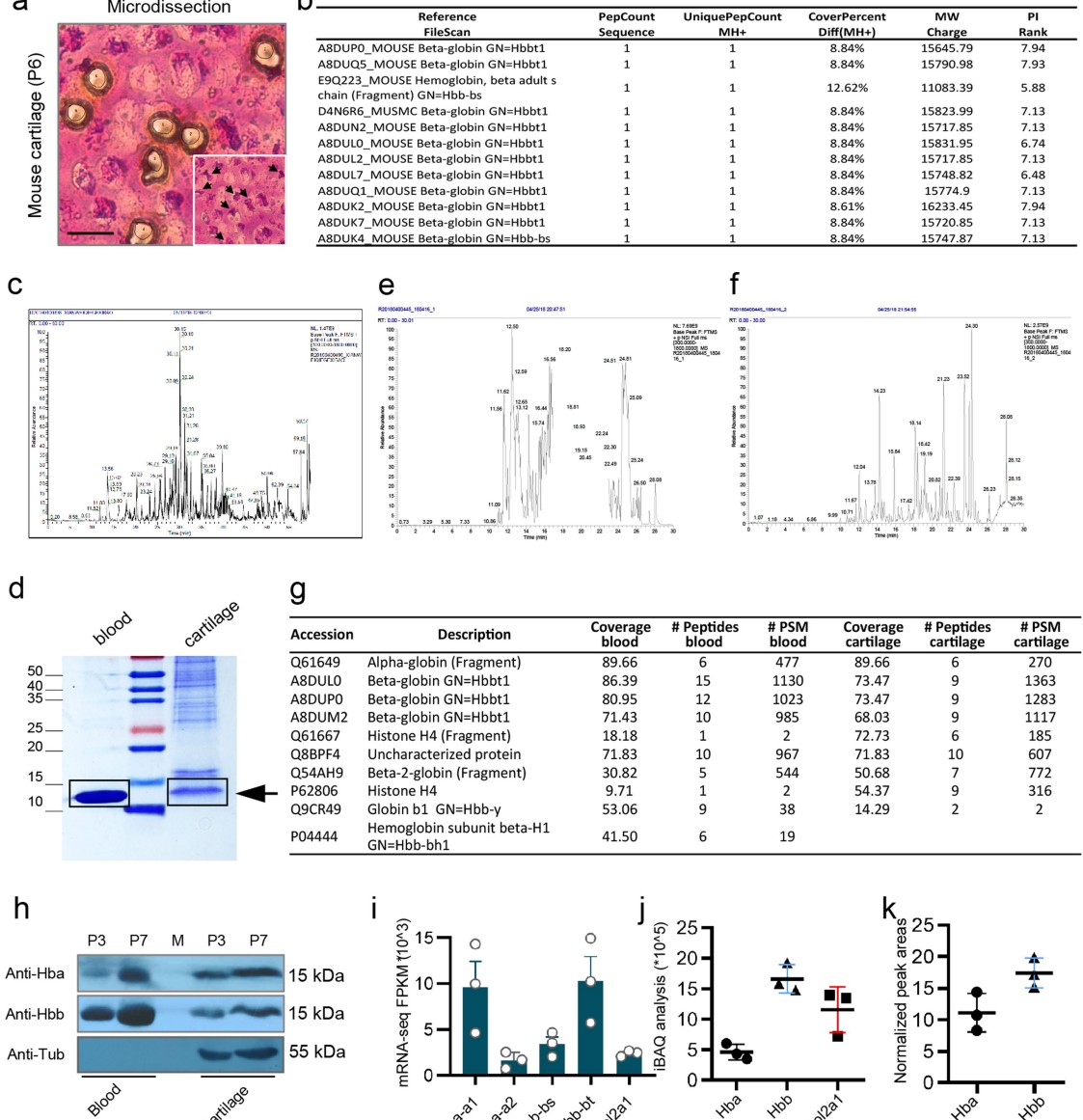

**Extended Data Fig. 1 | Identification of hemoglobin in chondrocytes.**
a, Images for cartilages of P6 (6 days postnatal) mice before (insert) and post laser-based microdissection. Black arrows indicated eosin-positive structures. Scale bar: 10 μm. b, List of top proteins in the eosin-positive structures identified by mass spectrometry analysis. For whole data, see Supplementary Table 1. c, Mass spectrometry analysis of samples from laser capture microdissection. The samples dissected from about 200 chondrocytes were pooled into the cap of a 0.5-mL microcentrifuge tube and dissolved in tissue extraction buffer for mass spectrometry. d, SDS-PAGE of protein extracts from chondrocytes and blood. Black arrow and rectangles indicated the position of hemoglobin. e, f, Mass spectrometry analysis of proteins from red blood cells (e) and cartilages (f) on SDS-PAGE gel. g, h, Mass spectrometry (g) and Western blot (h) confirmed the presence of hemoglobin beta subunit (Hbb), and alpha subunit (Hba) in cartilages. For whole data, see Supplementary Table 2. For gel source data, see Supplementary Fig. 1a. i-k, RNA-seq (i) and quantitative proteomic analysis (j, k) to quantify the expression of Hba and Hbb in cartilaginous tissues. Data are presented as mean ± SEM, Error bars represent SEM. (n = 3 biologically independent samples).

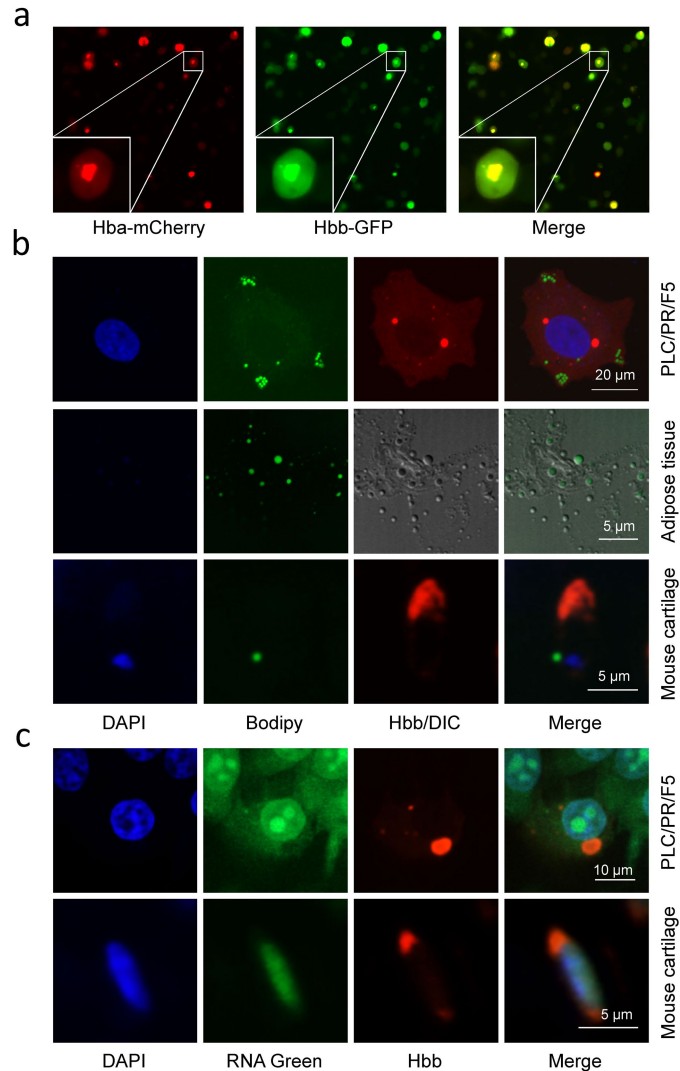

**Extended Data Fig. 2 | Spatial localization of lipid, RNA and Hedy.**
**a**, Representative images of hemoglobin bodies formed with Hba-mCherry and
Hbb-EGFP in HEK293T cells. **b**, Representative images for lipid and hemoglobin
staining in PLC/PR/F5 cells, adipose tissue, cartilage tissue. Nuclei are in blue,
Hbb in red, lipid droplets in green. Scale bars are indicated. **c**, Representative
images for RNA and hemoglobin staining in PLC/PR/F5 cells and cartilage
tissue. Nuclei are in blue, Hbb in red, RNA in green. Scale bars are indicated.

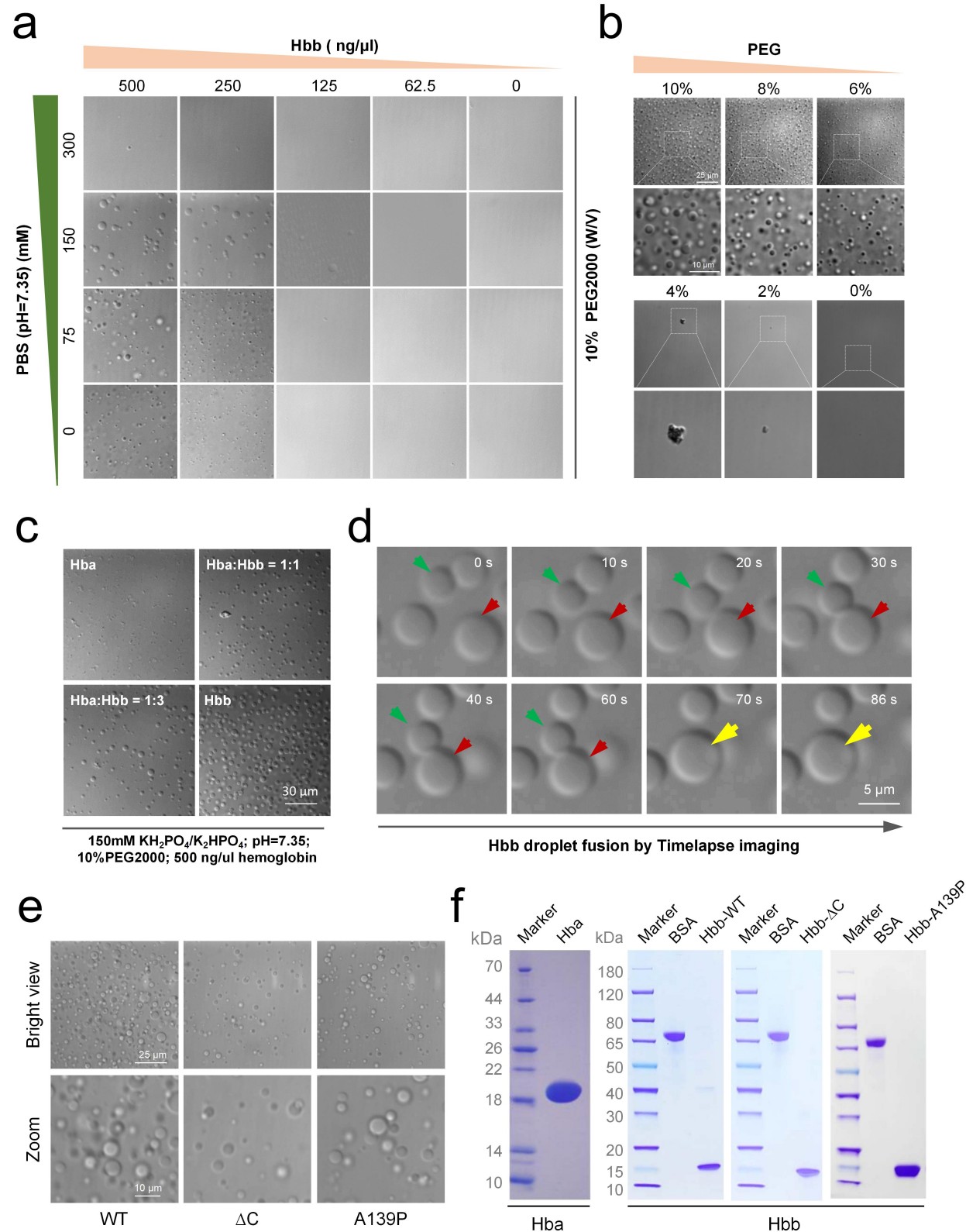

**Extended Data Fig. 3 | Liquid phase separation of the purified untagged Hbb in vitro. a**, Droplet formation of the purified untagged Hbb in different conditions, buffer (PBS: KH₂PO₄/K₂HPO₄, pH 7.35, PEG2000 10% (w/v)). Scale bar: 20 μm. **b**, Droplet formation of the purified untagged Hbb at different PEG concentrations in phase separation buffer (150 Mm KH₂PO₄/K₂HPO₄, pH 7.35, PEG2000 variable (w/v)). Scale bar: 20 μm. **c**, Droplet formation of the purified untagged Hba, Hbb and the mixture of them at the same protein concentration in phase separation buffer (150 mM KH₂PO₄/K₂HPO₄, pH 7.35, PEG2000 10% (w/v)). Scale bar: 30 μm. **d**, Timelapse imaging of the purified untagged Hbb droplet fusion. Scale bar: 5 μm. **e**, Droplet formation of the purified Hbb and its IDR mutants in phase separation buffer (150 mM KH₂PO₄/K₂HPO₄, pH 7.35, PEG2000 10% (w/v)). Scale bar: 25 μm, 10 μm. **f**, Coomassie brilliant blue stained gels of the purified Hba, Hbb and Hbb mutants.

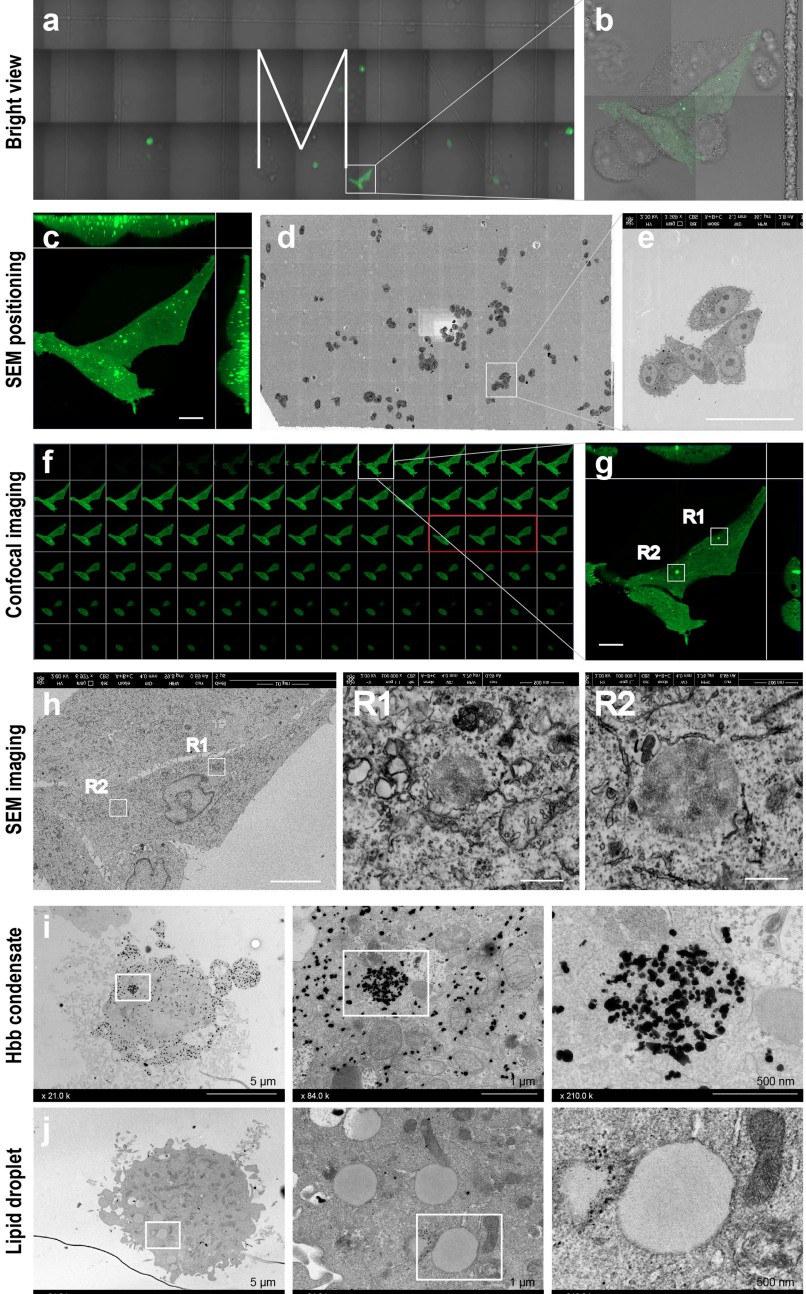

**Extended Data Fig. 4 | Identification of Hbb condensates by correlative light and electron microscopy and immune electron transmission microscopy. a**, Large field view of cells plated on grid glass bottom dish in merged bright and fluorescent channels. M marks the grid with cells of interest. **b**, zoom-in view of cells within the boxed region in (**a**). **c**, composite Z-stack view of cells expressing Hbb-EGFP in (**b**) by confocal microscopy. Scale bar: 10 μm. **d**, ultra-thin section image of cells within the field of (**a**) by scanning electron microscopy. **e**, zoom-in view of cells within the boxed region in (**d**). Scale bar: 50 μm. **f**, split confocal images for the cells of interest in (**c**) from a row of bottom-up with a Z-step of 0.15 μm. the images within red box are corresponding to the view in (**e**). **g**, zoom-in view of image within the boxed region in (**f**). Hbb-EGFP condensates are indicated as R1 and R2. Scale bar: 10 μm. **h**, scanning electron microscopy image (left) corresponding to the confocal image (**g**). Scale bar: 10 μm. R1 and R2 regions are displayed in the right in zoomed view. Scale bar: 500 nm. **i**, Hbb condensate identified by immune electron transmission microscopy. Boxed regions are zoomed in the right. Scale bars are indicated. **j**, Lipid droplets detected by electron transmission microscopy. Boxed regions are zoomed in the right. Scale bars are indicated.

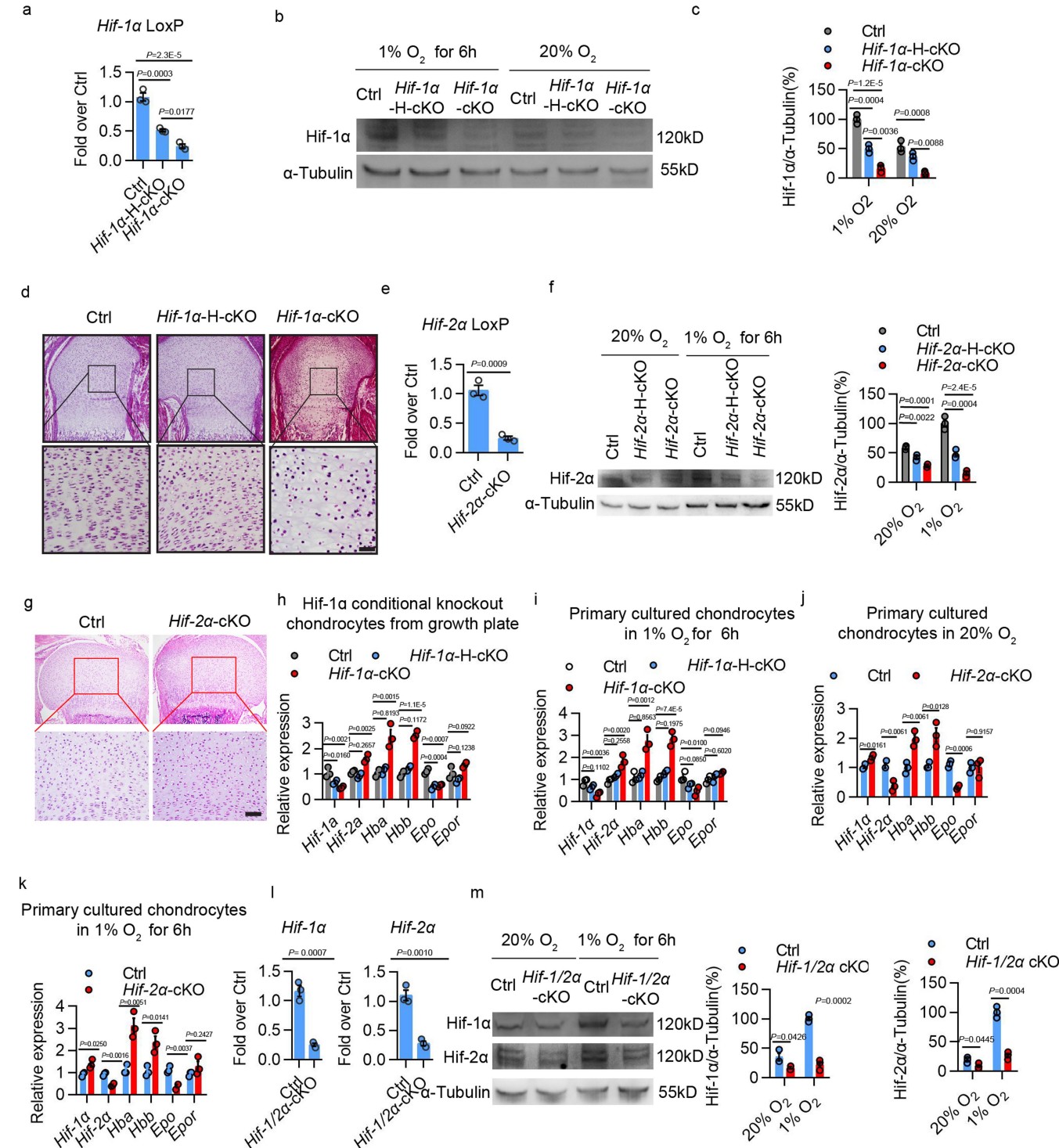

**Extended Data Fig. 5** | See next page for caption.

**Extended Data Fig. 5 | Expression of hemoglobin in chondrocytes with *Hif-1α* and/or *Hif-2α* deletion. a**, Detection of DNA recombinatrion by *Hif-1α*-LoxP qPCR of genomic DNA extracted from *Hif-1α*$^{F/F}$ (Ctrl), *Hif-1α*$^{F/+}$/*Col2a1-Cre*$^{ERT2}$ (heterozygous deletion, *Hif-1α*-H-cKO) or *Hif-1α*$^{F/F}$/*Col2a1-Cre*$^{ERT2}$ (*Hif-1α*-cKO) growth plates of P5 mice, which were treated by tamoxifen (100 mg/kg) for 4 days. Data were normalized to β2-microglobulin (n = 3 biologically independent samples). **b**, Western blot analysis of Hif-1α protein level in primary cultured chondrocytes in either 20% or 1% $O_2$ for 6 h. For gel source data, see Supplementary Fig. 1c. **c**, Quantification of all biological replicates for Extended Data Fig. 6b. Data were normalized to α-tubulin (n = 3 biologically independent samples). **d**, Conditional knockout of *Hif-1a* resulted in massive cell death in the center of cartilaginous growth plate. Scale bar: 50 μm. n = 6 biologically independent samples. **e**, *Hif-2α*-LoxP qPCR of genomic DNA extracted from *Hif-2α*$^{F/F}$ (Ctrl) or *Hif-2α*$^{F/F}$/*Col2a1-Cre*$^{ERT2}$ (*Hif-2α*-cKO) growth plates of P5 mice, which were treated by tamoxifen (100 mg/kg) for 4 days. Data were normalized to β2-microglobulin (n = 3 biologically independent samples). **f**, Quantification of Hif-2α protein by Western blot analysis of total protein lysate extracted from Ctrl (*Hif-2α*$^{F/F}$), *Hif-2α*-H-cKO (heterozygous deletion, *Hif-2α*$^{F/+}$/*Col2a1-Cre*$^{ERT2}$) or *Hif-2α*-cKO (*Hif-2α*$^{F/F}$/*Col2a1-Cre*$^{ERT2}$) primary cultured chondrocytes in either 20% or 1% $O_2$ for 6 h. A representative Western blot is shown on the left, and quantification of all biological replicates is provided on the right. Data were normalized to α-tubulin (n = 3). For gel source data, see Supplementary Fig. 1d. **g**, Conditional knockout of *Hif-2a* resulted in no massive cell death in the center of P5 *Hif-2α*-cKO cartilaginous growth plate. Scale bar: 50 μm. n = 6 biologically independent samples. **h**, Expression of the indicated genes in chondrocytes of mouse fetal growth plates upon conditional knockout of *Hif-1α*. n = 3 biologically independent samples. **i**, Expression of the indicated genes in primary cultured chondrocytes of mouse fetal growth plates upon conditional knockout of *Hif-1α* in 1% $O_2$ for 6 h. n = 3 biologically independent samples. **j**, **k**, Expression of the indicated genes in primary cultured chondrocytes of mouse fetal growth plates upon conditional knockout of *Hif-2α* in either 20% (**j**) or 1% $O_2$ (**k**) for 6 h. n = 3 biologically independent samples. **l**, *Hif-1α* and *Hif-2α* (*Hif-1/2α*) LoxP qPCR of genomic DNA extracted from *Hif-1α*$^{F/F}$/*Hif-2α*$^{F/F}$ (Ctrl) or *Hif-1α*$^{F/F}$/*Hif-2α*$^{F/F}$/*Col2a1-Cre*$^{ERT2}$ (*Hif-1/2α*-cKO) growth plates of P5 mice, which were treated by tamoxifen (100 mg/kg) for 4 days. Data were normalized to β2-microglobulin (n = 3 biologically independent samples). **m**, Quantification of Hif-1α and Hif-2α protein by Western blot analysis of total protein lysate extracted from Ctrl or *Hif-1/2α*-cKO primary cultured chondrocytes in either 20% or 1% $O_2$ for 6 h. A representative Western blot is shown on the left, and quantification of all biological replicates is provided on the right. Data were normalized to α-tubulin (n = 3 biologically independent samples). For gel source data, see Supplementary Fig. 1e. Error bars represent SEM. *P* values were calculated using one-way ANOVA tests (**a**, **c**, **f**, **h**, **i**) or two-tailed Student's *t*-test (**e**, **j**, **k**, **m**, **l**). The exact *P*-values of comparison are presented in the figures, respectively.

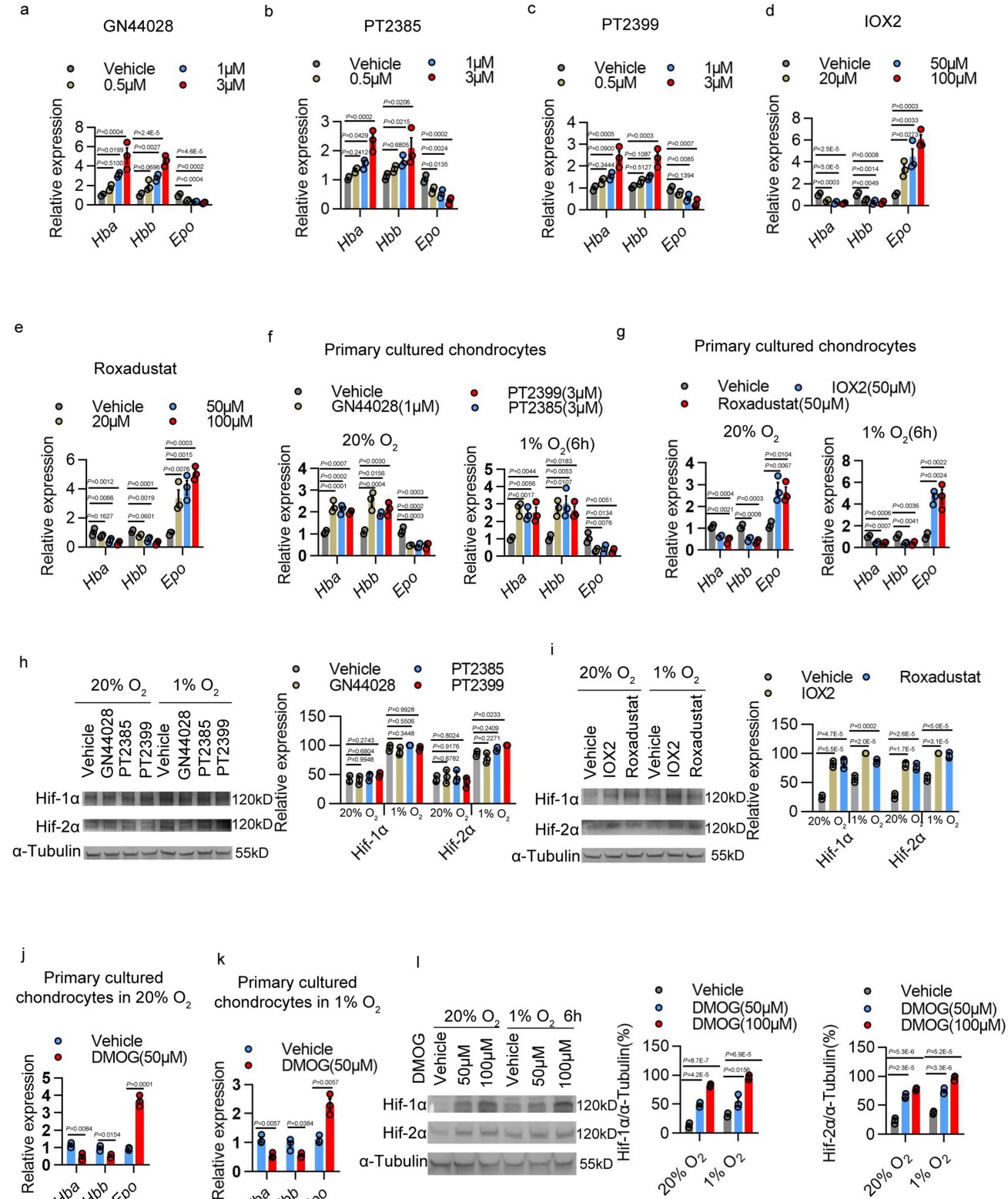

**Extended Data Fig. 6** | See next page for caption.

**Extended Data Fig. 6 | Effects of compounds regulating Hifs activity on hemoglobin expression. a**-**e**, Expression of *Hba*, *Hbb* and *Epo* mRNA of primary cultured chondrocytes in 20% $O_2$ with increased dose of compounds targeting either Hif-1α or Hif-2α for 6 h. The inhibitor of Hif-1α: GN44028. The inhibitors of Hif-2α: PT2385 or PT2399. The activator of Hif-1α and Hif-2α: IOX2 or Roxadustat. n = 3 biologically independent experiments. **f**, **g**, Expression of *Hba*, *Hbb* and *Epo* mRNA of primary cultured chondrocytes in either 20% or 1% $O_2$ with compounds targeting either Hif-1α or Hif-2α for 6 h. n = 3 biologically independent experiments. **h**, **i**, Quantification of Hif-1α or Hif-2α protein by Western blot analysis of total protein lysate extracted from primary cultured chondrocytes in either 20% or 1% $O_2$ with compounds targeting either Hif-1α or Hif-2α for 6 h. A representative Western blot is shown on the left. Quantification of all biological replicates is provided on the right. Data for Hif-1α or Hif-2α were normalized to α-tubulin (n = 3 biologically independent experiments). For gel source data, see Supplementary Fig. 1f,g. **j**, **k**, Expression of *Hba*, *Hbb* and *Epo* mRNA of primary cultured chondrocytes in either 20% (**j**) or 1% (**k**) $O_2$ with compound DMOG (50 μM), the activator of Hif-1α and Hif-2α for 6 h. n = 3 biologically independent experiments. **l**, Quantification of Hif-1α or Hif-2α protein by Western blot analysis of total protein lysate extracted from primary cultured chondrocytes in either 20% or 1% $O_2$ with compound DMOG for 6 h. A representative Western blot is the left. Quantification of all biological replicates is provided on the right. Data for Hif-1α or Hif-2α were normalized to α-tubulin (n = 3 biologically independent experiments). For gel source data, see Supplementary Fig. 1h. Error bars represent SEM. *P* values were calculated using one-way ANOVA tests (**a**-**i**, **l**) or two-tailed Student's *t*-test (**j**, **k**). The exact *P*-values of comparison are presented in the figures, respectively.

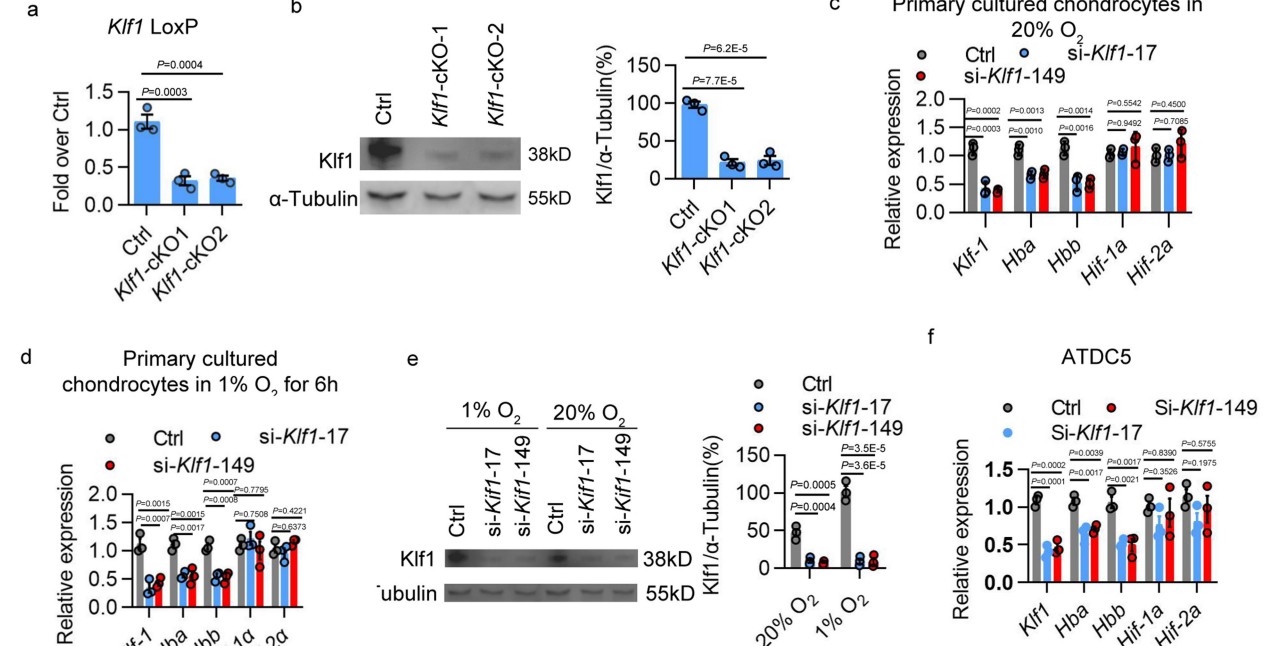

**Extended Data Fig. 7 | Regulation of hemoglobin expression by Klf1.**
**a**, *Klf1*-LoxP qPCR of genomic DNA extracted from *Klf1*^F/F (Ctrl) or *Klf1*^F/F/
*Col2a1-Cre*^ERT2 (*Klf1*-cKO) growth plates of P5 mice, which were treated by
tamoxifen (100 mg/kg) for 4 days. Data were normalized to β2-microglobulin
(n = 3 biologically independent samples). **b**, Quantification of Klf1 protein by
Western blot analysis of total protein lysate extracted from Ctrl or *Klf1*-cKO
primary cultured chondrocytes. A representative Western blot is shown on the
left, and quantification of all biological replicates is provided on the right. Data
were normalized to α-tubulin (n = 3 biologically independent samples). For gel
source data, see Supplementary Fig. 1i. **c, d** Expression of *Klf1*, *Hba*, *Hbb*, *Hif-1α*
and *Hif-2α* in the primary chondrocytes upon *Klf1* depletion by RNA interference

in either 20% (**c**) or 1% (**d**) O₂. n = 3 biologically independent experinmets.
**e**, Quantification of Klf1 protein by Western blot analysis of total protein lysate
extracted from Ctrl or *Klf1* depletion by RNA interference primary cultured
chondrocytes in either 20% or 1% O₂. A representative Western blot is shown on
the left, and quantification of all biological replicates is provided on the right.
Data were normalized to α-tubulin (n = 3 biologically independent experinmets).
For gel source data, see Supplementary Fig. 1j. **f**, Expression of *Hba*, *Hbb*, *Hif-1α*
and *Hif-2α* in ATDC5 chondrocyte cell lines upon *Klf1* knockdown as examined
by quantitative PCR. The data are mean with SEM of triplicate experiments.
Error bars represent SEM. *P* values were calculated using one-way ANOVA tests
(**a-f**). The exact *P*-values of comparison are presented in the figures, respectively.

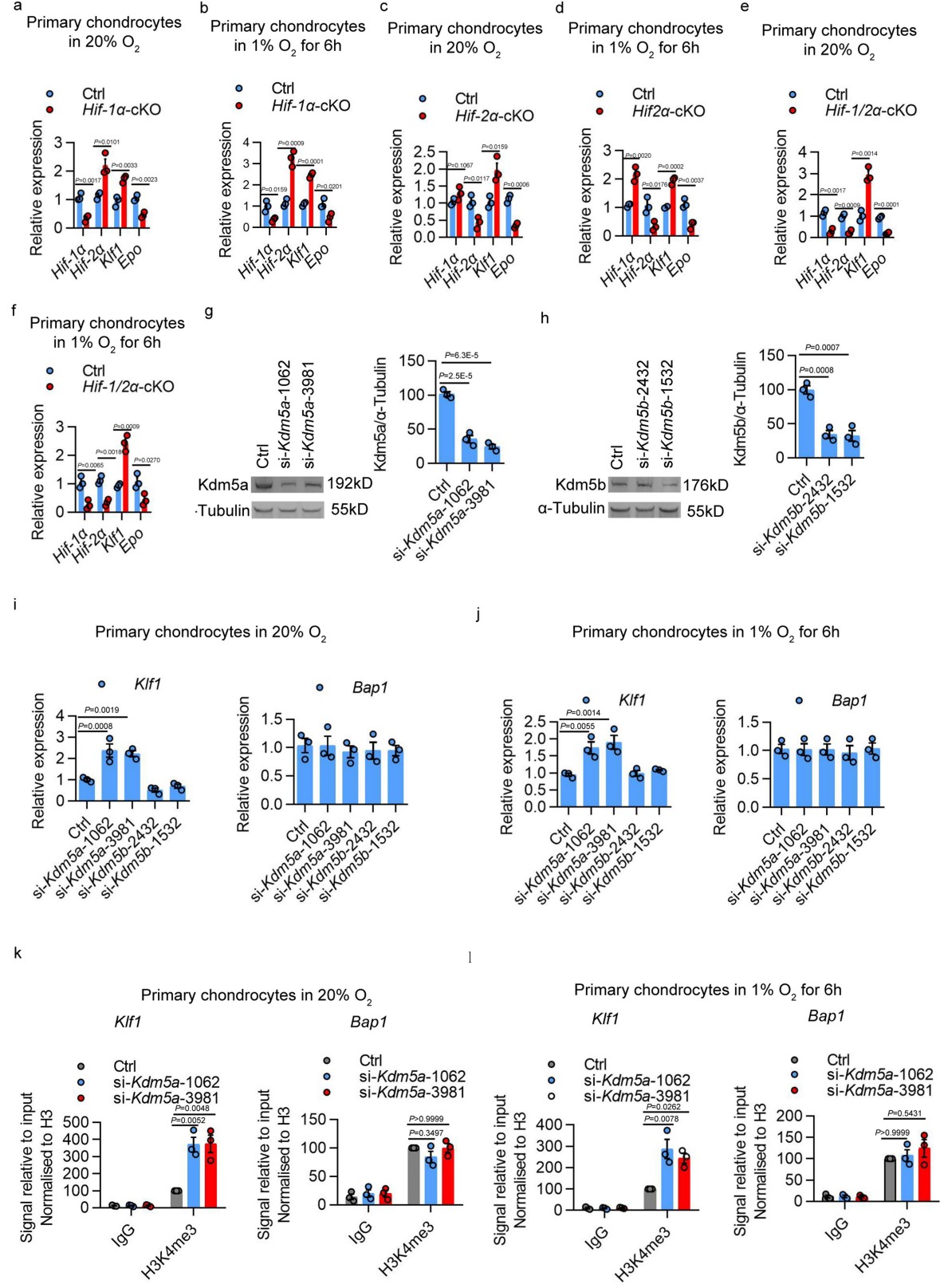

**Extended Data Fig. 8** | See next page for caption.

**Extended Data Fig. 8 | H3K4me3 demethylase Kdm5a rather than Kdm5b modulates *Klf1* expression. a**, **b**, Expression of the indicated genes in primary cultured chondrocytes of mouse fetal growth plates upon conditional knockout of *Hif-1α* in either 20% $O_2$ (**a**) or 1% $O_2$ for 6 h (**b**). n = 3 biologically independent samples. **c**, **d**, Expression of the indicated genes in primary cultured chondrocytes of mouse fetal growth plates upon conditional knockout of *Hif-2α* in either 20% $O_2$ (**c**) or 1% $O_2$ for 6 h (**d**). n = 3 biologically independent samples. **e**, **f**, Expression of the indicated genes in primary cultured chondrocytes of mouse fetal growth plates upon conditional knockout of both *Hif-1α* and *Hif-2α* (*Hif-1/2α*) in either 20% $O_2$ (**e**) or 1% $O_2$ for 6 h (**f**). n = 3 biologically independent samples. **g**, Quantification of Kdm5a protein by Western blot analysis of total protein lysate extracted from Ctrl or si-*Kdm5a* primary cultured chondrocytes. A representative Western blot is shown on the left, and quantification of all biological replicates is provided on the right. Data were normalized to α-tubulin (n = 3 biologically

independent experiments). For gel source data, see Supplementary Fig. 1k. **h**, Quantification of Kdm5b protein by Western blot analysis of total protein lysate extracted from Ctrl or si-*Kdm5b* primary cultured chondrocytes. A representative Western blot is shown on the left, and quantification of all biological replicates is provided on the right. Data were normalized to α-tubulin (n = 3 biologically independent experiments). For gel source data, see Supplementary Fig. 1l. **i**, **j**, Expression of the indicated genes in primary cultured chondrocytes upon knockdown of *Kdm5a* or *Kdm5b* in either 20% (**i**) or 1% $O_2$ (**j**) for 6 h. n = 3 biologically independent experiments. **k**, **l**, ChIP-qPCR analysis results of H3K4me3 for the indicated genes in primary cultured chondrocytes siRNA depleted of *Kdm5a* in either 20% (**k**) or 1% $O_2$ (**l**). The data are mean with SEM of triplicate experiments. *P* values were calculated using two-tailed Student's *t*-test (**a-f**) or one-way ANOVA tests (**g-l**). The exact *P*-values of comparison are presented in the figures, respectively.

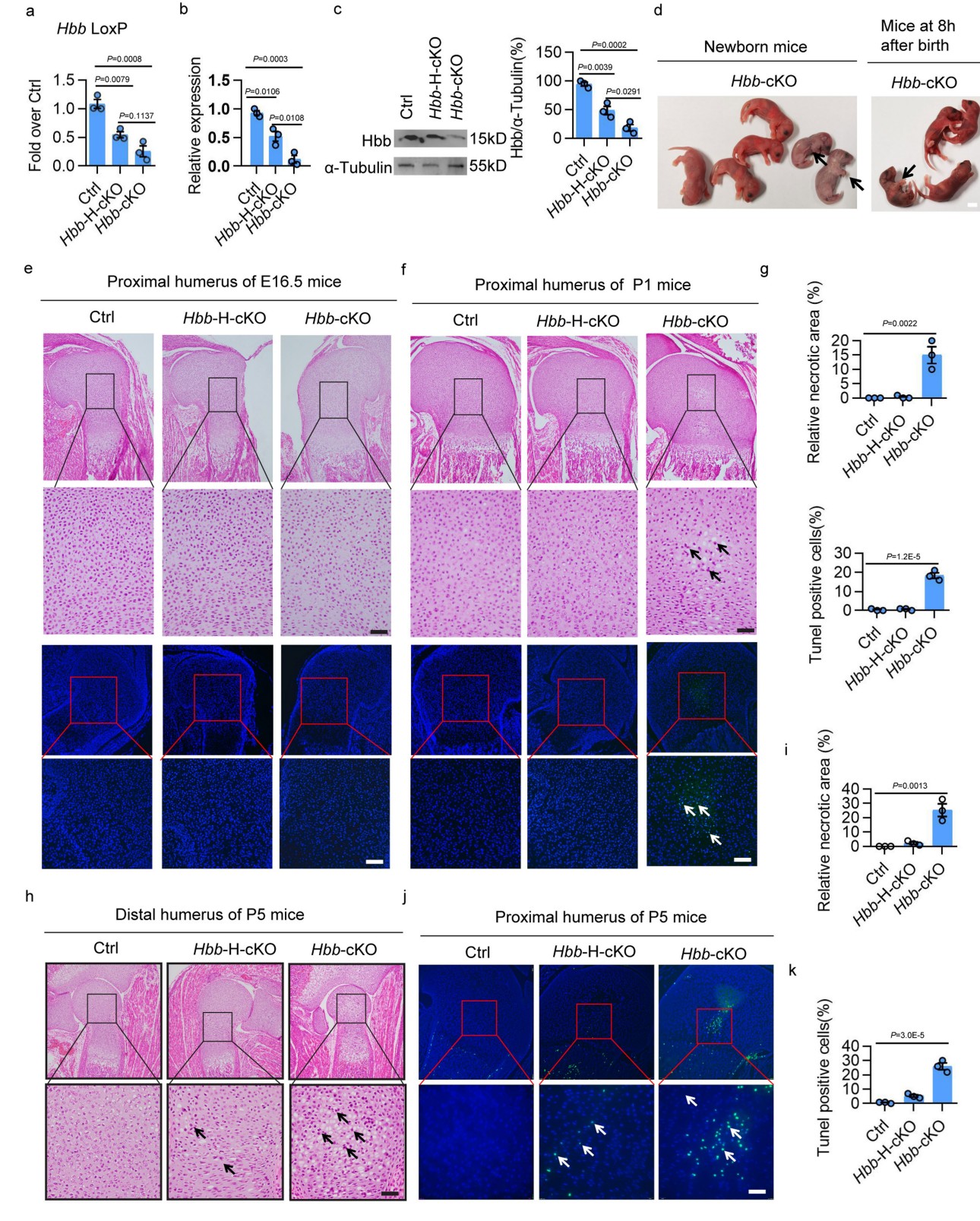

**Extended Data Fig. 9** | See next page for caption.

**Extended Data Fig. 9 | Phenotypes of mice with *Hbb* conditional knockout by *Col2a1-Cre*. a**, *Hbb*-LoxP qPCR of genomic DNA extracted from *Hbb*$^{F/F}$ (Ctrl), *Hbb*$^{F/+}$/*Col2a1-Cre*$^{ERT2}$ (heterozygous deletion, *Hbb*-H-cKO) or *Hbb*$^{F/F}$/*Col2a1-Cre*$^{ERT2}$ (*Hbb*-cKO) growth plates of P5 mice, which were treated by tamoxifen (100 mg/kg) for 4 days. Data were normalized to β2-microglobulin (n = 3 biologically independent samples). **b**, Quantification of *Hbb* mRNA by qRT-PCR of total RNA extracted from P5 *Hbb*$^{F/F}$ (Ctrl), *Hbb*-H-cKO or *Hbb*-cKO growth plates. Data were normalized to *Gapdh* (n = 3 biologically independent samples). **c**, Quantification of Hbb protein by Western blot analysis of total protein lysate extracted from P5 *Hbb*$^{F/F}$ (Ctrl), *Hbb*-H-cKO or *Hbb*-cKO growth plates. A representative Western blot is shown on the left, and quantification of all biological replicates is provided on the right. Data were normalized to α-tubulin (n = 3 biologically independent samples). For gel source data, see Supplementary Fig. 1q. **d**, For the *Hbb*-cKO mice, two of six newborn mice were dead fetus (left panel, black arrows). One more newborn died with purplish skin (right figure, black arrow) 8 hours after birth. The other three living newborn mice died in 1-7 days after birth with purplish skin (Supplementary Video 1). Scale bars: 3 mm. **e**, Histology examination (two upper panels) or TUNEL assay (two bottom panels) of proximal humeral cartilages of E16.5 *Hbb*$^{F/F}$ (Ctrl), *Hbb*-H-cKO or *Hbb*-cKO mice. Scale bars, 50 μm. n = 3 biologically independent samples. **f, g** Histology examination (two upper panels) or TUNEL assay (two bottom panels) of proximal humeral cartilages of P1 *Hbb*$^{F/F}$ (Ctrl), *Hbb*-H-cKO or *Hbb*-cKO mice. Quantification of all biological replicates is provided (**g**). Arrows indicated dead chondrocytes. Green fluorescence marked the dead cells. Scale bars, 50 μm. n = 3 biologically independent samples. **h, i**, Histological images of distal humerus from P5 *Hbb*-cKO mice showed massive chondrocytes death in the center of cartilaginous tissues upon conditional knockout of *Hbb* gene. Quantification of all biological replicates is provided (**i**). Arrows indicated dead chondrocytes. Scale bars, 50 μm. n = 3 biologically independent samples. **j, k** TUNEL assay of *Hbb*$^{F/F}$ (Ctrl), *Hbb*-H-cKO, and *Hbb*-cKO proximal humeral cartilages of P5 mice. Arrows indicated dead chondrocytes. Green fluorescence marked the dead cells. Quantification of all biological replicates is provided (**k**). Scale bars, 50 μm. n = 3 biologically independent samples. Error bars represent SEM. *P* values were calculated using one-way ANOVA tests (**a-c, g, i, k**). The exact *P*-values of comparison are presented in the figures, respectively.

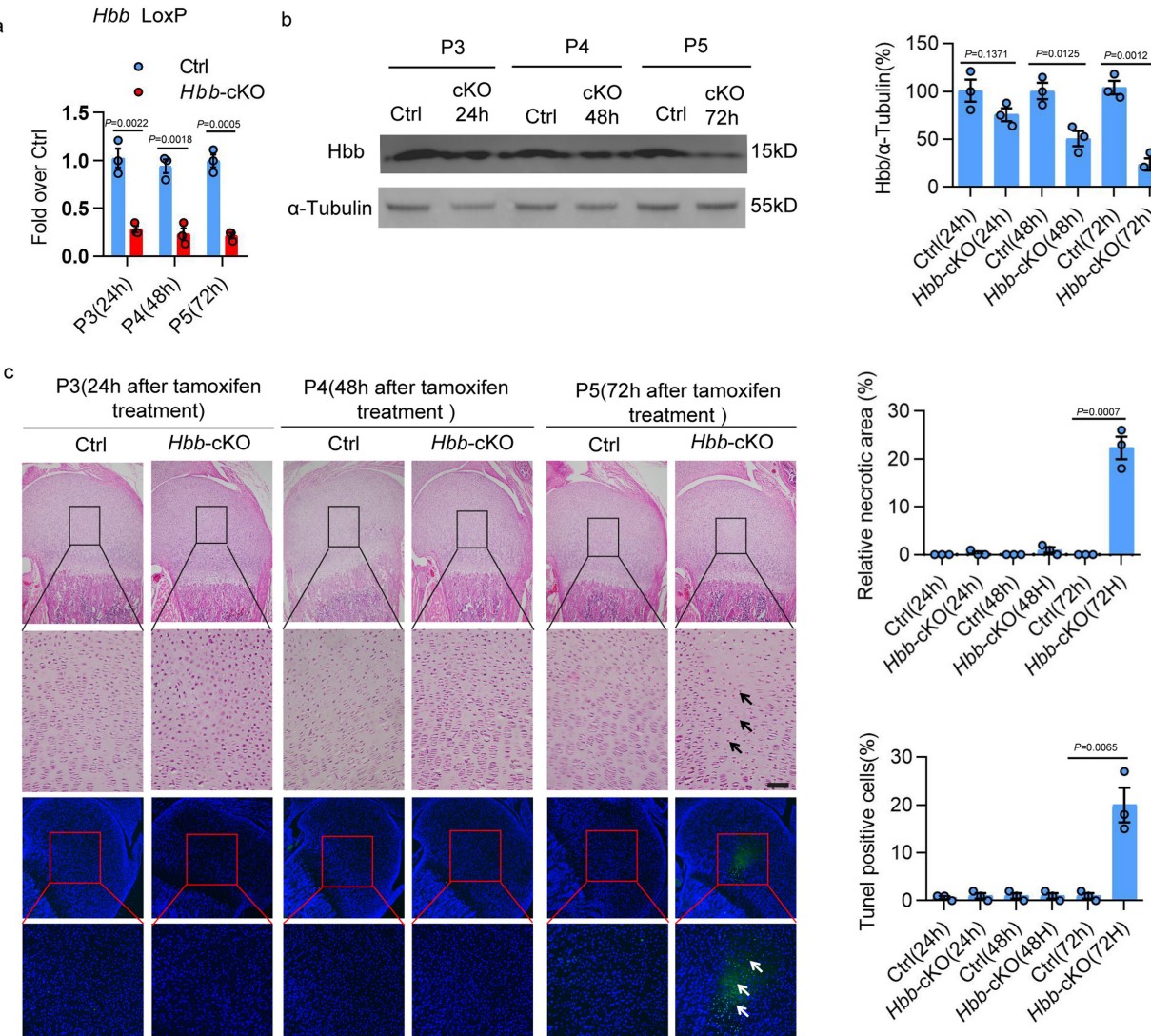

**Extended Data Fig. 10 | Phenotypes of the growth plate from mice with *Hbb* conditional knockout by *Col2a1-Cre*. a**, *Hbb*-LoxP qPCR of genomic DNA extracted from *Hbb*^F/F (Ctrl) or *Hbb*^F/F/*Col2a1-Cre*^ERT2 (*Hbb*-cKO) growth plates of P3, P4 or P5 mice, which were treated by tamoxifen (100 mg/kg) for once at P2. Data were normalized to β2-microglobulin (n = 3 biologically independent samples). **b**, Quantification of Hbb protein by Western blot analysis of total protein lysate extracted from *Hbb*^F/F (Ctrl) or *Hbb*^F/F/*Col2a1-Cre*^ERT2 (*Hbb*-cKO) growth plates of P3, P4 or P5 mice, which were treated by tamoxifen (100 mg/kg) for once at P2. A representative Western blot is shown on the left, and quantification of all biological replicates is provided on the right. Data were normalized to α-tubulin (n = 3 biologically independent samples). For gel source data, see Supplementary Fig. 1r. **c**, Histology examination (two upper panels) or TUNEL assay (two bottom panels) of proximal humeral cartilages of *Hbb*^F/F (Ctrl) or *Hbb*^F/F/*Col2a1-Cre*^ERT2 (*Hbb*-cKO) growth plates of P3, P4 or P5 mice, which were treated by tamoxifen (100 mg/kg) for once at P2. Quantification of all biological replicates is provided on the right. Arrows indicated dead chondrocytes. Green fluorescence marked the dead cells. Scale bars, 50 μm. n = 3 biologically independent samples. Error bars represent SEM. *P* values were calculated using two-tailed Student's *t*-test (**a-c**). The exact *P*-values of comparison are presented in the figures, respectively.

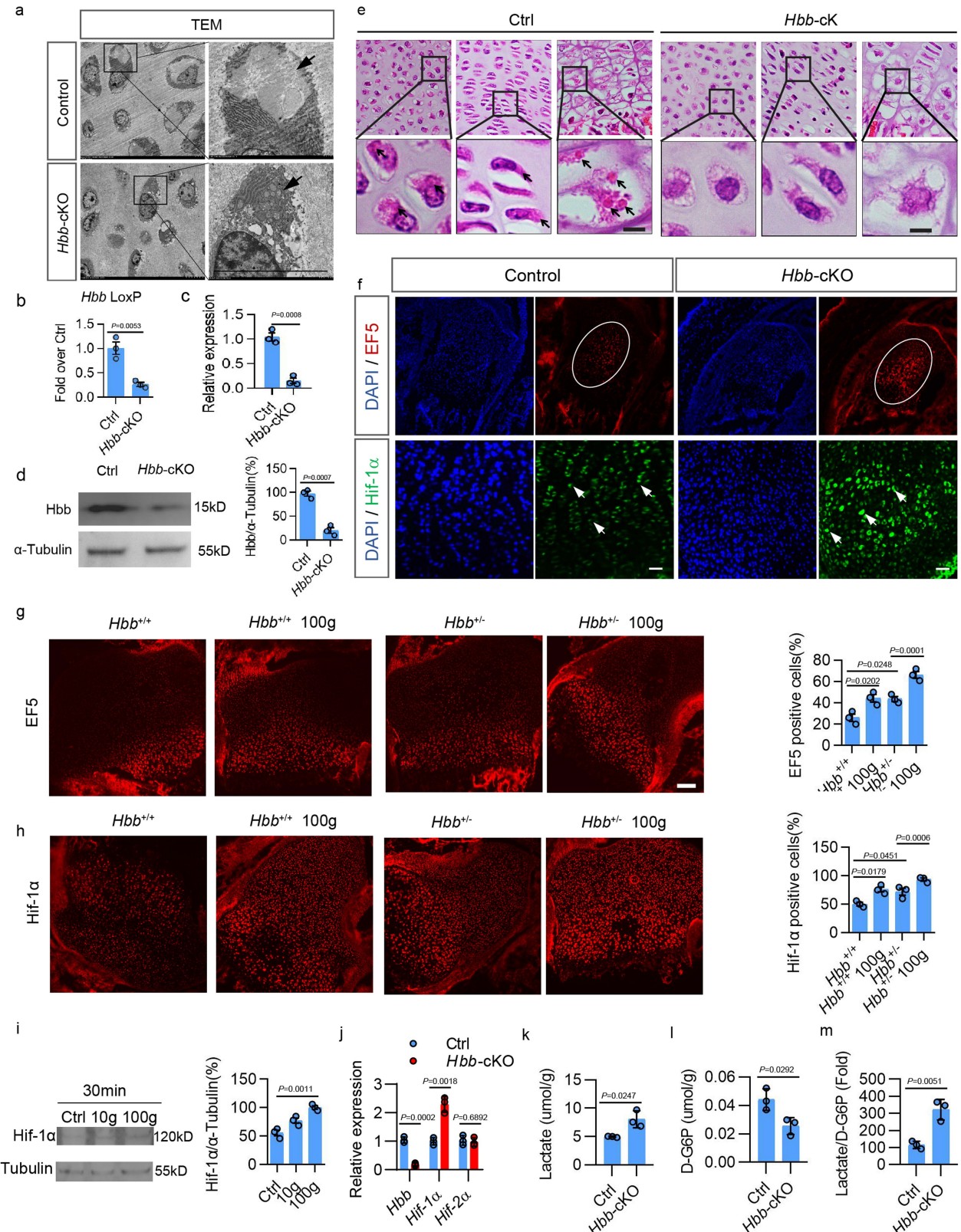

**Extended Data Fig. 11** | See next page for caption.

**Extended Data Fig. 11 | Impacts of *Hbb* conditional knockout on Hedy, hypoxia response and metabolism in chondrocytes of growth plate.**
**a**, Compromised formation of hemoglobin body in chondrocytes upon conditional *Hbb* knockout (*Hbb*-cKO) as indicated by TEM. Scale bar, 5 μm. **b**, *Hbb*-LoxP qPCR of genomic DNA extracted from *Hbb*^F/F (Ctrl) or *Hbb*^F/F/*Col2a1-Cre*^ERT2 (*Hbb*-cKO) growth plates of P3 mice, which were treated by tamoxifen (50 mg/kg) at E18.5. Data were normalized to β2-microglobulin (n = 3 biologically independent samples). **c**, Quantification of *Hbb* mRNA by qRT-PCR of total RNA extracted from *Hbb*^F/F (Ctrl) or *Hbb*^F/F/*Col2a1-Cre*^ERT2 (*Hbb*-cKO) growth plates of P3 mice, which were treated by tamoxifen (50 mg/kg) at E18.5. Data were normalized to *Gapdh* (n = 3 biologically independent samples). **d**, Quantification of Hbb protein by Western blot analysis of total protein lysate extracted from *Hbb*^F/F (Ctrl) or *Hbb*^F/F/*Col2a1-Cre*^ERT2 (*Hbb*-cKO) growth plates of P3 mice, which were treated by tamoxifen (50 mg/kg) at E18.5. A representative Western blot is shown on the left, and quantification of all biological replicates is provided on the right. Data were normalized to α-tubulin (n = 3 biologically independent samples). For gel source data, see Supplementary Fig. 1s. **e**, Impaired Hedy formation in *Hbb*-cKO cartilage of P3 mice, which were treated by tamoxifen (50 mg/kg) at E18.5. Black arrows indicated Hedies. Scale bars: 10 μm. n = 6 biologically independent samples. **f**, Representative images for EF5 (in red, up panel) and Hif-1α (in green, bottom panel) staining of cartilages from WT and *Hbb*-cKO mice. DAPI in blue for nuclei. Scale bars: 50 μm. **g**, Representative images for EF5 staining of newborn growth plates with the indicated genotypes, which were centrifuge at 0 or 100 g for 30 minutes at room temperature. Quantification of all biological replicates is provided on the right. Scale bars: 50 μm. The data are mean with SEM of triplicate experiments. **h**, Representative images for Hif-1α staining of cartilages with the indicated genotypes, which were centrifuge at 0 or 100 g for 30 minutes at room temperature. Quantification of all biological replicates is provided on the right. Scale bars: 50 μm. The data are mean with SEM of triplicate experiments. **i**, Quantification of Hif-1α protein by Western blot analysis of total protein lysate extracted from newborn growth plates, which were centrifuge at 0, 10 or 100 g for 30 minutes at room temperature. A representative Western blot is shown on the left, and quantification of all biological replicates is provided on the right. Data were normalized to α-tubulin (n = 3 biologically independent experiments). For gel source data, see Supplementary Fig. 1t. **j**, Expression of *Hif-1α* and *Hif-2α* in cartilages from *Hbb*^F/F (Ctrl) and *Hbb*-cKO mice. n = 3 biologically independent experiments. **k-m**, The metabolomic assay by mass spectrometry showed that the *Hbb*-cKO cartilages contained higher level of intracellular lactate (**k**) and lower level of intracellular glucose (**l**), leading to a significantly increased ratio of lactate over glucose (**m**). n = 3 biologically independent experiments. Error bars represent SEM. *P* values were calculated using two-tailed Student's *t*-test (**b-d**, **j-m**) or one-way ANOVA tests (**g-i**). The exact *P*-values of comparison are presented in the figures, respectively.

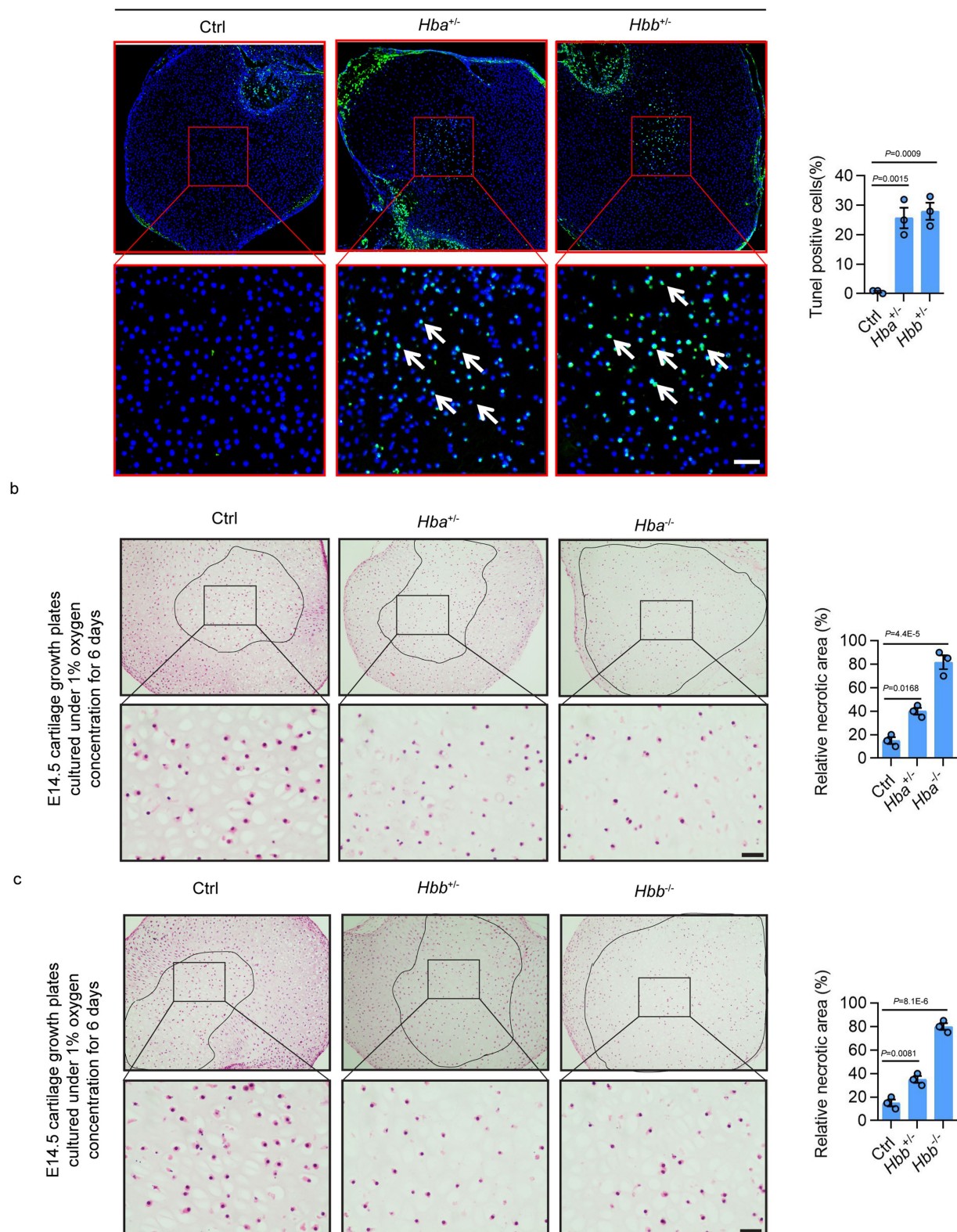

**Extended Data Fig. 12 | Hypoxia tolerance of cartilage with hemoglobin depletion. a**, TUNEL assay of cartilages of E18.5 mice cultured in vitro under hypoxia for 3 days. Quantification of all biological replicates is provided on the right. Scale bars: 50 μm. The data are mean with SEM of triplicate experiments. **b**, **c**, The humeral cartilage growth plates of E14.5 mice of the indicated genotypes were cultured in hypoxic environment (1% oxygen concentration) for 6 days, significantly more chondrocytes death occurred in the growth plates of *Hba* (**b**) or *Hbb* (**c**) homozygous knockout cartilages. The dead cell areas were marked with solid black lines. Scale bar: 50 μm. n = 3 biologically independent experiments. Error bars represent SEM. *P* values were calculated using one-way ANOVA tests. The exact *P*-values of comparison are presented in the figures, respectively.

# Reporting Summary

## Statistics

For all statistical analyses, confirm that the following items are present in the figure legend, table legend, main text, or Methods section.

| n/a | Confirmed | |
|---|---|---|
| ☐ | ☒ | The exact sample size (*n*) for each experimental group/condition, given as a discrete number and unit of measurement |
| ☐ | ☒ | A statement on whether measurements were taken from distinct samples or whether the same sample was measured repeatedly |
| ☐ | ☒ | The statistical test(s) used AND whether they are one- or two-sided *Only common tests should be described solely by name; describe more complex techniques in the Methods section.* |
| ☒ | ☐ | A description of all covariates tested |
| ☒ | ☐ | A description of any assumptions or corrections, such as tests of normality and adjustment for multiple comparisons |
| ☐ | ☒ | A full description of the statistical parameters including central tendency (e.g. means) or other basic estimates (e.g. regression coefficient) AND variation (e.g. standard deviation) or associated estimates of uncertainty (e.g. confidence intervals) |
| ☐ | ☒ | For null hypothesis testing, the test statistic (e.g. *F*, *t*, *r*) with confidence intervals, effect sizes, degrees of freedom and *P* value noted *Give P values as exact values whenever suitable.* |
| ☒ | ☐ | For Bayesian analysis, information on the choice of priors and Markov chain Monte Carlo settings |
| ☒ | ☐ | For hierarchical and complex designs, identification of the appropriate level for tests and full reporting of outcomes |
| ☒ | ☐ | Estimates of effect sizes (e.g. Cohen's *d*, Pearson's *r*), indicating how they were calculated |

*Our web collection on statistics for biologists contains articles on many of the points above.*

## Software and code

Policy information about availability of computer code

Data collection    Western blot images were acquired and analyzed via the BioRad Image Lab system. Electron micrographs were captured by Gatan digital camera (832 SC1000, Gatan, Warrendale, PA, USA) and its application software (Gatan Digital Micrograph software v3.0 ) . All data files of LC-MS/MS were created using Bioworks Browser rev.3.1 (Thermo Electron, San Jose, CA.). The acquired MS/MS spectra were searched against the concatenated target/reverse Glycine max database using the SEQUEST search engine (Proteome Discoverer Software v2.3.0.523). Cells timelapse imaging were taken with the Ultraview Vox confocal system (Perkin Elmer, Volocity v6.3.0). Oxygenation-dissociation analyser and its application software (BLOODOX-2018 Analyser, Softron Biotechnology, Beijing, China)was used to determinate oxygen dissociation curve. Red blood cells (RBCs) counts and hemoglobin concentration (Hb) were performed by automatic blood analyzer  (Sysmex Corporation, Japan, XP-100) and its application software (XT2000i1800i IPU). FRAP assay was conducted using the FRAP module of the Nikon confocal microscopy system (NIS Elements AR v4.50.00).  Flow cytometry was performed with aflow cytometer (Coulter-XL, USA), EXPO32 ADC Software was used for data collection and analysis. The serial sections for correlative light electron microscopy were finally automatically acquired by a Helios Nanolab 600i dual-beam SEM (Scanning Electron Microscope, Thermo Fisher, USA) with an automated imaging software(Auto SEE 1.58) and Fluorescence images were collected by a confocal microscope (Zeiss LSM 980) and  processed by accompanying software (ZEN v3.6095.01). Fluorescence quantitative PCR reaction was performed by Real time fluorescence quantitative PCR instrument (QTOWER3G, Jena Bioscience) and its application software (qPCRsoft v3.4).

| Data analysis | Statistical analyses were performed using GraphPad Prism 8.0.1 or SPSS 18.0. Western blot quantification was performed using NIH Image J 1.8.0.112. Real-Time PCR results were analyzed by Microsoft Excel (2306 Build 16.0.16529.20164). The fluorescence intensity was quantified with Nikon confocal microscopy system (NIS Elements AR 4.50.00). RNA-seq data was analyzed and visualized using Fastp (version 0.14.1), Microsoft Excel (2306 Build 16.0.16529.20164), Hisat2 (version 2.2.0), circos (version 2.5.0), Stringtie (version 2.1.7), and edgeR (version 4.2.1). |
|---|---|

For manuscripts utilizing custom algorithms or software that are central to the research but not yet described in published literature, software must be made available to editors and reviewers. We strongly encourage code deposition in a community repository (e.g. GitHub). See the Nature Portfolio guidelines for submitting code & software for further information.

# Data

Policy information about availability of data

All manuscripts must include a data availability statement. This statement should provide the following information, where applicable:

- Accession codes, unique identifiers, or web links for publicly available datasets
- A description of any restrictions on data availability
- For clinical datasets or third party data, please ensure that the statement adheres to our policy

RNA-Seg data of the present study havebeen deposited in NCBI Gene ExpressionOmnibus (GEO) with accession codes GSE182640. Bioinformatics analysis of H3K4me3 modification region was performed at Cistrome Data Browser (http://cistrome.org/db/#/). Histone methylation data from work done by Ohba S et al (DOI: 10.1016/j.celrep.2015.06.013). Data of microdissection sample mass spectrometry, SDS-PAGE samplemass spectrometry, oligonucleotides and construct information have been deposited in supplementary tables.

# Research involving human participants, their data, or biological material

Policy information about studies with human participants or human data. See also policy information about sex, gender (identity/presentation), and sexual orientation and race, ethnicity and racism.

| Reporting on sex and gender | We report  the patient whom cartilage was displayed in the manuscript. We did not do a analysis of sex-specific difference, due to sample size is rather limited. |
|---|---|
| Reporting on race, ethnicity, or other socially relevant groupings | We did not do a analysis of variable on groupings, due to sample size is rather limited. Human sample was not the main research in our manuscript. In addition,We did not find significant variable in animal experiments. |
| Population characteristics | Our study did not involve the population characteristics. |
| Recruitment | Human healthy articular cartilage specimens from patients with acute trauma were collected from the Department of Pathology of Xijing Hospital. |
| Ethics oversight | Human articular cartilage specimens were collected with an informed consent and approval of the project by the Research Ethics Board of the Xijing Hospital. |

Note that full information on the approval of the study protocol must also be provided in the manuscript.

# Field-specific reporting

Please select the one below that is the best fit for your research. If you are not sure, read the appropriate sections before making your selection.

☒ Life sciences        ☐ Behavioural & social sciences        ☐ Ecological, evolutionary & environmental sciences

For a reference copy of the document with all sections, see nature.com/documents/nr-reporting-summary-flat.pdf

# Life sciences study design

All studies must disclose on these points even when the disclosure is negative.

| Sample size | No statistical methods were used to predetermine sample size, which were chosen based on previous experiments and comparable studies in literature (PMID: 30651640, PMID: 32807933, PMID: 34163069 ). For key experiments, several test runs were performed to determine the suitable sample size. The sample sizes we chose are sufficient and suitable based on the following ingredients: 1.  Effect size: the great difference in outcome measure between control and experimental groups (more than two-fold). 2. Variability: no considerable overlap in outcome measure between control and experimental groups. 3. Significance level : the statistical test Student's t-test and ANOVA with post-test used finds that the probability of the result is even lower than the chance of a type 1 error), indicating that the hypothesis is true. Sample size for each experiment is indicated in the legend. |
|---|---|
| Data exclusions | No exclusions other than caused by missing information have been made. |
| Replication | At least three biological replicates were performed for all in vivo experiments, most in vitro experiments were repeated at least three times. For non-quantitative data (micrographs, western blots and so on), findings were reproduced at least three times by replicating the |

experiments and/or cross-validating with orthogonal approaches, and representative results are shown. Details of each exact number of replicates are provided in the figure legends.

Randomization
No statistical methods were used for randomization.
For in vitro experiments, cells were isolated from randomly chosen wild-type or knockout mice.
For in vivo experiments, mice were randomly allocated into experimental groups.

Blinding
Blinding was widely used in the study. Data collection and analysis, such as immunostaining, qRT-PCR, and Western blot were frequently performed by participants other than the experiment designer. During these data collection and analysis steps, all participants were routinely blinded to group allocation.

# Reporting for specific materials, systems and methods

We require information from authors about some types of materials, experimental systems and methods used in many studies. Here, indicate whether each material, system or method listed is relevant to your study. If you are not sure if a list item applies to your research, read the appropriate section before selecting a response.

## Materials & experimental systems

| n/a | Involved in the study |
|-----|----------------------|
| ☐ | ☒ Antibodies |
| ☐ | ☒ Eukaryotic cell lines |
| ☒ | ☐ Palaeontology and archaeology |
| ☐ | ☒ Animals and other organisms |
| ☒ | ☐ Clinical data |
| ☒ | ☐ Dual use research of concern |
| ☒ | ☐ Plants |

## Methods

| n/a | Involved in the study |
|-----|----------------------|
| ☒ | ☐ ChIP-seq |
| ☐ | ☒ Flow cytometry |
| ☒ | ☐ MRI-based neuroimaging |

## Antibodies

Antibodies used

Primary antibodies:
Epitope, Host Species , Concentration used, Manufacturer, Clone/Polyclonal, Catalog Number
Hif1a, Rabbit, Western Blotting 1:1000/Immunofluorescence 1:400, CST, D1S7W, 36169
Hbb, Rabbit, Western Blotting 1:1000/Immunofluorescence 1:400, Invitrogen, Polyclonal, PA5-60287
GFP, Rabbit, Western Blotting 1:1000, CST, Polyclonal, 2555
GFP, Rabbit, Electron Microscopy 1:100, Abcam, Polyclonal, ab290
Hba, Rabbit, Western Blotting 1:1000/Immunofluorescence 1:200, Invitrogen, SN70-09, MA5-32328
Hif2α/Epas1, Rabbit, Rabbit, Western Blotting 1:1000, Novus, polyconal, NB100-12
Klf1/Eklf, Rabbit, Western Blotting 1:1000, Abnova, ployclonal, PAB5895
Klf1/Eklf, Mouse, ChIP-qPCR 5ug/ml, Active motif, 7B2, 61233
AMPKα1, Rabbit, Western Blotting 1:500, ABclonal, polyclonal, A1229
pAMPK,Rabbit, Western Blotting 1:500, ABclonal, ARC1547, AP1002
Caspase 3, Rabbit, Western Blotting 1:500/Immunofluorescence 1:200, ABclonal, ARC0133, A19654
Kdm5a, Rabbit, Western Blotting 1:500, ABclonal, ARC1120, A4755
Kdm5b, Rabbit, Western Blotting 1:500, ABclonal, polyclonal, A7772
H3K4me3, Rabbit, Chip-qPCR 5ug/ml, ABclonal, polyclonal, A2375
H3, Rabbit, chip-qPCR 5ug/ml, Proteintech, polyclonal, 17168-1-AP
Nonspecific IgG, rabbit, Chip-qPCR 5ug/ml, Proteintech, polyclonal, 30000-0-AP
α-Tubulin, rabbit, Western Blotting 1:3000, CST, p11H10, 2125

secondary antibodies
Epitope, Host Species , Concentration used, Manufacturer, Clone/Polyclonal, Catalog Number
Anti-Rabbit IgG (whole molecule)–Peroxidase antibody produced in goat, goat, 1:10000, Millipore, polyclonal, A0545
Anti-Mouse IgG (Fab specific) antibody produced in goat, goat, 1:10000, Millipore, polyclonal, M4155
Horseradish peroxidase enzyme (HRP)-coupled goat anti-rabbit secondary antibody, goat, 1:500, Abcam, polyclonal, ab7090

Validation

All antibodies were obtained from indicated commercial vendors with ensured quality. In addition, all the antibodies have been used in multiple experiments to detect intended proteins in control samples with expected molecular weight to validate their effectiveness in our study. Antibodies target for GFP were validated Specific affinity for EGFP in multiple literatures.
Primary antibodies:
 Hif-1α, Reactivity: H, M, Mk. Application: WB IP,IF, F, ChIP, C&R. (https://www.cellsignal.cn/products/primary-antibodies/hif-1a-d1s7w-xp-rabbit-mab/36169?site-search-type=Products&N=4294956287&Ntt=36169&fromPage=plp&_requestid=5968605)
 Hbb, Reactivity: H, highest antigen sequence identityto the following orthologs: Mouse-96%, Rat-96%. Application:
IHC ; IICC/IF.(https://www.thermofisher.cn/cn/zh/antibody/product/HBB-Antibody-Polyclonal/PA5-60287)
 GFP, Reactivity: species independent. Application: WB; IHC. (https://www.cellsignal.cn/products/primary-antibodies/gfp-antibody/2555?site-search-type=Products&N=4294956287&Ntt=2555&fromPage=plp&_requestid=5969550)
 GFP, Reactivity: species independent. Application: WB; IHC; Electron Microscopy. (https://www.abcam.cn/products/primary-antibodies/gfp-antibody-ab290.html)

Hba Reactivity: H, M, Rat. Application: Western Blot (WB) ; IHC. (https://www.thermofisher.cn/cn/zh/antibody/product/Hemoglobin-alpha-Antibody-clone-SN70-09-Recombinant-Monoclonal/MA5-32328)
Hif-2a/Epas1, Reactivity: H, M, Rat, Fi, Ha, Pm, Rb, Re, Sh. Application: Chip; IHC; ELISA ; WB. (https://www.novusbio.com/products/hif-2-alpha-epas1-antibody_nb100-122)
Klf1/Eklf, Reactivity: M. Application: WB; .(https://www.abnova.com/products/products_detail.asp?catalog_id=PAB5895)
Klf1/Eklf, Reactivity: M. Application: chip-qPCR; (https://www.activemotif.com.cn/catalog/details/61233/eklf-antibody-mab-clone-7b2)
AMPKα1, Reactivity: H, M, Rat. Application: WB ; IF/ICC. (https://abclonal.com.cn/catalog/A1229)
p-AMPK, Reactivity: H, M. Application: WB. (https://abclonal.com.cn/catalog/AP1002)
Caspase3, Reactivity: H, M, Rat. Application: WB; IF/ICC. (https://abclonal.com.cn/catalog/A19654)
Kdm5a, Reactivity: H, M, Rat. Application: WB; IP; ChIP. (https://abclonal.com.cn/catalog/A4755)
Kdm5b, Reactivity: H, M. Application: WB; IF/ICC. (https://abclonal.com.cn/catalog/A7772)
H3K4me3, Reactivity: H, M, Rat. Application: WB; IF/ICC. (https://abclonal.com.cn/catalog/A2375)
H3, Rabbit, Reactivity: H, M, Rat. Application:  FC, IF, IHC, IP, WB, ELISA. (https://www.ptgcn.com/products/Histone-H3-Antibody-17168-1-AP.htm)
Nonspecific IgG, Reactivity: H, M, Rat. Application:  FC, IF, IHC, IP, WB, ELISA. (https://www.ptgcn.com/products/IgG-control-Antibody-30000-0-AP.htm)
α-Tubulin, Reactivity: H, M, Rat, Mk, Dm, Z, B, Pg. Application:  IF, IHC, IP, WB. (https://www.cellsignal.cn/products/primary-antibodies/a-tubulin-11h10-rabbit-mab/2125?site-search-type=Products&N=4294956287&Ntt=2125&fromPage=plp&_requestid=5981232)
secondary antibodies
Anti-Rabbit IgG (whole molecule)–Peroxidase antibody produced in goat, Reactivity: R. Application:  IF, IHC, WB. (https://www.sigmaaldrich.cn/CN/en/product/sigma/a0545)
Anti-Mouse IgG (Fab specific) antibody produced in goat, Reactivity: M. , Application:  IF, IHC, WB. (https://www.sigmaaldrich.cn/CN/en/product/sigma/m4155)
Horseradish peroxidase enzyme (HRP)-coupled goat anti-rabbit secondary antibody, Reactivity: R. Application:  ICC/IF, IHC-P, WB, ELISA, Immunomicroscopy, Dot blot, IHC-Fr. (https://www.abcam.cn/products/secondary-antibodies/goat-rabbit-igg-hl-hrp-preadsorbed-ab7090.html)

# Eukaryotic cell lines

Policy information about cell lines and Sex and Gender in Research

| Cell line source(s) | 293T, HepG2, HCT116, PLC/PR/F5, PC12 cell lines were purchased from Procell Life Science&Technology Co.,Ltd. ATDC5 cell line was purchased from NingboMingZhoubioCO.,Ltd, which was derived from ECACC. |
|---|---|
| Authentication | All cell lines were authenticated by STR profiling. |
| Mycoplasma contamination | All cell lines were mycoplasma-free after PCR verification. |
| Commonly misidentified lines (See ICLAC register) | There is no  commonly misidentified cell line were used in the study. |

# Animals and other research organisms

Policy information about studies involving animals; ARRIVE guidelines recommended for reporting animal research, and Sex and Gender in Research

| Laboratory animals | Mice heterozygous for the alpha-globin null allele (Hba+/-, both of the adult hemoglobin genes, alpha 1 and alpha 2, and the region between them were deleted and replaced with a neomycin resistance cassette by homologous recombination) or for the beta-globin null allele (Hbb+/-, a genomic fragment encompassing all of Hbb-b1 and a 5' portion of Hbb-b2 was replaced with a neo cassette inserted by homologous recombination.) were produced by crossing mice of Hbatm1Paz Hbbtm1Tow Tg (HBA-HBBs) 41Paz/J (Jackson Labs, No:003342)46 with wildtype C57BL/6J mice. The Hbatm1Paz Hbbtm1Tow Tg (HBA-HBBs) 41Paz/J mice are called sickle cell mice (Berkeley model), which are homozygous for the both alpha-globin and beta-globin null allele and carrying the human sickle transgene (Hba0/0 Hbb0/0Tg (Hu-miniLCRα1GγAγδβS). Mice homozygous for Hba or Hbb knock-out mutation die in utero for severe anemia. Hif-1aF/F (No.007561)47, Col2a1-CreERT2 (No.006774)48, Prx1-Cre (No.005584)49 mice were from Jackson Labs. Hif-2αF/F (No.NM-CKO-200163) mice were from Shanghai Model Organisms Center, Inc. To specifically knockout Hif-1α or/and Hif-2α gene in chondrocytes, Hif-1αF/F or/and Hif-2αF/F mice were bred to Col2a1-CreERT2 mice. To conditionally delete Hbb in mesenchymal cells or chondrocytes, HbbF/F mice were bred to Prx1-Cre or Col2a1-CreERT2 mice. To investigate the role of Klf1 in chondrocytes, we generated conditional Klf1-floxed mice with C57BL/6J-background by homologous recombination. A targeting vector was designed to replace the 2nd exon of Klf1 (Sup. Fig. 3a). To conditionally delete Klf1 in chondrocytes, Klf1F/F mice were bred to Col2a1-CreERT2.

Studies were conducted on E14.5 to 12 month old mus musculus of the C57BL/6 Strain.

Mice were housed in a room on a 12:12 light:dark cycle, a 72+/-2 F temperature setting and humidity of 30-70%. |
|---|---|
| Wild animals | There is no wild animal in our study. |
| Reporting on sex | Sex-based analyses were not performed, due to no obvious sex and gender difference was observed in our study. |

| Field-collected samples | There is no field collected samples were used in the study. |
| Ethics oversight | Animal studies were approved by the Institutional Animal Care and Use Committee at the Fourth Military Medical University. |

Note that full information on the approval of the study protocol must also be provided in the manuscript.

# Flow Cytometry

## Plots

Confirm that:

☒ The axis labels state the marker and fluorochrome used (e.g. CD4-FITC).

☒ The axis scales are clearly visible. Include numbers along axes only for bottom left plot of group (a 'group' is an analysis of identical markers).

☒ All plots are contour plots with outliers or pseudocolor plots.

☒ A numerical value for number of cells or percentage (with statistics) is provided.

## Methodology

| Sample preparation | Analysis of total ROS: Primary chondrocytes were used to guantify total ROS accumulation with a ROS assav kit (Cayman Chemicals) according to the manufacturer's instructions. After a brief trypsinization with Trypsin-EDTA (0.05%) at 37℃,cells were incubated in dark with 5mM dihydroethidium (DHE) probe for 30 minutes at 37℃.Thereafter,cells were washed twice with ROS staining buffer.<br>Analysis of mitochondrial ROS : According to the manufacturer's instructions, the mitochondrial ROS accumulation was performed with a MitoSOXassay kit (Invitrogen). Briefly, primary chondrocytes isolated and cultured were incubated with 5mM MitoSOX probe for 20 minutes at 37℃ in dark.Aftera brief trypsinization with Trypsin-EDTA (0.05%) at 37℃,cells were washed twice with FACS buffer (1x DPBS +5% FBS). |
| Instrument | Flow cytometry was performed with a flow cytometer(Coulter-XL, USA). |
| Software | EXPO32 ADC Software was used for analysis. |
| Cell population abundance | Flow sorting was not performed. |
| Gating strategy | Cells were gated by FSC-A x SSC-A to exclude debris and then by FSC-H x FSC-W following SSC-H x SSC-W to exclude cell doublets.<br>1. Set a gate based on the negative control (with a confidence interval of about 2% for the negative control) 2. When the fluorescence signal of the test sample is greater than the negative control, it is defined as positive. |

☒ Tick this box to confirm that a figure exemplifying the gating strategy is provided in the Supplementary Information.

