## [Peer Review File · Nature]

Manuscript Title: An Extra-Erythrocyte Role of Hemoglobin Body in Chondrocyte Hypoxia Adaption

Reviewer Comments & Author Rebuttals

Reviewer Reports on the Initial Version:

Referees' comments:

Referee #1 (Remarks to the Author):

In this study, Zhang and colleagues demonstrate that growth plate chondrocytes accumulate hemoglobin in their cytoplasm, and this is critical for their survival as loss of chondrocytic hemoglobin leads to a severe increase of intracellular hypoxia and massive cell death. Moreover, the authors show that accumulation of hemoglobin in chondrocytes does not depend on the HIFs but it is regulated by KLF1.

The findings are novel and exciting. The notion that hypoxic chondrocytes need a reservoir of intracellular hemoglobin to survive hypoxia is fascinating and unveils a novel biological mechanism hypoxic cells adopt to overcome hypoxia.

A series of issues need to be addressed to strengthen the authors' conclusions and their biological relevance:

1. In Figures 1 and 2, the authors provide a series of convincing evidence that growth plate chondrocytes accumulate hemoglobin in their cytoplasm across different time points, different bones, and in both mice and humans. In Figure 3, they show that exposure to hypoxia increases the accumulation of chondrocytic hemoglobin, but this occurs in a HIF-independent manner. Curiously, either genetic loss of HIF1a or pharmacological inhibition (GN44208) augments expression of Hba and Hbb (Figure 3, panels d-e); do the authors have an explanation for those findings? Along those lines, differently from what stated by the authors, Roxadustat increases both HIF1a and HIF2a activity; why the effects of Roxadustat treatment are different from IOX2? The authors should carefully document levels of HIF1 and HIF2a proteins and key downstream targets in all their genetic and pharmacological manipulation. Minor point: Figure 3 panel d is inappropriately labelled as "articular cartilage".
2. In addition, they should prove that primary chondrocytes cultured in hypoxia and lacking HIF1a still display an increase of Hba and Hbb expression.
3. They should prove that primary chondrocytes cultured in hypoxia and lacking HIF1a still display an increase in KLF1 expression.
4. Furthermore, the experiment shown in Figure 3 panel g should be repeated in hypoxia.
5. Chondrocyte death in Figure 4 panels a-b should be proven by TUNEL and appropriately quantified.
6. Is the cell death secondary to loss of hemoglobin caspase-dependent?
7. More in general, growth plate development should be systematically studied in all the mouse

models presented in the paper across different time points, and efficient recombination of the gene of interest should be carefully documented in each of those mouse models.

8. How do the authors explain the significant difference in P50 between red blood cells and chondrocytes (Figure 4 panel g)?

9. At the end of the story, the reader is left with the issue of why loss of chondrocytic hemoglobin causes cell death. The authors imply that this is due to the increased hypoxia, which is likely. But, then why increased intracellular hypoxia does cause cell death? Along those lines, it is critical the authors measure ATP/ADP, pAMK/AMP, and mitochondrial ROS.

10. Another possibility is that, paradoxically, prolonged activation of the HIFs due to augmented hypoxia increases cell death. To exclude this possibility, it would be helpful to use a temporally induced conditional knockout of Hba and show presence of cell death within 48 hrs/72 hrs from the induction.

11. The TEXT has numerous typos and grammar issues that need to be corrected.

Referee #2 (Remarks to the Author):

A. Summary: Cartilage is poorly vascularized and as a result is a highly hypoxic microenvironment. The authors report the presence of cytosolic condensates of hemoglobin in chondrocytes from various murine and human sources. They provide evidence that the same globin switching that occurs in erythroid cells during development also occurs in chondrocytes. Finally, they demonstrated that globin gene expression was induced by exposure of chondrocytes to hypoxia, which they claimed was driven by hypoxia-induced expression of Klf1, a known transactivator of globin gene expression in erythroid cells.

B. Novelty. Although globin gene expression has been observed in many cell types, there was no good evidence that functional hemoglobin was produced in cells other than erythrocytes. The authors propose that the hemoglobin condensate represents an oxygen storage depot in chondrocytes.

C. Most of the claims are well supported by the data but there are several key issues that need to be addressed.

1. Erythroid hemoglobin binds O₂ in the lungs, where the PO₂ is high and releases O₂ in peripheral tissues where PO₂ is low. In contrast, the hemoglobin condensates in chondrocytes are static: they remain in the same location with the same low PO₂ for their entire lifetimes. How then is O₂ picked up and released? Do O₂ levels vary in cartilage e.g. during exercise?

2. The authors claim that the hypoxic induction of Klf1 and its subsequent activation of globin gene transcription occurs in a manner independent of hypoxia-inducible factors. How, then, is Klf1 expression induced? Klf1 gene expression has been shown to be upregulated in cells from patients who are homozygous for a missense mutation in the VHL gene that impairs the O₂-dependent degradation of HIF-1α and HIF-2α (PMID: 23993337), strongly suggesting that hypoxia-induced expression of Klf1 is HIF-dependent. The authors should knock down the expression of HIF-1α AND HIF-2α in primary chondrocytes and analyze the effect of culturing the cells under high (20%) and low (1%) O₂ conditions or in the presence of the HIF stabilizer dimethylxylglycine. The experiments should include analysis of known HIF target genes as positive controls.

3. The characterization of some of the mouse mutants was not adequate -- "pale" and "purplish" are

crude descriptions. Do these animals have normal circulating RBC counts and hemoglobin levels? Is there increased tissue hypoxia? If so, what is the mechanism for these systemic findings?

Referee #3 (Remarks to the Author):

I am an expert in biological phase separation and higher-order protein assembly, so I will only comment on the data claiming that Hemoglobin bodies form through phase separation.

In this paper, Zhang et al. make a provocative claim that Beta globin is sequestered into micron-scale condensates, called “Hemoglobin bodies”, inside mouse chondrocytes. They claim that Hemoglobin bodies assemble through hypoxia-induced phase separation of Beta-globin. If true, this finding could provide an explanation for how poorly vascularized tissues survive low-oxygen stress. However, the findings are extremely counterintuitive and contradict much of what we know about Hemoglobin structure. Much more work is required to understand the assembly mechanism of Hemoglobin bodies and rule out plausible alternative explanations. Thus, I do not recommend publication at this time.

Major Comments:

1. To me, the conclusion that Beta-globin can condense into micron-scale droplets is counterintuitive. Max Perutz won a Nobel prize in 1962 for solving the 3D structure of Hemoglobin; the structure of many hemoglobin subunits have been studied since, and they have confirmed the compact, globular structure described by Perutz. X-ray crystallography requires concentrating protein to extremely high levels, which is needed to form crystals. It also requires ordered packing of the molecules to make a crystal lattice. One cannot use X-ray crystallography to study phase-separating proteins because, at high concentrations, the proteins will form liquid droplets or hydrogels with too much disorder to create a useful crystal diffraction pattern. Thus, all evidence to date indicates that Hemoglobin is highly structured and does not form liquid droplets.

2. The authors' claim that mouse Beta-globin has an intrinsically disordered domain on its C-terminus is a dangerous and misleading over-interpretation. This unstructured region is 5 amino acids long (see <https://alphafold.ebi.ac.uk/entry/A8DUK4>). By that metric, almost every protein in the proteome has an intrinsically disordered tail region. Most well-described intrinsically disordered proteins (e.g., NPM1, FUS, P body proteins) have disordered regions containing 100's of amino acids. In my 10 years of studying protein condensation I have never seen a protein that could phase separate using a disordered tail of only 5 amino acids.

3. The authors need to rule out plausible alternative explanations for how Hemoglobin bodies form. It is possible that Hbb is binding to the surface of a neutral lipid droplet or some other large structure. Neutral lipid droplets are surrounded by phospholipid monolayers, and they can look exactly like the big droplets shown in figure 2h (for a reference, see figure 5C from this paper: <https://www.molbiolcell.org/doi/pdf/10.1091/mbc.E19-05-0284>). That could explain the large droplet-like morphologies and the results of the droplet fusion experiment in figure 2e. Turnover may be caused by binding and unbinding from the lipid droplet surface. The C-terminal tail of Hbb

may be a binding motif to allow this.

4. Suggested experiments to strengthen the manuscript:

- a. Reconstitute Hbb droplets in vitro using purified Hbb. This will be a definitive test of whether Hbb is sufficient to phase separate.
- b. Assess the lipid and nucleic acid content of the extracted Hemoglobin bodies. Simple stains are a good way to start (e.g., Nile Red or BODIPY for lipids, SYTO stains for RNA). Mass spectrometry-based lipidomics would be the next step.
- c. Figure 2k needs a control for expression levels. It is possible that mutations in the C-terminus of Hbb cause expression levels to drop, thus preventing accurate analysis of localization.
- d. Provide a control to show that the localization of Hbb is not an artifact of the GFP tag on the C-terminus. Repeat Figure 2c using immunofluorescence of cells expressing untagged Hbb.

Author Rebuttals to Initial Comments:
An Extra-erythrocyte Role of Hemoglobin body in Hypoxia Adaption of Chondrocytes

(Manuscript ID: **2021-11-18288**)

We thank the editor a lot for offering us the opportunity to revise the manuscript and the reviewers very much for their many excellent comments. Over the past year, great efforts were made to overcome all kinds of troubles caused by COVID-19 in reagent supply, animal breeding, and lab member quarantine, etc. to perform experiments essential to address all the reviewers' concerns. Now in the present version of the manuscript, we have made changes in line with the reviewers' suggestions and we hope that you find the manuscript has been significantly improved and suitable for publication in the journal of *Nature*. Please find below a point-by-point response to each of the reviewers' comments and suggestions.

Referees' comments:

Referee #1 (Remarks to the Author):

In this study, Zhang and colleagues demonstrate that growth plate chondrocytes accumulate hemoglobin in their cytoplasm, and this is critical for their survival as loss of chondrocytic hemoglobin leads to a severe increase of intracellular hypoxia and massive cell death. Moreover, the authors show that accumulation of hemoglobin in chondrocytes does not depend on the HIFs but it is regulated by KLF1.

The findings are novel and exciting. The notion that hypoxic chondrocytes need a reservoir of intracellular hemoglobin to survive hypoxia is fascinating and unveils a novel biological mechanism hypoxic cells adopt to overcome hypoxia.

Response: We thank the referee for the nice words.

A series of issues need to be addressed to strengthen the authors' conclusions and their biological relevance:

1. In Figures 1 and 2, the authors provide a series of convincing evidence that growth plate chondrocytes accumulate hemoglobin in their cytoplasm across different time points, different bones, and in both mice and humans. In Figure 3, they show that exposure to hypoxia increases the accumulation of chondrocytic hemoglobin, but this occurs in a HIF-independent manner. Curiously, either genetic loss of HIF1a or pharmacological inhibition (GN44208) augments expression of Hba and Hbb (Figure 3, panels d-e); do the authors have an explanation for those findings?

Response: We thank the referee for the insightful comment. Yes, we did find that activation of HIF pathway was not required for the upregulation of chondrocyte hemoglobin induced by hypoxia, which, however, does not rule out that HIF pathway may play a role in suppressing the expression of chondrocyte hemoglobin. In line with this notion, HIF-1 α was recently reported to be able to alleviate hypoxia by suppressing mitochondrial oxidative respiration in chondrocytes. Accordingly, knockdown or inhibition of HIF-1 α resulted in increased oxygen consumption and enhanced chondrocyte hypoxia [Yao *et al*, *Dev Cell*, 2019, PMID: 31105007], which induces the expression of *Hba* and *Hbb*. The question then was: what pathway other than the well-known HIF pathway may mediate hemoglobin upregulation by hypoxia? As proposed in the submitted manuscript, Klf1 turned out to be a critical mediator linked to hemoglobin expression as shown in Figure 3f-k. However, a gap remained between hypoxia and Klf1. During the revision process, we managed to make progress on this point. Inspired by the fact that hypoxia could induce HIF-independent histone methylation via the inhibition of KDM5a demethylase, an oxygen sensor [Batie *et al*, *Science*, 2019, PMID: 30872526], we validated that both hypoxia and depletion of KDM5a, but not KDM5b, led to increased H3K4me3 methylation of the Klf1 locus, resulting in the increased expression of *Klf1*, and subsequent upregulation of *Hba* and *Hbb* transcription. These results suggest that the inactivation of KDM5a by hypoxia mediates *Klf1* upregulation, which was included in the revised manuscript as Extended Data Fig. 9 and Extended Data Fig. 10.

- Along those lines, differently from what stated by the authors, Roxadustat increases both HIF1a and HIF2a activity; why the effects of Roxudastat treatment are different from IOX2? The authors should carefully document levels of HIF1 and HIF2a proteins and key downstream targets in all their genetic and pharmacological manipulation.

Response: We thank the referee for the insightful comment. As suggested, we carefully titrated the concentrations of compounds and examined their effects on the expression levels of *Hbb*, *Hba* and *Epo*. It turns out that Roxudastat, in a specified concentration, could produce effects similar to those by IOX2. The related results were included in the revised manuscript as Extended Data Fig. 7a-i. Moreover, we found that PT2385 and PT2399, in a defined range of concentrations, could also produce effects similar to those by *HIF* knockout in terms of *Hbb* and *Hba* expression. Also as suggested, we examined the expression of HIF1/2 α and their downstream targets where applicable. The related results were shown in Extended Data Fig. 6, 7 and Fig. 3d-e.

- Minor point: Figure 3 panel d is inappropriately labelled as “articular cartilage”.

Response: We apologize for the mistake, which was corrected to “*Hif-1α* conditional knockout chondrocytes from growth plate” in the revised Extended Data Fig. 6h.

2. In addition, they should prove that primary chondrocytes cultured in hypoxia and lacking HIF1a still display an increase of *Hba* and *Hbb* expression.

Response: As suggested, we examined the *Hif-1α*-cKO primary chondrocytes cultured in hypoxia, and did find an increase of *Hba* and *Hbb* expression, which was included as Extended Data Fig. 6i in the revised manuscript. Moreover, we also confirmed that knocking out *Hif-1α* or *Hif-2α*, either alone or in combination, could all significantly increase the expression of *Hba* and *Hbb* in chondrocytes in the conditions of either 20% or 1% O₂ (Fig. 3d, e, Extended Data Fig. 6). These results were further confirmed by chemical inhibition of Hif-1α with GN44028, or Hif-2α with PT2385 and PT2399, as well as treatment with activators of Hif-1/2α, such as IOX2, Roxadustat or DMOG (Extended Data Fig. 7). In Fact, a recent study by Feng et al showed that *Hif-1α* knockdown resulted in elevated expression of *Hbb* in erythroblasts [Feng et al, *Nature*, 2022, PMID: 36224385], providing an independent validation for our finding in this study.

3. They should prove that primary chondrocytes cultured in hypoxia and lacking HIF1a still display an increase in KLF1 expression.

Response: As suggested, we confirmed that knocking out *Hif-1α* and *Hif-2α*, either alone or in combination, could significantly increase *Klf1* expression in chondrocytes cultured in conditions of either 20% or 1% O₂ (Extended Data Fig. 9a-f).

4. Furthermore, the experiment shown in Figure 3 panel g should be repeated in hypoxia.

Response: As suggested, we repeated the experiment in Figure 3 panel g in hypoxia (1% O₂) and obtained similar results, that is, *Hba* and *Hbb* expression were downregulated upon *Klf1* depletion, which was shown as Extended Data Fig. 8d-f in the revised manuscript. Moreover, for further validation, we established a *Klf1*^{Flox/Flox} mouse line (Sup. Fig. 3), and found that conditional knockout of *Klf1* in chondrocytes resulted in decreased transcription of both *Hbb* and *Hba* under 20% or 1% O₂. The results were included in the revised manuscript as Figure 3g-h, and Extended Data Fig. 8a, b as well, to replace the original Fig. 3g that was moved into Extended Data (Extended Data Fig. 8c). Importantly, large areas of necrosis appeared in the center of cartilage tissues upon conditional knockout of *Klf1* (Sup. Fig.4), a phenotype resembling that of the *Hbb* knockout.

5. Chondrocyte death in Figure 4 panels a-b should be proven by TUNEL and appropriately quantified.

Response: We thank the referee for the comment. As suggested, Chondrocyte death in Figure 4a was confirmed by TUNEL staining and quantified as shown in Extended Data Fig. 12f. And Chondrocyte death in Figure 4b was confirmed by TUNEL staining and quantified as shown in Extended Data Fig. 15j-k.

6. Is the cell death secondary to loss of hemoglobin caspase-dependent?

Response: We thank the referee for the comment. To address this question, we examined the expression of cleaved caspase-3 in chondrocytes from *Hbb*-cKO mice. Though immunoblotting revealed the expression of cleaved caspase-3, no difference was detected across different genotypes (control, heterozygous and homozygous) of *Hbb* mice (Extended Data Fig. 18e-g), indicating little correlation between caspase-3 activation and cell death, thus arguing against a prominent role of caspase-dependent apoptosis in the cell death of chondrocytes upon hemoglobin depletion. Actually, our results fit well with the study by Yao et al, who also detected Caspase-3-independent cell death of chondrocytes induced by hypoxia [Yao et al, *Dev Cell*, 2019, PMID: 31105007].

7. More in general, growth plate development should be systematically studied in all the mouse models presented in the paper across different time points, and efficient recombination of the gene of interest should be carefully documented in each of those mouse models.

Response: As suggested, we systematically studied growth plate development in the mouse models across different time points, the related results were included as Extended Data Fig. 11, 12, 13, 15, 16 in the revised manuscript. And the shreds of evidence for efficient recombination of the gene of interest were presented together with each of the corresponding studies (Extended Data Fig. 11-13, 15, 16).

8. How do the authors explain the significant difference in P50 between red blood cells and chondrocytes (Figure 4 panel g)?

Response: We thank the referee for the question, which is a point that also interested us when we got the result. There are at least two likely mechanisms underlying the left-shift P50 in chondrocytes. First, the composition of hemoglobin was reported to be a key determinant of P50, while the native hetero-tetrameric hemoglobin in red blood cells has an optimized P50 that facilitates oxygen binding and releasing where needed, the *Hbb* monomer or unequilibrated hemoglobin tetramer with more *Hbb* displayed higher binding to

oxygen [Weber et al, *J Mol Biol*, 1995, PMID: 7666421, Papassotiriou et al, *Pediatr Hematol Oncol*, 1997, PMID: 9211537, Fabry et al, *Proc Natl Acad Sci U S A*, 1992, PMID: 1465454], rendering the left-shift of P50. Actually, as indicated by the mass spectrometry analysis, the hemoglobin proteins in chondrocytes are not produced in an equilibrium ratio, instead, the majority of them are Hbb with a rough ratio of (Hba : Hbb \approx 3 : 5), thus promoting the left-shift of P50. Second, the tight spatial conformation turned out to be another promoting factor of P50 left-shift [Zhang and Palmer et al, *Biotechnol Prog*, 2010, PMID: 20564360]. While in our case, hemoglobin protein in chondrocytes could form compact condensate via phase separation, thus would further increase the left-shift of P50. It's actually conceivable that increased P50 would facilitate the oxygen storage in chondrocytes within a hypoxia microenvironment to make a short-range supply when needed, thus sustaining chondrocyte survival.

9. At the end of the story, the reader is left with the issue of why loss of chondrocytic hemoglobin causes cell death. The authors imply that this is due to the increased hypoxia, which is likely. But, then why increased intracellular hypoxia does cause cell death? Along those lines, it is critical the authors measure ATP/ADP, pAMK/AMP, and mitochondrial ROS.

Response: We thank the referee for the question, which actually remains open for the field yet. A mostly related study was recently reported by the Schipani group, who also found that increased hypoxia, by enhanced mitochondrial oxidation resulting from Hif-1 α -depletion, could promote chondrocyte death, however, with the underlying mechanisms unclear [Yao et al, *Dev Cell*, 2019, PMID: 31105007]. They ruled out the involvement of factors such as intracellular ROS and ATP availability and found that the hypoxia-dependent cell death of Hif-1 α -deficient chondrocytes was Caspase-3-independent. It turns out that the cell death that occurred in chondrocytes in our study is resembling the one reported in their study in multiple ways. As addressed above, the cell death induced by Hbb-depletion is Caspase-3-independent (Extended Data Fig. 18e-g). Moreover, as suggested, we measured the indexes including ATP/ADP, pAMK/AMP, and mitochondrial ROS etc., and found that Hbb depletion neither decreased the intracellular ATP level, ATP/ADP ratio and the pAMPK/AMPK ratio as well (Extended Data Fig. 18a-b), or increased the total and mitochondrial ROS in normoxia or hypoxia conditions as compared to the controls (Extended Data Fig. 18c-d). Thus, it's unlikely that altered energy and ROS levels might play a dominant role in promoting chondrocyte death by Hbb depletion. Considering the complexity of cell death regulation, we may expect that deciphering the exact

mechanism underlying severe hypoxia-induced chondrocyte death would constitute an independent project to be addressed in the future.

10. Another possibility is that, paradoxically, prolonged activation of the HIFs due to augmented hypoxia increases cell death. To exclude this possibility, it would be helpful to use a temporally induced conditional knockout of *Hba* and show presence of cell death within 48 hrs/72 hrs from the induction.

Response: We thank the referee for putting forward this hypothesis that prolonged HIFs activation due to augmented hypoxia may promote cell death in the growth plate. As suggested, we performed the temporally induced conditional knockout of *Hbb* by tamoxifen treatment, and found that the *Hbb* level gradually declined to a significant level by 72 hrs post tamoxifen induction when a large amount of cell death took place in the growth plate (Extended Data Fig. 16a-c), which is consistent with the proposed hypothesis. However, treatment with HIFs activators, including IOX2, Roxadustat and DMOG, respectively, was all able to completely rescue the cartilage death induced by either *Hba* or *Hbb* deficiency under hypoxia (Extended Data Fig. 19a-d), strongly suggesting that HIFs activation, instead of promoting, may actually protect chondrocytes from cell death induced by augmented hypoxia upon *Hba/Hbb* knockout.

11. The TEXT has numerous typos and grammar issues that need to be corrected.

Response: We thank the referee for the comment. Accordingly, the final draft was sent to the language service for editing to kill the typos and grammar issues as much as possible before submission. Please find a certificate from the language service submitted as the supplemental file.

Referee #2 (Remarks to the Author):

A. Summary: Cartilage is poorly vascularized and as a result is a highly hypoxic microenvironment. The authors report the presence of cytosolic condensates of hemoglobin in chondrocytes from various murine and human sources. They provide evidence that the same globin switching that occurs in erythroid cells during development also occurs in chondrocytes. Finally, they demonstrated that globin gene expression was induced by exposure of chondrocytes to hypoxia, which they claimed was driven by hypoxia-induced expression of Klf1, a known transactivator of globin gene expression in erythroid cells.

B. Novelty. Although globin gene expression has been observed in many cell types, there was no good evidence that functional hemoglobin was produced in cells other

than erythrocytes. The authors propose that the hemoglobin condensate represents an oxygen storage depot in chondrocytes.

C. Most of the claims are well supported by the data but there are several key issues that need to be addressed.

Response: We thank the referee for the nice words.

1. Erythroid hemoglobin binds O₂ in the lungs, where the PO₂ is high and releases O₂ in peripheral tissues where PO₂ is low. In contrast, the hemoglobin condensates in chondrocytes are static: they remain in the same location with the same low PO₂ for their entire lifetimes. How then is O₂ picked up and released? Do O₂ levels vary in cartilage e.g. during exercise?

Response: We thank the referee for the comment. Actually, although cartilage is avascular, nutrients and gas (such as oxygen) are not evenly distributed, instead, there is a gradient for them [Yao et al, *Dev Cell*, 2019, PMID: 31105007, Egli et al, *Int Rev Cell Mol Biol*, 2011, PMID: 21749898]. Generally, the concentration is higher at the periphery region and lower at the middle region, which is also true for oxygen as validated in our study in Figure 4d. This is attributed to mitochondrial respiration [Yao et al, *Dev Cell*, 2019, PMID: 31105007], as well as other processes such as epigenetic modifications of histones, redox reactions in the endoplasmic reticulum, cytoplasmic translation of proteins, p53 activity, and the DNA damage response, which also consume oxygen [Liu and Simon et al, *Cancer Biol & Ther*, 2004, PMID, Fels and Koumenis et al, *Cancer Biol & Ther*, 2006, PMID, Olcina et al, *Mol Cell Oncol*, 2014, PMID, Salminen et al, *Cell Mol Life Sci*, 2015, PMID]. Similar to nutrients, oxygen is transported from periphery to the center via diffusion, which, interestingly, is facilitated by pressure differences induced by the cyclic motion and the resulting compaction and relief of the tissue [Mow et al, *Osteoarthritis Cartilage*, 1999, PMID: 10367014]. This would conceivably help drive the pick-up and releasing of oxygen within chondrocytes by the hemoglobin condensates. As a way of proof of concept, we designed a compaction assay as a surrogate of exercise, where the cartilages were centrifugated to increase pressure loading. As a result, we find that increased pressure did cause a hypoxia effect as evidenced by enhanced EF5 staining and upregulated HIF1a expression (Extended Data Fig. 17e-g), suggesting oxygen consumption in chondrocytes during exercise. Moreover, in another experiment where ATDC cells transfected with hemoglobin cultured in hypoxia condition (1% O₂), we found that the cells expressing hemoglobin are more tolerable to hypoxia as compared with adjacent cells without hemoglobin as indicated by nuclear

localization of HIF1a (Extended Data Fig. 21b-c). These data, together with the data provided in Fig.4h, i, suggest an essential role of hemoglobin condensates in local oxygen supply upon hypoxia within cells in cartilages.

2. The authors claim that the hypoxic induction of Klf1 and its subsequent activation of globin gene transcription occurs in a manner independent of hypoxia-inducible factors. How, then, is Klf1 expression induced? Klf1 expression has been shown to be upregulated in cells from patients who are homozygous for a missense mutation in VHL that impairs the O₂-dependent degradation of HIF-1alpha and HIF-2alpha (PMID: 23993337), strongly suggesting that hypoxia-induced expression of Klf1 is HIF-dependent. The authors should knock down the expression of HIF-1alpha AND HIF-2alpha in primary chondrocytes and analyze the effect of culturing the cells under high (20%) and low (1%) O₂ conditions or in the presence of the HIF stabilizer dimethylxallylglycine. The experiments should include analysis of known HIF target genes as positive controls.

Response: We thank the referee for the insightful comment and constructive advice. As suggested, we confirmed that knocking out *Hif-1α* and *Hif-2α*, either alone or in combination, could all significantly increase the expression of *Klf1* in the chondrocytes under either 20% or 1% O₂ conditions (note: the HIF target Epo was included as positive control) (Extended Data Fig. 9a-f), arguing against a positive role of HIFs in Klf upregulation in chondrocytes.

For the obvious discrepancy between our results and the hypothesis deduced from the study [Zhang et al, *Blood Cells Mol Dis*, 2014, PMID: 23993337] cited by the referee, there are two alternative explanations: first, yes, according to the cited study, it's likely that VHL mutation may upregulate Klf1 through HIFs, which, however, does not rule out the involvement of other VHL targets since VHL is an E3 ligase known to facilitate a large number of other molecules for degradation in addition to the well-known HIFs [Zhang and Zhang et al, *Biomedicines*, 2018, PMID: 29562667]. Second, the cited study worked on peripheral blood mononuclear cells, while our study worked on chondrocytes, the context-based difference of signaling transduction is another important factor affecting gene regulation and should be taken into account. Detail deciphering the underlying mechanisms is an interesting topic, however, beyond the scope of this study, warranting careful investigation in the future.

Next, for the question "How is Klf1 is induced via a HIFs-independent mechanism?" During the revision process, we managed to make progress on this point. Inspired by the fact that hypoxia could induce HIF-independent histone methylation via the inhibition of KDM5a demethylase, an oxygen sensor

[Batie et al, *Science*, 2019, PMID: 30872526], we validated that both hypoxia and depletion of KDM5a, but not KDM5b, led to increased H3K4me3 methylation of the *Klf1* locus, resulting in the increased expression of *Klf1*. These results suggest that the inactivation of KDM5a by hypoxia mediates *Klf1* upregulation, which was included in the revised manuscript as Extended Data Fig. 9g-l and Extended Data Fig.10.

3. The characterization of some of the mouse mutants was not adequate -- "pale" and "purplish" are crude descriptions. Do these animals have normal circulating RBC counts and hemoglobin levels? Is there increased tissue hypoxia? If so, what is the mechanism for these systemic findings?

Response: We thank the referee for the comment and apologize for the crude descriptions, which were removed from the revised manuscript. To make a more sophisticated phenotype description, as suggested, we examined the peripheral blood of P5 mice with different genotypes and found that *Hba*^{+/-}, *Hbb*^{+/-} or *Hbb*^{F/F}/*Prx1*-Cre mice exhibited reduced red blood cell numbers and hemoglobin content, and also increased tissue hypoxia (Extended Data Fig. 14a-j). These systemic phenotypes resemble those of mouse models for thalassemias, where red blood cell count and hemoglobin were decreased, which was attributed, at least partially, to increased hemolysis of defective red blood cells [Yang et al, *Proc Natl Acad Sci U S A*, 1995, PMID: 8524813, Pászty et al, *Nat Genet*, 1995, PMID: 7550311]. To exclude the effects of the systemic anemia-related hypoxia on the survival of chondrocytes in the center of growth plates, the *Col2a1*-Cre^{ERT2} mice were crossed with *Hbb*^{F/F} mice to produce chondrocyte specific tamoxifen-induced *Hbb* knockout (*Hbb*-cKO) mice, which exhibited normal levels of red blood cell count and hemoglobin content, no anemia and hypoxia in other tissues, such as liver and muscle (Extended Data Fig. 14k-p, Extended Data Fig. 15a-c). Histological examination detected increased and massive dead chondrocytes in the center of the cartilages of the P1 and P5 *Hbb*-cKO mice (Fig. 4b, Extended Data Fig. 15f-l), indicating a crucial role of *Hbb* in the survival of chondrocytes in the center of growth plates. The conclusion was further supported by the hypoxia tolerance experiment by culturing humeral cartilage growth plates in hypoxic environment (1% O₂) *in vitro* (Fig. 4f, Extended Data Fig. 20a-c).

Referee #3 (Remarks to the Author):

I am an expert in biological phase separation and higher-order protein assembly, so I

will only comment on the data claiming that Hemoglobin bodies form through phase separation.

In this paper, Zhang et al. make a provocative claim that Beta globin is sequestered into micron-scale condensates, called “Hemoglobin bodies”, inside mouse chondrocytes. They claim that Hemoglobin bodies assemble through hypoxia-induced phase separation of Beta-globin. If true, this finding could provide an explanation for how poorly vascularized tissues survive low-oxygen stress. However, the findings are extremely counterintuitive and contradict much of what we know about Hemoglobin structure. Much more work is required to understand the assembly mechanism of Hemoglobin bodies and rule out plausible alternative explanations. Thus, I do not recommend publication at this time.

Major Comments:

1. To me, the conclusion that Beta-globin can condense into micron-scale droplets is counterintuitive. Max Perutz won a Nobel prize in 1962 for solving the 3D structure of Hemoglobin; the structure of many hemoglobin subunits have been studied since, and they have confirmed the compact, globular structure described by Perutz. X-ray crystallography requires concentrating protein to extremely high levels, which is needed to form crystals. It also requires ordered packing of the molecules to make a crystal lattice. One cannot use X-ray crystallography to study phase-separating proteins because, at high concentrations, the proteins will form liquid droplets or hydrogels with too much disorder to create a useful crystal diffraction pattern. Thus, all evidence to date indicates that Hemoglobin is highly structured and does not form liquid droplets.

Response: We thank the referee for the comment. In order to make a professional addressing of this issue, we consulted a structural scientist at CAS Center for Excellence in Biomacromolecules, Institute of Biophysics, Chinese Academy of Sciences, Beijing, China — Dr. Jingjin Ding (jd Ding@ibp.ac.cn), who had published a dozen of papers in high profile journals including Nature, Cell and Science etc. in recent years, and is an experienced expert in structural biology. He advised that, in most cases, proteins/molecules only efficiently make crystal lattices under a specified condition where the concentrations for proteins, salt and other assistant factors are carefully controlled. The same is true for phase-separation to be taken place. And the conditions for crystal formation and phase-separation are unlikely, or at least infrequently, the same. In fact, based on his experiences of protein crystallization, proteins could exist in four major forms of being either solved, or precipitated as inclusion body, or phase-separated as liquid droplet, or crystalized, depending on the conditions the proteins are in. Thus, when a

protein/molecule could be crystalized under one condition does not rule out the possibility that it could form liquid droplet by phase separation under another condition. Consistent with this idea, we check the PDB database for proteins that were reported to undergo LLPS to form liquid droplet, such as cGAS and FUS [Xie et al, *Proc Natl Acad Sci U S A*, 2019, PMID: 31142647, Murray et al, *Cell*, 2017, PMID: 28942918], they each also have a corresponding 3-D structure reported based on analysis of either the crystal lattices or fibril aggregates (6EDC.pdb for cGAS by X-RAY DIFFRACTION; 5W3N.pdb for FUS by solid state NMR). Thus, in theory, hemoglobin is not denied the ability to form liquid droplets as further validated with the data provided subsequently in the following addressing.

2. The authors' claim that mouse Beta-globin has an intrinsically disordered domain on its C-terminus is a dangerous and misleading over-interpretation. This unstructured region is 5 amino acids long (see <https://alphafold.ebi.ac.uk/entry/A8DUK4>). By that metric, almost every protein in the proteome has an intrinsically disordered tail region. Most well-described intrinsically disordered proteins (e.g., NPM1, FUS, P body proteins) have disordered regions containing 100's of amino acids. In my 10 years of studying protein condensation I have never seen a protein that could phase separate using a disordered tail of only 5 amino acids.

Response: We thank the referee for the insightful comment. Yes, the classical intrinsically disordered region (IDR) is about 100 amino acids for most of the reported LLPS proteins as the referee pointed out. The IDR sequences for Hbb in this study were identified by a different online server (<https://d2p2.pro/>), which reported a N-terminal sequence (1-9 aa) and a C-terminal sequence (140-147 aa). Since functional experiment validated the impacts of the C-terminal sequence on condensate formation of Hbb (truncation of either 139-147 aa or 140-147 aa displayed the same phenotype) (Fig. 2i-k), we proposed that the short C-terminal sequence might represent a type of non-classical IDR that is functional in small proteins like Hbb.

3. The authors need to rule out plausible alternative explanations for how Hemoglobin bodies form. It is possible that Hbb is binding to the surface of a neutral lipid droplet or some other large structure. Neutral lipid droplets are surrounded by phospholipid monolayers, and they can look exactly like the big droplets shown in figure 2h (see figure 5C from this paper: <https://www.molbiolcell.org/doi/pdf/10.1091/mbc.E19-05-0284>). That could explain the large droplet-like morphologies and the results of the droplet fusion experiment in

figure 2e. Turnover may be caused by binding and unbinding from the lipid droplet surface. The C-terminal tail of Hbb may be a binding motif to allow this.

Response: We thank the referee for the constructive comment. To test whether the structures we identified are lipid droplets surrounded by phospholipid monolayers. We performed correlative light electron microscopy and immunoelectron transmission microscopy on the *Hbb*-transfected cells, which indicated that the hemoglobin bodies observed by confocal microscope are membraneless condensate structures clearly different from lipid droplets that are surrounded by phospholipid monolayers (Fig. 2h and Extended Data Fig. 5). The result indicates that the structures we identified is distinct from the phospholipid monolayer-enclosed lipid droplets and the double-leaflet membrane-enclosed organelles as well. For further confirmation, we stained the Hbb transfected cells with Bodipy to indicate lipid droplets, confocal microscopic imaging showed that the hemoglobin body are not colocalized with lipid droplets at all. The same is true for the staining of mouse cartilage (Extended Data Fig. 2b-c). Thus, these results argue against the hypothesis that the hemoglobin bodies we detected are lipid droplets surrounded by Hbb binding on the surface.

4. Suggested experiments to strengthen the manuscript:
 - a. Reconstitute Hbb droplets *in vitro* using purified Hbb. This will be a definitive test of whether Hbb is sufficient to phase separate.

Response: We thank the referee for the constructive comment. Yes, we agree with the referee that reconstitute liquid droplet formation *in vitro* with purified Hbb is a definitive experiment to prove phase separation. Accordingly, we first titrated the purified untagged Hbb protein in 10% PEG200 solutions of different salt concentrations, and found an optimized condition (150mM $\text{KH}_2\text{PO}_4/\text{K}_2\text{HPO}_4$; pH=7.35; 10% PEG2000) for phase separation of Hbb. Under this condition, Hbb droplets started to form at the concentration of 125 ng/ul, reaching an efficient formation at the concentrations of 250 ng/ul and 500 ng/ul (Extended Data Fig. 3a). We then examined the formation of purified Hba and its combination of different ratios with Hbb at the optimized condition, it turned out Hba alone, instead of forming liquid droplets, tended to precipitate, but could form liquid droplets in the presence of Hbb (Extended Data Fig. 3b). To further confirm the formation of liquid droplets by Hbb, we performed timelapse imaging to track the droplet fusion, indicating that these Hbb droplet could efficiently fused with each other rapidly (Extended Data Fig. 3c). Altogether, the available data support that Hbb is able to phase separate *in vitro* as it did *in vivo*.

- b. Assess the lipid and nucleic acid content of the extracted Hemoglobin bodies. Simple stains are a good way to start (e.g., Nile Red or BODIPY for lipids, RNA GREEN stains for RNA). Mass spectrometry-based lipidomics would be the next step.

Response: We thank the referee for the constructive comment. We stained the Hbb transfected cells with Bodipy to indicate lipid droplets, and RNA Green to indicate RNA, and DAPI to indicate DNA. Confocal microscopy showed that hemoglobin body are not colocalized with either lipid droplets or RNA/DNA at all. The same is true for the staining of mouse cartilage (Extended Data Fig. 2b-c). Moreover, overexposure imaging showed that the Bodipy signals for lipid and RNA Green signals for RNA actually display a pattern mutually exclusive to the hemoglobin body (Extended Data Fig. 2b-c, upper panels), arguing against that lipid and nucleic acids are enriched components of hemoglobin body. Considering that Hbb alone could efficiently form liquid droplets *in vitro* (Extended Data Fig. 3a-c), we were discouraged to expect a fruitful result from the lipidomics analysis as suggested.

- c. Figure 2k needs a control for expression levels. It is possible that mutations in the C-terminus of Hbb cause expression levels to drop, thus preventing accurate analysis of localization.

Response: We thank the referee the advice. Accordingly, we examined the expression of the truncated mutants used in Figure 2k by Western blot, which reported comparable protein levels of different mutants (Extended Data Fig. 4c), suggesting it's unlikely that the reduced foci formation of Hbb mutants was secondary to their altered expression levels.

- d. Provide a control to show that the localization of Hbb is not an artifact of the GFP tag on the C-terminus. Repeat Figure 2c using immunofluorescence of cells expressing untagged Hbb.

Response: We thank the referee for the insightful comment. As suggested, we repeated Figure 2c by immunostaining cells expressing untagged Hbb with anti-Hbb antibody, and observed typical Hbb foci as shown in Extended Data Fig. 2b-c. Moreover, we used the purified untagged Hbb proteins to reconstitute liquid droplets *in vitro* under an optimized condition as shown in Extended Data Fig. 3a-c. For further validation, we purified the GFP-tagged Hbb protein and use it to also successfully reconstitute droplet formation *in vitro* as shown in Extended Data Fig. 4a-b. Thus, these results supported Hbb could form droplets *in vitro* and in cells *in vivo*.

Reference

1. Yao Q, Khan MP, Merceron C, LaGory EL, Tata Z, Mangiavini L, *et al.* Suppressing Mitochondrial Respiration Is Critical for Hypoxia Tolerance in the Fetal Growth Plate. *Dev Cell* 2019, **49**(5): 748-763.e747.
2. Batie M, Frost J, Frost M, Wilson JW, Schofield P, Rocha S. Hypoxia induces rapid changes to histone methylation and reprograms chromatin. *Science* 2019, **363**(6432): 1222-1226.
3. Feng R, Mayuranathan T, Huang P, Doerfler PA, Li Y, Yao Y, *et al.* Activation of γ -globin expression by hypoxia-inducible factor 1 α . *Nature* 2022, **610**(7933): 783-790.
4. Weber RE, Malte H, Braswell EH, Oliver RW, Green BN, Sharma PK, *et al.* Mass spectrometric composition, molecular mass and oxygen binding of Macrobdella decora hemoglobin and its tetramer and monomer subunits. *J Mol Biol* 1995, **251**(5): 703-720.
5. Papassotiriou I, Kanavakis E, Stamoulakatou A, Kattamis C. Tissue oxygenation in patients with hemoglobinopathy H. *Pediatr Hematol Oncol* 1997, **14**(4): 323-334.
6. Fabry ME, Nagel RL, Pachnis A, Suzuka SM, Costantini F. High expression of human beta S- and alpha-globins in transgenic mice: hemoglobin composition and hematological consequences. *Proc Natl Acad Sci U S A* 1992, **89**(24): 12150-12154.
7. Zhang N, Palmer AF. Polymerization of human hemoglobin using the crosslinker 1,11-bis(maleimido)triethylene glycol for use as an oxygen carrier. *Biotechnol Prog* 2010, **26**(5): 1481-1485.
8. Egli RJ, Wernike E, Grad S, Luginbühl R. Physiological cartilage tissue engineering effect of oxygen and biomechanics. *Int Rev Cell Mol Biol* 2011, **289**: 37-87.
9. Liu L, Simon MC. Regulation of Transcription and Translation by Hypoxia. *Cancer Biol & Ther* 2004, **3**(6): 492-497.
10. Fels DR, Koumenis C. The PERK/eIF2 α /ATF4 module of the UPR in hypoxia resistance and tumor growth. *Cancer Biol & Ther* 2006, **5**(7): 723-728.
11. Olcina MM, Grand RJA, Hammond EM. ATM activation in hypoxia - causes and consequences. *Mol Cell Oncol* 2014, **1**(1): e29903.
12. Salminen A, Kauppinen A, Kaarniranta K. 2-Oxoglutarate-dependent dioxygenases are sensors of energy metabolism, oxygen availability, and iron homeostasis: potential role in the regulation of aging process. *Cell Mol Life Sci* 2015, **72**(20): 3897-3914.
13. Mow VC, Wang CC, Hung CT. The extracellular matrix, interstitial fluid and ions as a mechanical signal transducer in articular cartilage. *Osteoarthritis Cartilage* 1999, **7**(1): 41-58.
14. Zhang X, Zhang W, Ma SF, Miasniakova G, Sergueeva A, Ammosova T, *et al.* Iron deficiency modifies gene expression variation induced by augmented hypoxia sensing. *Blood Cells Mol Dis* 2014, **52**(1): 35-45.
15. Zhang J, Zhang Q. VHL and Hypoxia Signaling: Beyond HIF in Cancer.

Biomedicines 2018, **6**(1).

16. Yang B, Kirby S, Lewis J, Detloff PJ, Maeda N, Smithies O. A mouse model for beta 0-thalassemia. *Proc Natl Acad Sci U S A* 1995, **92**(25): 11608-11612.
17. Pászty C, Mohandas N, Stevens ME, Loring JF, Liebhaber SA, Brion CM, *et al.* Lethal alpha-thalassaemia created by gene targeting in mice and its genetic rescue. *Nat Genet* 1995, **11**(1): 33-39.
18. Xie W, Lama L, Adura C, Tomita D, Glickman JF, Tuschl T, *et al.* Human cGAS catalytic domain has an additional DNA-binding interface that enhances enzymatic activity and liquid-phase condensation. *Proc Natl Acad Sci U S A* 2019, **116**(24): 11946-11955.
19. Murray DT, Kato M, Lin Y, Thurber KR, Hung I, McKnight SL, *et al.* Structure of FUS Protein Fibrils and Its Relevance to Self-Assembly and Phase Separation of Low-Complexity Domains. *Cell* 2017, **171**(3): 615-627.e616.

Reviewer Reports on the First Revision:

Referees' comments:

Referee #1 (Remarks to the Author):

This is a revised manuscript on the role of hemoglobin in chondrocyte and its role in cellular adaptation to hypoxia.

The authors have satisfactorily addressed reviewers' comments and concerns.

Referee #2 (Remarks to the Author):

Fantastic work. The authors are to be congratulated on the quantity and quality of the data presented in this manuscript, which strongly support the conclusion that chondrocytes express globin proteins in phase-separated cytoplasmic bodies as a means of maintaining oxygen homeostasis in cartilage. This is a major histological and physiological finding.

Referee #3 (Remarks to the Author):

Again, as my expertise is with protein phase separation, I will only comment on the conclusion that Hedy forms through LLPS. Overall, I am satisfied with the effort made by the authors to address my concerns. Although I was initially skeptical that a well structured protein like Hbb could phase separate, the authors have provided convincing data in the revisions to support their conclusion. I am in favor of publication.

I have only a few minor points that would strengthen the manuscript. Given the profound implications of this study, and that there will be many (like me) who will initially doubt that Hemoglobins could phase separate, it is necessary to add a few controls and measurements.

1. Show coomassie stained gels of the purified proteins to evaluate purity.
2. Remove the C-terminal IDR of Hbb and test if this mutant protein still phase separates in vitro. This would be the most definitive way to demonstrate that the short IDR is sufficient to drive LLPS.
3. Determine minimal concentration of PEG required for Hbb phase separation. I know from extensive experience that high levels of PEG forces unphysiological interactions. For example, alpha/beta tubulin will condense into amorphous aggregates in 10% PEG. The authors need to show how Hbb assembles (or not) in 0,2,4,6, and 8% PEG.
4. Line 122. The statement that IDRs are the motifs required for phase separation is incorrect. LLPS was originally characterized in a synthetic system built on interactions between structured domains (SH3-PRM modules; Li and Rosen, 2012 Nature). Many other non-IDRs have since been shown to drive LLPS. Please restate this to say, “. . .disordered regions (IDR), which are often enriched in phase separating proteins, . . .”.

Author Rebuttals to First Revision:

An Extra-erythrocyte Role of Hemoglobin body in Hypoxia Adaption of Chondrocytes

(Manuscript ID: **2021-11-18288**)

We thank the Editor and the Reviewers for your recognition and recommendation. We have performed experiments according to the third reviewer's advices and obtained positive results, which were updated in revised manuscript. Please find below a point-by-point response.

Referee #3 (Remarks to the Author):

Again, as my expertise is with protein phase separation, I will only comment on the conclusion that Hedy forms through LLPS. Overall, I am satisfied with the effort made by the authors to address my concerns. Although I was initially skeptical that a well structured protein like Hbb could phase separate, the authors have provided convincing data in the revisions to support their conclusion. I am in favor of publication.

I have only a few minor points that would strengthen the manuscript. Given the profound implications of this study, and that there will be many (like me) who will initially doubt that Hemoglobins could phase separate, it is necessary to add a few controls and measurements.

1. Show coomassie stained gels of the purified proteins to evaluate purity.

Response: We thank the referee for the thoughtful comment. We included the Coomassie stained gels of the purified proteins as **Extended Data Fig. 3f and 4a** in the revised manuscript, which indicated that the purified proteins have ideal purity.

2. Remove the C-terminal IDR of Hbb and test if this mutant protein still phase separates in vitro. This would be the most definitive way to demonstrate that the short IDR is sufficient to drive LLPS.

Response: We thank the referee the constructive advice. Accordingly, we examined the droplet formation of the truncated mutants *in vitro*. Consistent with the results in cells, the droplet formation of Hbb protein decreased upon the short IDR truncation or A139P mutation, which was included as **Extended Data Fig. 3e** in the revised manuscript. The text was modified accordingly (page 4, line 131).

3. Determine minimal concentration of PEG required for Hbb phase separation. I know from extensive experience that high levels of PEG forces unphysiological interactions. For example, alpha/beta tubulin will condense into amorphous

aggregates in 10% PEG. The authors need to show how Hbb assembles (or not) in 0,2,4,6, and 8% PEG.

Response: We thank the referee for the constructive comment. We examined the droplet formation of Hbb protein in the suggested PEG concentrations *in vitro*. It turns out that the minimal PEG concentration for Hbb droplet formation is 6%, which was included as Extended Data Fig. 3b in the revised manuscript.

4. Line 122. The statement that IDRs are the motifs required for phase separation is incorrect. LLPS was originally characterized in a synthetic system built on interactions between structured domains (SH3-PRM modules; Li and Rosen, 2012 Nature). Many other non-IDRs have since been shown to drive LLPS. Please restate this to say, “. . .disordered regions (IDR), which are often enriched in phase separating proteins, . . .”.

Response: We thank the reviewer for the professional advice. We have rephrased the sentence as suggested (page 4, line 124) and cited the quoted reference in the revised manuscript.

Reviewer Reports on the Second Revision:

Referees' comments:

Referee #3 (Remarks to the Author):

Overall, I am pleased with author's work. They have done all of the revision experiments I asked. I am in favor of publication.

However, I strongly suggest more transparency in reporting the results in the main text concerning Extended Data Fig. 3e (line 131). The data clearly show that Hbb mutants can still phase separate, although to a lesser extent than wild-type Hbb. This could be due to the high concentration of PEG, which can overpower the system and drive unphysiological assembly. I don't think this is a significant issue for this study as the overall conclusion is still supported.

But, the authors must state what was observed and not gloss over it. So I recommend changing line 131 to state something like:

"These mutations impaired, but did not completely prevent, condensation of Hbb in vitro (Extended Data Fig. 3e). Thus, the C-terminal IDR . . ."

Other than than change in text, I recommend no further revisions.